# A druggable copper-signalling pathway that drives inflammation

Stéphanie Solier[1,22], Sebastian Müller[1,22], Tatiana Cañeque[1,22], Antoine Versini[1,22], Arnaud Mansart[2], Fabien Sindikubwabo[1], Leeroy Baron[1], Laila Emam[2], Pierre Gestraud[3], G. Dan Pantoş[4], Vincent Gandon[5,6], Christine Gaillet[1], Ting-Di Wu[7,8], Florent Dingli[9], Damarys Loew[9], Sylvain Baulande[10], Sylvère Durand[11], Valentin Sencio[12], Cyril Robil[12], François Trottein[12], David Péricat[13], Emmanuelle Näser[13,14], Céline Cougoule[13], Etienne Meunier[13], Anne-Laure Bègue[15], Hélène Salmon[15], Nicolas Manel[15], Alain Puisieux[1], Sarah Watson[16], Mark A. Dawson[17,18], Nicolas Servant[3], Guido Kroemer[11,19,20], Djillali Annane[2,21] & Raphaël Rodriguez[1 ✉]

Inflammation is a complex physiological process triggered in response to harmful stimuli[1]. It involves cells of the immune system capable of clearing sources of injury and damaged tissues. Excessive inflammation can occur as a result of infection and is a hallmark of several diseases[2–4]. The molecular bases underlying inflammatory responses are not fully understood. Here we show that the cell surface glycoprotein CD44, which marks the acquisition of distinct cell phenotypes in the context of development, immunity and cancer progression, mediates the uptake of metals including copper. We identify a pool of chemically reactive copper(II) in mitochondria of inflammatory macrophages that catalyses NAD(H) redox cycling by activating hydrogen peroxide. Maintenance of NAD$^+$ enables metabolic and epigenetic programming towards the inflammatory state. Targeting mitochondrial copper(II) with supformin (LCC-12), a rationally designed dimer of metformin, induces a reduction of the NAD(H) pool, leading to metabolic and epigenetic states that oppose macrophage activation. LCC-12 interferes with cell plasticity in other settings and reduces inflammation in mouse models of bacterial and viral infections. Our work highlights the central role of copper as a regulator of cell plasticity and unveils a therapeutic strategy based on metabolic reprogramming and the control of epigenetic cell states.

Inflammation is a complex physiological process that enables clearance of pathogens and repair of damaged tissues. However, uncontrolled inflammation driven by macrophages and other immune cells can result in tissue injury and organ failure. Effective drugs against severe forms of inflammation are scarce[5,6], and there is a need for therapeutic innovation[7].

The plasma membrane glycoprotein CD44 is the main cell surface receptor of hyaluronates[8–10]. It has been associated with biological programmes[11] that involve cells capable of acquiring distinct phenotypes independently of genetic alterations, which is commonly defined as cell plasticity[12,13]. For instance, inflammatory macrophages are marked by increased expression of CD44 and its functional implication in this context has been demonstrated[14,15]. However, the mechanisms by which CD44 and hyaluronates influence cell biology remain elusive[14,16–18]. The recent discovery that CD44 mediates the endocytosis of iron-bound hyaluronates in cancer cells links membrane biology to the epigenetic regulation of cell plasticity, where increased iron uptake promotes the activity of α-ketoglutarate (αKG)-dependent demethylases involved in the regulation of gene expression[19]. Hyaluronates have been shown to induce the expression of pro-inflammatory cytokines in alveolar macrophages (AMs)[20], and macrophage activation relies on complex regulatory mechanisms occurring at the chromatin level[21–23]. This body

[1]Equipe Labellisée Ligue Contre le Cancer, Institut Curie, CNRS, INSERM, PSL Research University, Paris, France. [2]Paris Saclay University, UVSQ, INSERM, 2I, Montigny-le-Bretonneux, France. [3]CBIO-Centre for Computational Biology, Institut Curie, INSERM, Mines ParisTech, Paris, France. [4]Department of Chemistry, University of Bath, Bath, UK. [5]Institut de Chimie Moléculaire et des Matériaux d'Orsay, CNRS, Paris Saclay University, Orsay, France. [6]Laboratoire de Chimie Moléculaire, CNRS, Ecole Polytechnique, Institut Polytechnique de Paris, Palaiseau, France. [7]Institut Curie, PSL Research University, Paris, France. [8]Multimodal Imaging Center, Paris Saclay University, CNRS, INSERM, Orsay, France. [9]CurieCoreTech Mass Spectrometry Proteomic, Institut Curie, PSL Research University, Paris, France. [10]ICGex Next-Generation Sequencing Platform, Institut Curie, PSL Research University, Paris, France. [11]Metabolomics and Cell Biology Platforms, Institut Gustave Roussy, Villejuif, France. [12]Université de Lille, CNRS, INSERM, CHU Lille, Institut Pasteur de Lille, CIIL, Lille, France. [13]Institof of Pharmacology and Structural Biology, University of Toulouse, CNRS, Toulouse, France. [14]Cytometry and Imaging Core facility, Institute of Pharmacology and Structural Biology, University of Toulouse, CNRS, Toulouse, France. [15]Institut Curie, INSERM, PSL Research University, Paris, France. [16]Department of Medical Oncology, Institut Curie, PSL Research University, Paris, France. [17]Peter MacCallum Cancer Centre and Sir Peter MacCallum Department of Oncology, Melbourne, Victoria, Australia. [18]Centre for Cancer Research, University of Melbourne, Melbourne, Victoria, Australia. [19]Centre de Recherche des Cordeliers, University of Paris, Sorbonne University, INSERM, Institut Universitaire de France, Paris, France. [20]Institut du Cancer Paris CARPEM, Department of Biology, Hôpital Européen Georges Pompidou, AP-HP, Paris, France. [21]Department of Intensive Care, Hôpital Raymond Poincaré, AP-HP, Garches, France. [22]These authors contributed equally: Stéphanie Solier, Sebastian Müller, Tatiana Cañeque, Antoine Versini. ✉e-mail: raphael.rodriguez@curie.fr

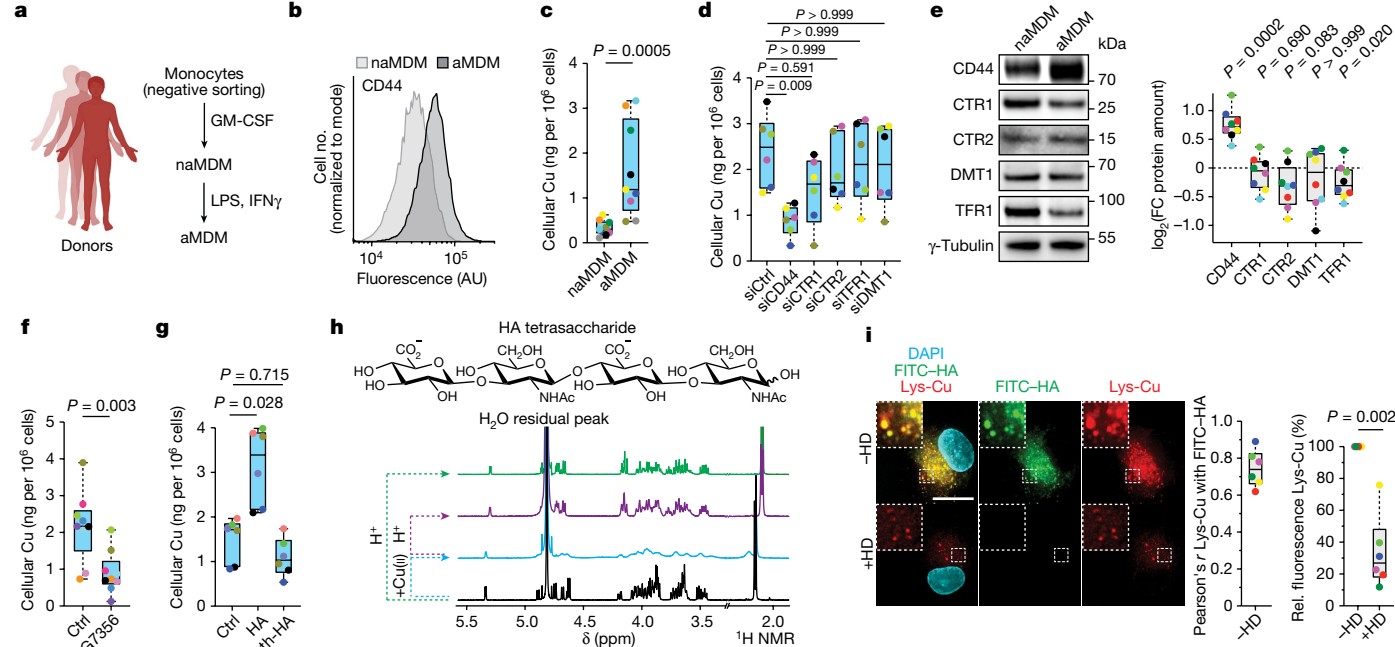

**Fig. 1 | CD44 mediates copper uptake. a**, Experimental setup used to generate inflammatory monocyte-derived macrophages (MDMs). **b**, Flow cytometry of CD44 in MDMs. Data are representative of $n = 13$ donors. AU, arbitrary units. **c**, ICP-MS of cellular copper in MDMs ($n = 9$ donors). **d**, ICP-MS of cellular copper in aMDMs with short interfering RNA (siRNA) knockdown of indicated receptors and transporters ($n = 6$ donors). Copper transporter 1 (CTR1) is encoded by *SLC31A1*, CTR2 is encoded by *SLC31A2*, transferrin receptor 1 (TFR1) is encoded by *TFRC*, and divalent metal transporter 1 (DMT1) is encoded by *SLC11A2*. siCtrl, control siRNA. **e**, Representative western blots of metal transporters in MDMs ($n = 7$ donors). FC, fold change. **f**, ICP-MS of cellular copper in MDMs treated with anti-CD44 antibody RG7356 during activation ($n = 7$ donors). **g**, ICP-MS of cellular copper in MDMs treated with hyaluronate (0.6–1 MDa) (HA) or permethylated hyaluronate (meth–HA) during activation ($n = 6$ donors). **h**, Molecular structure of hyaluronate tetrasaccharide (top) and $^1$H NMR spectra (bottom) of copper–hyaluronate complexation experiment, recorded at 310 K in $D_2O$. **i**, Fluorescence microscopy of a lysosomal copper(II) probe (Lys-Cu) and FITC–hyaluronate in aMDMs treated with hyaluronidase (HD). At least 30 cells were quantified per donor ($n = 6$ donors). Scale bar, 10 μm. Rel., relative. **c**,**e**,**f**,**i**, Two-sided Mann–Whitney test. **d**,**g**, Kruskal–Wallis test with Dunn's post test. In all box plots in the main figures, boxes represent the interquartile range, centre lines represent medians and whiskers indicate the minimum and maximum values. In graphs, each coloured dot represents an individual donor for a given panel.

of work raises the question of whether a general mechanism involving CD44-mediated metal uptake regulates macrophage plasticity and inflammation.

Here we show that macrophage activation is characterized by an increase of mitochondrial copper(II), which occurs as a result of CD44 upregulation. Mitochondrial copper(II) catalyses NAD(H) redox cycling, thereby promoting metabolic changes and ensuing epigenetic alterations that lead to an inflammatory state. We developed a metformin dimer that inactivates mitochondrial copper(II). This drug induces metabolic and epigenetic shifts that oppose macrophage activation and dampen inflammation in vivo.

## CD44 mediates cellular uptake of copper

To study the role of metals in immune cell activation, we generated inflammatory macrophages using human primary monocytes isolated from blood (Fig. 1a). Activated monocyte-derived macrophages (aMDMs) were characterized by the upregulation of CD44, CD86 and CD80, together with a distinct cell morphology (Fig. 1b and Extended Data Fig. 1a–c).

Using inductively coupled plasma mass spectrometry (ICP-MS), we detected higher levels of cellular copper, iron, manganese and calcium in aMDMs compared with non-activated MDMs (naMDMs) (Fig. 1c and Extended Data Fig. 1d). In contrast to other metal transporters, knocking down CD44 antagonized metal uptake (Fig. 1d and Extended Data Fig. 1e,f) and, unlike CD44, levels of these other metal transporters did not increase upon macrophage activation (Fig. 1e). Of note, levels of

these transporters remained unchanged under CD44-knockdown conditions (Extended Data Fig. 1g). Treating MDMs with an anti-CD44 antibody[24] antagonized metal uptake upon activation (Fig. 1f and Extended Data Fig. 2a). Conversely, supplementing cells with hyaluronate upon activation increased metal uptake, whereas addition of a permethylated hyaluronate[25], which is less prone to metal binding, had no effect (Fig. 1g and Extended Data Fig. 2b). Inflammatory macrophages were also characterized by the upregulation of hyaluronate synthases (HAS) and the downregulation of the copper export proteins ATP7A and ATP7B (Extended Data Fig. 2c). Nuclear magnetic resonance revealed that hyaluronate interacts with copper(II) and that this interaction can be reversed by lowering the pH (Fig. 1h). Fluorescence microscopy showed that labelled hyaluronate colocalized with a lysosomal copper(II) probe[26] in aMDMs (Fig. 1i). Cotreatment with hyaluronidase—which degrades hyaluronates—or knocking down CD44 reduced lysosomal copper(II) staining (Fig. 1i and Extended Data Fig. 2d). In aMDMs, the copper transporter CTR2 colocalized with the endolysosomal marker LAMP2, and CTR2 knockdown led to increased lysosomal copper(II) staining (Extended Data Fig. 2e,f). Collectively, these data indicate that in aMDMs, CD44 mediates the endocytosis of specific metals bound to hyaluronate, including copper.

## Mitochondrial Cu(II) regulates cell plasticity

We evaluated the capacity of copper(I) and copper(II) chelators, including ammonium tetrathiomolybdate (ATTM), D-penicillamine (D-Pen), EDTA and trientine to interfere with macrophage activation.

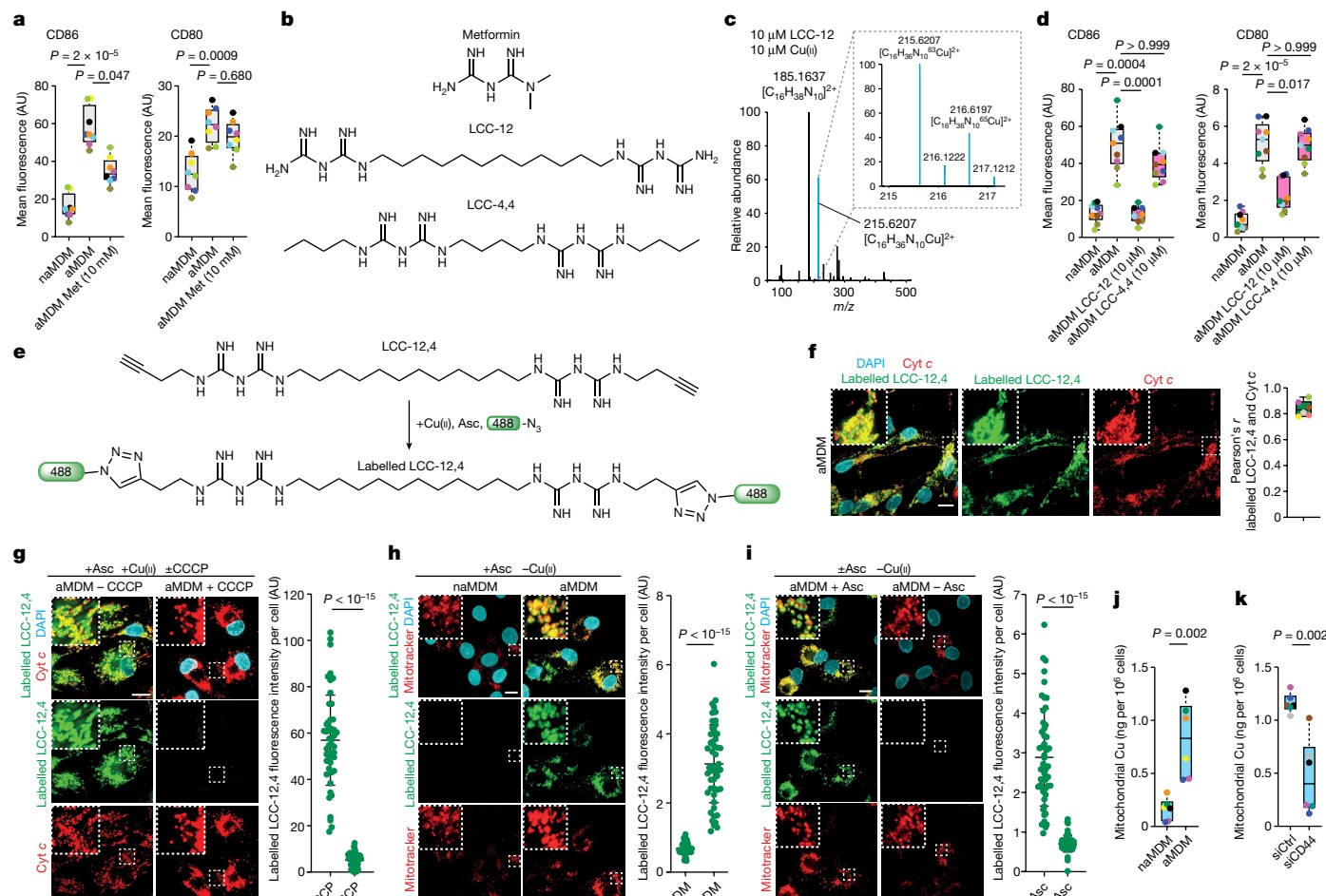

**Fig. 2 | Development of a small molecule inactivator of mitochondrial copper(II). a**, Flow cytometry of cell surface markers in MDMs treated with metformin (Met) (n = 8 donors). **b**, Molecular structures of metformin, LCC-12 and LCC-4,4. **c**, HRMS of a Cu–LCC-12 complex. **d**, Flow cytometry of cell surface markers in MDMs treated with LCC-12 or LCC-4,4 (n = 9 donors). **e**, Experimental procedure of in-cell labelling of LCC-12,4. **f**, Fluorescence microscopy of labelled LCC-12,4 in aMDMs (n = 6 donors). At least 50 cells were quantified per donor. Cyt c, cytochrome c. **g**, Fluorescence microscopy of labelled LCC-12,4 in aMDMs treated with CCCP. **h**, Fluorescence microscopy of labelled LCC-12,4 in MDMs.

In-cell labelling is performed with ascorbate and without added copper(II). **i**, Fluorescence microscopy of labelled LCC-12,4 in aMDMs. In-cell labelling is performed in the presence or absence of ascorbate (asc) and without added copper(II). **j**, ICP-MS of mitochondrial copper in MDMs (n = 6 donors). **k**, ICP-MS of mitochondrial copper in aMDMs under CD44-knockdown conditions (n = 6 donors). **g**–**i**, Two-sided unpaired t-test, representative of n = 3 donors. Data are mean ± s.d. **a**,**d**, Kruskal–Wallis test with Dunn's post test. **j**,**k**, Two-sided Mann–Whitney test. In graphs, each coloured dot represents an individual donor for a given panel. Scale bars, 10 μm.

We also studied metformin, a biguanide used for the treatment of type-2 diabetes, because it can form a bimolecular complex[27] with copper(II). Metformin partially antagonized CD86 upregulation, albeit at high concentrations, in contrast to the marginal effects of other copper-targeting molecules (Fig. 2a and Extended Data Fig. 3a).

To reduce the entropic cost inherent to the formation of bimolecular Cu(Met)$_2$ complexes[27,28], we tethered two biguanides with methylene-containing linkers to produce the lipophilic copper clamps LCC-12 and LCC-4,4 (Fig. 2b), which contain 12 and 4 linking methylene groups, respectively. LCC-4,4 displays distal butyl substituents to exhibit a lipophilicity similar to that of LCC-12. We compared simulated structures of copper(II) complexes with the lowest energies using molecular dynamics and discrete Fourier transform with a Cu(Met)$_2$ complex using the crystal structure of the latter as benchmark[28] (Extended Data Fig. 3b). Cu–LCC-12 adopted a geometry similar to that of Cu(Met)$_2$, whereas Cu–LCC-4,4 lacked bonding angle symmetry and exhibited imine–copper bonds out of plane. The calculated free energy of Cu–LCC-4,4 was 16.6 kcal mol$^{-1}$ higher than that of Cu–LCC-12, suggesting that Cu–LCC-4,4 is a less stable copper(II) complex. High-resolution mass spectrometry (HRMS) confirmed the formation

of monometallic copper biguanide complexes, with Cu–LCC-12 being the most stable (Fig. 2c and Extended Data Fig. 3c). LCC-12 did not form stable complexes with other divalent metal ions (Extended Data Fig. 3d). A reduction in the UV absorbance of LCC-12 upon addition of copper(II) chloride indicated complex formation at low micromolar concentrations. This was confirmed by the appearance of coloured solutions characteristic of metal complexes (Extended Data Fig. 3e,f). Notably, even at a 1,000-fold lower dose, LCC-12 antagonized the induction of CD86 and CD80 in aMDMs more potently than metformin (Fig. 2d). The effect of LCC-4,4 used at 10 μM was moderate, consistent with the reduced capacity of this analogue to form a complex with copper(II). As reported for metformin[29], LCC-12 induced AMPK phosphorylation, albeit at a much lower concentration, suggesting that phenotypes induced by metformin are linked to copper(II) targeting (Extended Data Fig. 3g).

Next, we evaluated the effect of LCC-12 on other cell types that can upregulate CD44 upon exposure to specific biochemical stimuli. LCC-12 interfered with the activation of dendritic cells and T lymphocytes and the expression of several cell surface molecules on alternatively activated macrophages (Extended Data Fig. 4a,b). By contrast, LCC-12 did not interfere with the activation of neutrophils, a process that is

not marked by CD44 upregulation. Copper signalling has previously been linked to cancer progression[30–32]. Human non-small cell lung carcinoma cells and mouse pancreatic adenocarcinoma cells undergoing epithelial–mesenchymal transition (EMT)—a cell biology programme that can promote the acquisition of the persister cancer cell state and metastasis[12,33]—were characterized by CD44 upregulation and increased cellular copper. Consistently, LCC-12 interfered with EMT, as shown by the levels of the epithelial marker E-cadherin, mesenchymal markers vimentin and fibronectin, the EMT transcription factors Slug and Twist as well as the levels of pro-metastatic protein CD109 (Extended Data Fig. 4c,d). These data support a general mechanism involving copper that regulates cell plasticity.

Nanoscale secondary ion mass spectrometry (NanoSIMS) imaging of aMDMs revealed a subcellular localization of the isotopologue $^{15}N,^{13}C$-LCC-12 that overlapped with the signals of $^{197}Au$-labelled cytochrome $c$, suggesting that LCC-12 targets mitochondria (Extended Data Fig. 5a,b). Fluorescent in-cell labelling of the biologically active but-1-yne-containing analogue LCC-12,4 using click chemistry[34] gave rise to a cytoplasmic staining pattern that colocalized with cytochrome $c$ (Fig. 2e,f). The mitochondrial staining of LCC-12,4 was reduced upon cotreatment with carbonyl cyanide chlorophenylhydrazone (CCCP), a small molecule that dissipates the inner mitochondrial proton gradient, indicating that LCC-12 accumulation in mitochondria is driven by its protonation state (Fig. 2g). Labelling alkyne-containing small molecules in cells requires a copper(I) catalyst generated in situ from added copper(II) and ascorbate[34–37]. We investigated whether the mitochondrial copper(II) content in aMDMs would allow in-cell labelling without the need to experimentally add a copper catalyst. Fluorescent labelling of LCC-12,4 used at a concentration of 100 nM, which is lower than the biologically active dose of LCC-12, occurred in aMDMs in the absence of added copper(II) and a strong staining was observed only in aMDMs when ascorbate was used for labelling (Fig. 2h,i). Furthermore, the fluorescence intensity of labelled LCC-12,4 was reduced when a 100-fold molar excess of LCC-12 was used as a competitor (Extended Data Fig. 5c). These data support the existence of a druggable pool of chemically reactive copper(II) in mitochondria. Consistent with this, levels of copper increased in mitochondria upon macrophage activation together with those of manganese (Fig. 2j and Extended Data Fig. 5d,e), whereas levels of copper in the endoplasmic reticulum and nucleus remained unaltered (Extended Data Fig. 5f,g). Notably, aMDMs were characterized by an increase of nuclear iron, hinting at an increased activity of αKG-dependent demethylases as previously shown in cancer cells undergoing EMT[19] (Extended Data Fig. 5f). LCC-12 treatment did not alter the total cellular and mitochondrial copper content of aMDMs, indicating that LCC-12 does not act as a cuprophore[38] (Extended Data Fig. 5h,i). By contrast, LCC-12 reduced the fluorescence of a mitochondrial copper(II) probe[39] in aMDMs, supporting direct copper binding in mitochondria (Extended Data Fig. 5j). Notably, the mitochondrial metal transporters SLC25A3 and SLC25A37 were upregulated in aMDMs (Extended Data Fig. 5k). Knocking down the expression of these transporters or CD44 did not reduce labelled LCC-12,4 fluorescence (Extended Data Fig. 5l–n), whereas knocking down CD44 led to marked reduction of mitochondrial copper (Fig. 2k). This indicates that, unlike the proton gradient, mitochondrial copper does not drive mitochondrial accumulation of biguanides. As a control, labelling an alkyne-containing derivative of the copper(II) chelator trientine, which did not exhibit a potent effect against macrophage activation, revealed nuclear accumulation, providing a rationale for the lack of biological activity of this and potentially other copper-targeting drugs in this context (Extended Data Fig. 5o,p).

## Cu(II) regulates NAD(H) redox cycling

Higher mitochondrial levels of manganese in aMDMs pointed to a functional role of the superoxide dismutase 2 (SOD2) in the context of macrophage activation. The amount of SOD2 protein increased in mitochondria upon activation, whereas the amount of catalase decreased (Fig. 3a,b). Mitochondrial hydrogen peroxide, a product of superoxide dismutase and substrate of catalase, increased accordingly (Fig. 3c,d). In cell-free systems, copper(II) can catalyse the reduction of hydrogen peroxide by various organic substrates[40,41]. In the presence of copper(II), NADH reacted with hydrogen peroxide to yield NAD+, whereas the absence of copper(II) yielded a complex mixture of oxidation products (Fig. 3e and Extended Data Fig. 6a). Consistently, copper(II) favoured the conversion of 1-methyl-1,4-dihydronicotinamide (MDHNA), a structurally less complex surrogate of NADH, into 1-methylnicotinamide (MNA+), whereas a product of epoxidation was formed preferentially in the absence of copper (Extended Data Fig. 6b,c). Thus, copper(II) redirects the reactivity of hydrogen peroxide towards NADH. Under reaction conditions similar to those found in mitochondria, NADH was rapidly consumed to yield NAD+ in the presence of copper(II) (Fig. 3f and Extended Data Fig. 6d). This reaction was inhibited by LCC-12, whereas the effects of LCC-4,4 and metformin were marginal (Fig. 3f). Molecular modelling supported a reaction mechanism in which copper(II) activates hydrogen peroxide, facilitating its reduction through the transfer of a hydride from NADH (Extended Data Fig. 6e). Copper(II) acts as a catalyst that lowers the energy of the transition state with a geometry favouring this reaction. Molecular modelling also supported the inactivation of this reaction by biguanides through direct copper(II) binding (Extended Data Fig. 6f).

Mitochondrial NADH levels were higher and NAD+ levels were lower in aMDMs compared with naMDMs, suggesting an enhanced activity of mitochondrial enzymes reliant on NAD+ (Fig. 3g and Supplementary Table 1). Treating MDMs with LCC-12 during activation led to a reduction of NADH and NAD+ (Fig. 3g and Supplementary Table 1). This suggests that copper(II) catalyses the reduction of hydrogen peroxide by NADH to produce NAD+ and that biguanides can interfere with this redox cycling, leading instead to other oxidation by-products (Fig. 3e). NADH and copper were found in mitochondria of aMDMs at an estimated substrate:catalyst ratio of 2:1, which is even more favourable for this reaction to take place than the 20:1 ratio used in the cell-free system (Extended Data Fig. 6g). Macrophage activation was accompanied by altered levels of several metabolites whose production depends on NAD(H) (Fig. 3h and Supplementary Table 2). LCC-12-induced metabolic reprogramming of aMDMs was marked by a reduction of αKG and acetyl-coenzyme A (acetyl-CoA) (Fig. 3i). LCC-12 also caused a reduction of extracellular lactate and accumulation of glyceraldehyde 3-phosphate in aMDMs consistent with the reduced activity of NAD+-dependent glyceraldehyde 3-phosphate dehydrogenase (Extended Data Fig. 6h,i). Collectively, these data support the central role of mitochondrial copper(II) in the maintenance of a pool of NAD+ that regulates the metabolic state of inflammatory macrophages.

## Mitochondrial Cu(II) regulates transcription

Transcription is co-regulated by chromatin-modifying enzymes, whose expression levels and recruitment at specific genomic loci shape gene expression. The turnover of specific enzymes such as iron-dependent demethylases and acetyltransferases relies on αKG and acetyl-CoA[42]. The finding that LCC-12 interfered with the production of these metabolites and opposed macrophage activation pointed to epigenetic alterations that affect the expression of inflammatory genes. We analysed the transcriptomes of aMDMs versus those of naMDMs by RNA sequencing (RNA-seq) (Supplementary Table 3) and compared them to transcriptomics data obtained from bronchoalveolar macrophages of individuals infected with severe acute respiratory syndrome coronavirus 2 (SARS-CoV-2)[43] and from human macrophages exposed in vitro to *Salmonella typhimurium*[44], *Leishmania major*[45] or *Aspergillus fumigatus*[46] (Supplementary Table 4). Gene ontology (GO) analysis revealed three groups of GO terms comprising upregulated genes, belonging

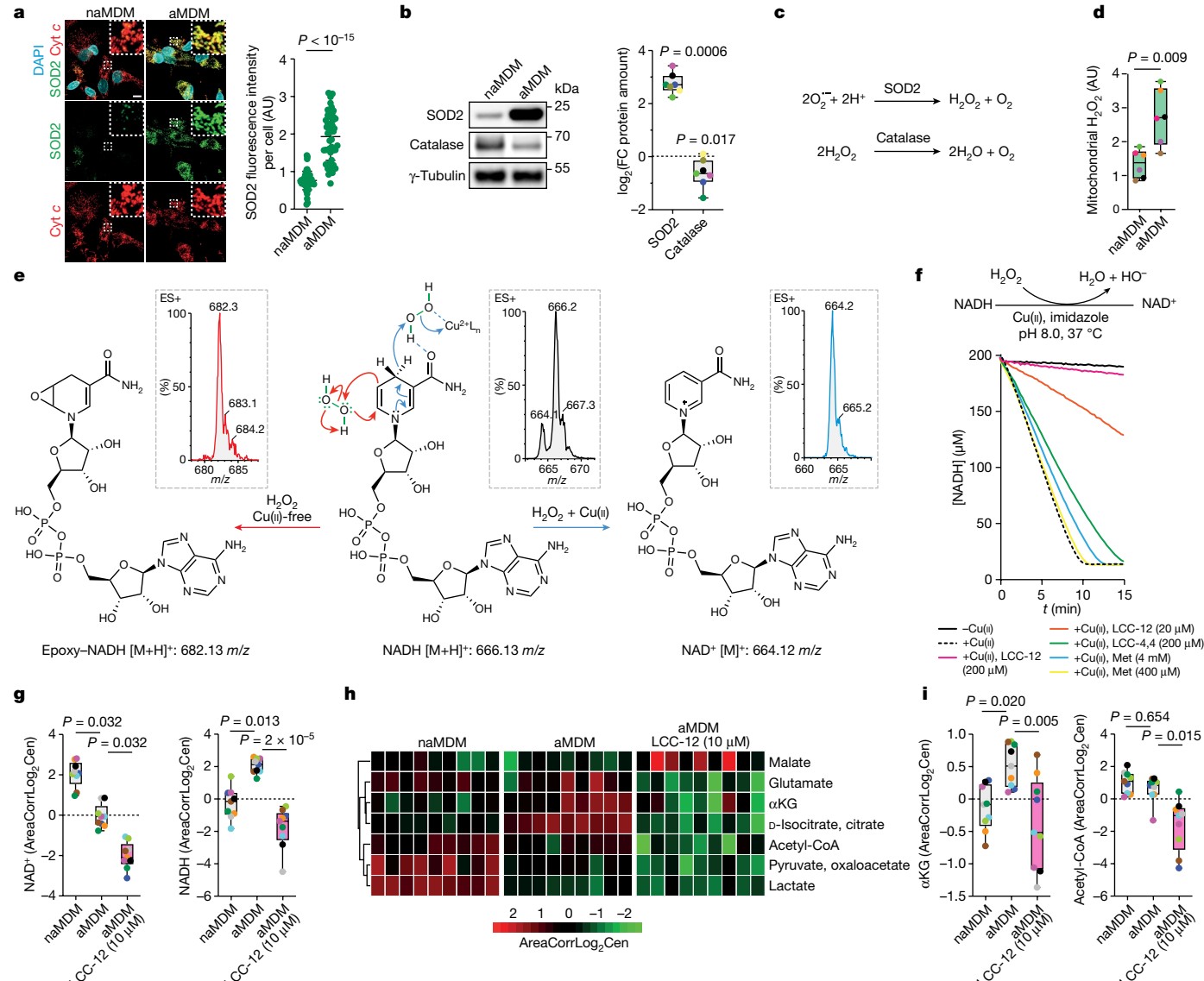

**Fig. 3 | Mitochondrial copper(II) regulates NAD(H) redox cycling.**
**a**, Fluorescence microscopy of SOD2 in MDMs. Representative of $n = 4$ donors.
At least 50 cells were quantified per donor. Scale bar, 10 μm. Two-sided
unpaired $t$-test. Data are mean ± s.d. **b**, Representative western blots of SOD2
and catalase in MDMs ($n = 7$ donors). **c**, Regulation of $H_2O_2$ levels by SOD2 and
catalase. **d**, Flow cytometry of mitochondrial $H_2O_2$ in MDMs ($n = 6$ donors).
**e**, Reaction of NADH with $H_2O_2$ under copper(II)-catalysed or copper-free
conditions. Experimental mass spectrometry peaks and calculated masses of
molecular ions are indicated. ES+, electrospray ionization mass spectrometry.
**f**, Kinetics of NADH oxidation in the presence of $H_2O_2$ and copper(II). Data are
representative of $n = 3$ independent experiments. **g**, Metabolomics of NAD⁺
and NADH of mitochondria from MDMs treated with LCC-12 ($n = 9$ donors).
AreaCorrLog₂Cen data correspond to raw areas, corrected for analytical bias
using GRMeta R package, then corrected areas are log₂ transformed and
centered on means. **h**, Metabolomics heat map highlighting metabolites
whose biosynthesis is dependent on NAD(H) in MDMs treated with LCC-12
($n = 9$ donors). **i**, Metabolomics of αKG and acetyl-CoA of MDMs treated with
LCC-12 ($n = 9$ donors). **b**,**d**, Two-sided Mann–Whitney test. **g**,**i**, Kruskal–Wallis
test with Dunn's post test. In graphs, each coloured dot represents an individual
donor for a given panel.

to inflammation, metabolism and chromatin (Fig. 4a). Notably, the GO
terms of these genes included endosomal transport, cellular response
to copper ion, response to hydrogen peroxide and positive regulation
of mitochondrion organization. Similar signatures were obtained for
macrophages exposed to distinct pathogens (Extended Data Fig. 7a,b
and Supplementary Table 5), as defined by GO terms and increased RNA
amounts for genes involved in inflammation (Fig. 4b and Extended
Data Fig. 7c). aMDMs exhibited upregulated genes encoding CD44,
sorting nexin 9 (SNX9), a regulator of CD44 endocytosis, and metal-
lothioneins (MT2A and MT1X) involved in copper transport and storage,
whereas expression levels of *ATP7A* and *ATP7B* were downregulated
(Supplementary Table 3). Genes involved in chromatin and histone
modifications were upregulated in aMDMs and similar genes encoding

iron-dependent demethylases and acetyltransferases were upregu-
lated in bronchoalveolar macrophages from individuals infected with
SARS-CoV-2, as well as in macrophages exposed to other pathogens
(Fig. 4c and Extended Data Fig. 7d). These data indicate that distinct
classes of pathogens trigger similar epigenetic alterations[47], leading
to the inflammatory cell state. In aMDMs, variations in protein lev-
els including increases in iron-dependent demethylases and acetyl-
transferases were consistent with the RNA-seq data (Extended Data
Fig. 8a,b and Supplementary Table 6). Changes in levels of specific
demethylases and acetyltransferases were associated with alterations
of their targeted marks (Extended Data Fig. 8c,d). Chromatin immu-
noprecipitation sequencing (ChIP–seq) revealed a global increase of
the permissive acetyl marks H3K27ac, H3K14ac and H3K9ac together

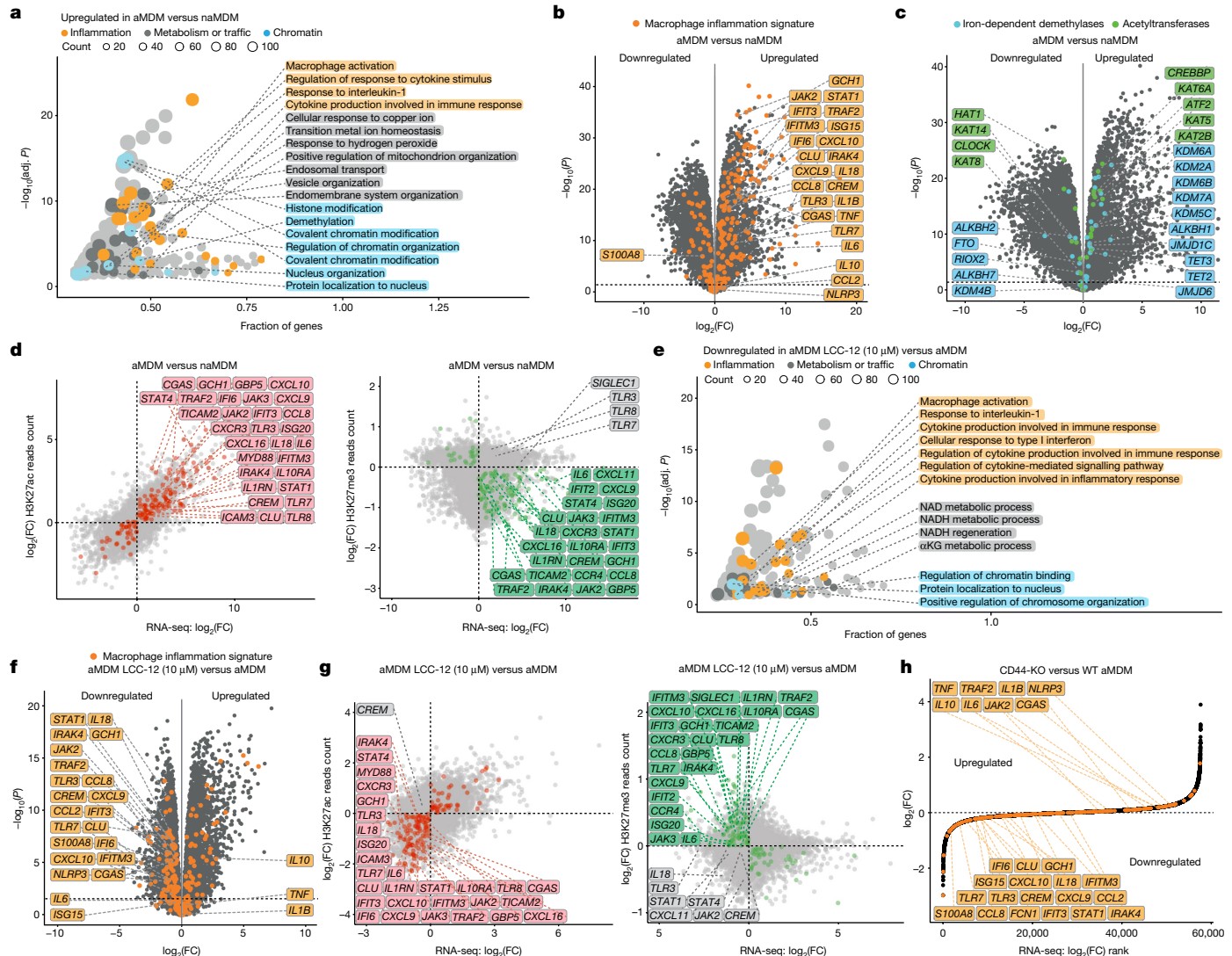

**Fig. 4 | Mitochondrial copper(II) regulates the epigenetic states and transcriptional programmes of inflammatory macrophages. a**, GO term analysis of upregulated genes in aMDMs ($n = 10$ donors). Adj. $P$, adjusted $P$ value. **b**, RNA-seq analysis of MDMs. Macrophage inflammatory signature genes are highlighted. The dashed line indicates an adjusted $P$ value of 0.05 ($n = 10$ donors). **c**, RNA-seq analysis of MDMs. Iron-dependent demethylase and acetyltransferase signature genes are highlighted. The dashed line indicates an adjusted $P$ value of 0.05 ($n = 10$ donors). **d**, Correlation for a representative donor of ChIP–seq reads count of histone marks in genes against RNA-seq of gene transcripts in MDMs ($n = 10$ donors). **e**, GO term analysis of genes in aMDMs ($n = 10$ donors) whose expression levels are downregulated upon treatment with LCC-12 ($n = 5$ donors). **f**, RNA-seq analysis of aMDMs ($n = 10$ donors) and MDMs treated with LCC-12 during activation ($n = 5$ donors). Macrophage inflammatory signature genes are highlighted. The dashed line indicates an adjusted $P$ value of 0.05. **g**, Correlation for a representative donor of ChIP–seq reads count of histone marks in genes against RNA-seq of gene transcripts in aMDMs ($n = 10$ donors) and MDMs treated with LCC-12 during activation ($n = 5$ donors). **h**, RNA-seq analysis of CD44-knockout (KO) and wild-type (WT) aMDMs. Representative of $n = 4$ donors. Gating strategy is shown in the Supplementary Information. Macrophage inflammatory signature genes are highlighted. The dashed line indicates an adjusted $P$ value of 0.05. In **a–c**, **e**, **f**, Differential gene expression was assessed with the limma/voom framework. GO enrichment was assessed with the enrichGO method from clusterProfiler. $P$ values were corrected for multiple testing with the Benjamini–Hochberg procedure.

with a reduction of repressive methyl marks H3K27me3 and H3K9me2 at inflammatory gene loci, with consistent effects on the transcriptional profile of aMDMs (Fig. 4d, Extended Data Fig. 8e,f and Supplementary Table 7). LCC-12 treatment induced a downregulation of genes related to NAD(H) and αKG metabolism, regulation of chromatin and inflammation (Fig. 4e and Supplementary Table 8). Inactivating mitochondrial copper(II) also promoted the downregulation of inflammatory genes at the RNA and protein levels (Fig. 4f, Extended Data Fig. 9a,b and Supplementary Table 3), reflecting a complex epigenetic reprogramming toward a distinct cell state (Extended Data Fig. 9c). LCC-12 treatment reduced H3K27ac, H3K14ac and H3K9ac and increased H3K27me3 and H3K9me2 levels (Extended Data Fig. 9d),

which was associated with the downregulation of targeted inflammatory genes (Fig. 4g, Extended Data Fig. 9e,f and Supplementary Table 7). Thus, the LCC-12-induced decreases in αKG and acetyl-CoA were associated with a reduced activity of iron-dependent demethylases and acetyltransferases, respectively. Notably, knocking down expression of SOD2 or the mitochondrial copper transporter SLC25A3 reduced the inflammatory signature of macrophages (Extended Data Fig. 9g,h). Similarly, knocking out CD44 antagonized epigenetic programming of inflammation in aMDMs without adversely affecting the expression of other metal transporters (Fig. 4h, Extended Data Fig. 9i–k and Supplementary Table 9). Together, these data indicate that hydrogen peroxide is a driver of cell plasticity and that mitochondrial copper(II) controls

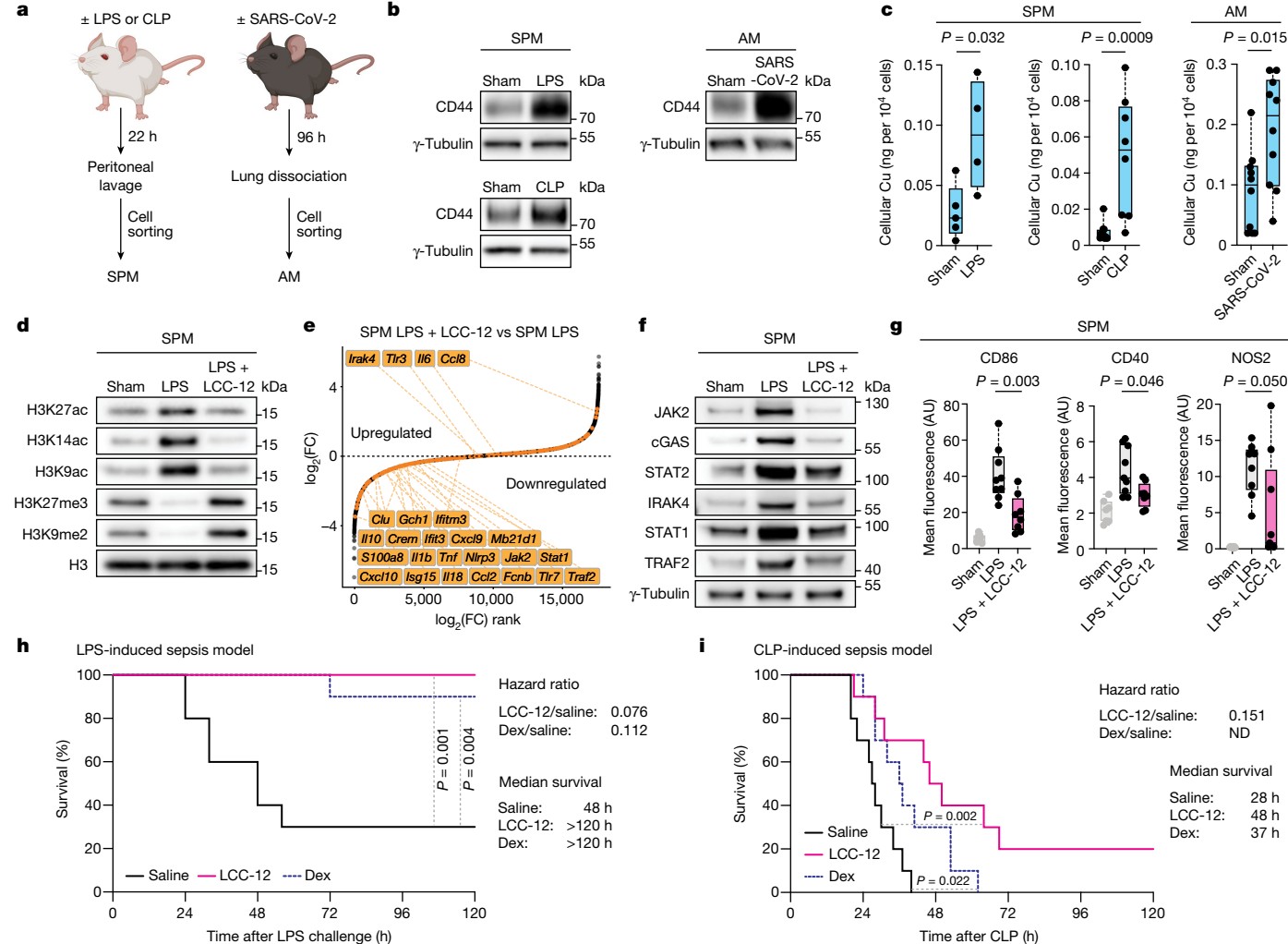

**Fig. 5 | Pharmacological inactivation of mitochondrial copper(II) attenuates inflammation in vivo. a**, Experimental setup to isolate SPMs and AMs. The gating strategy is shown in the Supplementary Information. **b**, Western blots of CD44 in inflammatory macrophages isolated from mice. Macrophages from 7–10 mice were pooled per condition. **c**, ICP-MS of cellular copper in SPMs or AMs from control mice (sham) and mice undergoing acute inflammation. LPS: sham ($n = 5$ mice), LPS-treated ($n = 4$ mice); CLP: sham ($n = 10$ mice), with CLP ($n = 8$ mice); SARS-CoV-2: sham ($n = 10$ mice), SARS-CoV-2-infected ($n = 10$ mice). **d**, Western blots of histone marks in SPMs from mice treated with LPS and LCC-12. Macrophages from 4–7 mice pooled per condition. H3 is a sample processing control. **e**, Rank-order plot for RNA-seq of SPMs from mice treated with LPS and LCC-12. **f**, Western blots of proteins involved in inflammation in SPMs from mice treated with LPS and LCC-12. Macrophages from 4–7 mice were pooled per condition. **g**, Flow cytometry of SPMs from mice treated with LPS and LCC-12 ($n = 7$–9 mice). **h**, Kaplan–Meier survival curves of mice treated with LPS (20 mg kg$^{-1}$ per single dose; intraperitoneal injection; $n = 10$ mice) and LCC-12 (0.3 mg kg$^{-1}$ 2 h before challenge, then 24 h, 48 h, 72 h and 96 h after challenge; intraperitoneal injection; $n = 10$ mice) or dexamethasone (10 mg kg$^{-1}$ per dose 1 h before challenge; oral gavage; $n = 10$ mice). **i**, Kaplan–Meier survival curves of mice subjected to CLP and treated with LCC-12 (0.3 mg kg$^{-1}$ 4 h, 24 h, 48 h, 72 h and 96 h after CLP; intraperitoneal injection; $n = 10$ mice), dexamethasone (1.0 mg kg$^{-1}$ at time of CLP; intraperitoneal injection; $n = 10$ mice) or a saline solution (intraperitoneal injection; $n = 10$ mice). In **d**–**g**, LCC-12 (0.3 mg kg$^{-1}$) was injected 6 h after LPS and samples were collected 22 h after LPS. **c**,**g**, Two-sided Mann–Whitney test. **h**,**i**, Mantel–Cox log-rank test. Hazard ratio calculated using the Mantel–Haenszel method. ND, not determined.

the availability of essential metabolic intermediates required for the activity of chromatin-modifying enzymes, which enables rapid transcriptional changes underlying the acquisition of distinct cell states.

## Cu(II) inactivation reduces inflammation

We investigated the role of copper signalling in well-established mouse models of acute inflammation: (1) endotoxaemia induced by lipopolysaccharide (LPS), reflecting our mechanistic model of macrophage activation; (2) cecal ligation and puncture (CLP), which recapitulates the pathophysiology of subacute polymicrobial abdominal sepsis occurring in humans[48]; and (3) a model of viral infection, namely SARS-CoV-2. The inflammatory states of small peritoneal

macrophages (SPMs) isolated from LPS and CLP mice models, and AMs isolated from SARS-CoV-2-infected mice, were characterized by upregulated CD44 and increased cellular copper (Fig. 5a–c). Effectors of the copper-signalling pathway, including HAS and SOD2 as well as specific epigenetic modifiers, were also upregulated in inflammatory macrophages (Extended Data Fig. 10a–c). Histone mark targets of these epigenetic modifiers were altered accordingly (Fig. 5d and Extended Data Fig. 10b,c). Intraperitoneal administration of LCC-12 in LPS-treated mice caused reductions in H3K27ac, H3K14ac and H3K9ac and increases in H3K27me3 and H3K9me2 (Fig. 5d), which were associated with reduced inflammation (Fig. 5e–g and Supplementary Table 10). In line with this result, intraperitoneal administration of LCC-12 fully protected mice from LPS-induced death and prevented the reduction

of body temperature (Fig. 5h and Extended Data Fig. 10d), performing better than high-dose dexamethasone, which is used for the clinical management of acute inflammation. In CLP-induced sepsis, LCC-12 also increased the survival rate (Fig. 5i). LCC-12 administered by inhalation to SARS-CoV-2-infected K18-hACE2 mice altered the expression of genes involved in the regulation of chromatin and downregulated the expression of inflammatory genes (Extended Data Fig. 10e,f and Supplementary Tables 11 and 12). Together, these data indicate that targeting mitochondrial copper(II) interferes with the acquisition of the inflammatory state in vivo and confers therapeutic benefits.

## Discussion

CD44 has previously been linked to development, immune responses and cancer progression. Here we have shown that CD44 mediates the cellular uptake of specific metals, including copper, thereby regulating immune cell activation. We identified a chemically reactive pool of copper(II) in mitochondria that characterizes the inflammatory state of macrophages. Our data support a cellular mechanism whereby the activation of hydrogen peroxide by copper(II) enables oxidation of $NADH$ to replenish the pool of $NAD^+$. Maintenance of this redox cycling is required for the production of key metabolites that are essential for epigenetic programming. In this context, copper(II) acts directly as a metal catalyst, in contrast to its dynamic metalloallosteric effect in other processes[30]. Transcriptomic shifts in macrophages exposed to distinct classes of pathogens substantiate the general nature of this mechanism.

We designed a dimer of biguanides that is able to inactivate mitochondrial copper(II), thereby triggering metabolic and epigenetic reprogramming that reduces the inflammatory cell state and increases survival in preclinical models of acute inflammation. We have thus illustrated the pathophysiological relevance of this copper(II)-triggered molecular chain of events. Acute inflammation is therefore reminiscent of a metabolic disease that can be rebalanced by targeting mitochondrial copper(II) to restrict the generation of key metabolites required to initiate and maintain the inflammatory state (Extended Data Fig. 10g). LCC-12 selectively targets mitochondrial copper(II), which is more abundant in the disease inflammatory cell state than in the basal state. This drug also interfered with the process of EMT in cancer cells, supporting a wider role for this copper-signalling pathway in the regulation of transcriptional changes beyond inflammation. Thus, CD44 may be characterized as a regulator of cell plasticity. Metformin exhibits positive effects on human health and is being studied as an anti-ageing drug[49,50]. However, investigation of its mechanism of action is hampered by its poor pharmacology resulting in low potency, which necessitates the administration of high doses. Thus LCC-12—which we rename 'supformin'—exhibits improved biological and preclinical characteristics over metformin, making it a suitable drug-like small molecule for revealing novel mechanistic features of biguanides. Overall, our findings highlight the central role of mitochondrial copper(II) as a regulator of cell plasticity and unveil a therapeutic strategy based on the control and fine-tuning of epigenetic cell states.

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

# Methods

## Ethics statement

Peripheral blood samples were collected from 128 healthy donors at Etablissement Français du Sang (EFS). The use of EFS blood samples from anonymous donors was approved by the Institut National de la Santé et de la Recherche Médicale committee. Written consent was obtained from all the donors. Survival assessment using the LPS mouse model was conducted at Fidelta according to 2010/63/EU and national legislation regulating the use of laboratory animals in scientific research and for other purposes (Official Gazette 55/13). An institutional committee on animal research ethics (CARE-Zg) oversaw that animal-related procedures were not compromising the animal welfare. Flow cytometry, ICP-MS, western blotting and RNA-seq using the LPS mouse models were performed in accordance with French laws concerning animal experimentation (#2021072216346511) and approved by Institutional Animal Care and Use Committee of Université de Saint-Quentin-en-Yvelines (C2EA-47). All animal work using the CLP model was conducted in accordance with French laws concerning animal experimentation (#2021072216346511) and approved by Institutional Animal Care and Use Committee of Université de Saint-Quentin-en-Yvelines (C2EA-47). All animal work concerning RNA-seq on the SARS-CoV-2 model was performed within the biosafety level 3 facility of the Institut Pasteur de Lille, after validation of the protocols by the local committee for the evaluation of the biological risks and complied with current national and institutional regulations and ethical guidelines (Institut Pasteur de Lille/B59-350009). The experimental protocols using animals were approved by the institutional ethical committee Comité d'Ethique en Experimentation Animale (CEEA) 75, Nord-Pas-de-Calais. The animal study was authorized by the Education, Research and Innovation Ministry under registration number APAFIS#25517-2020052608325772v3. Animal work concerning cytometry, ICP-MS and western blotting on the SARS-CoV-2 model was performed within the biosafety level 3 facility of the University of Toulouse. This work was overseen by an Institutional Committee on Animal Research Ethics (license APAFIS#27729–2020101616517580 v3, Minister of Research, France (CEEA-001)), to ensure that animal-related procedures were not compromising the animal welfare.

## Antibodies

Antibodies are annotated below as follows. WB, western blot; FCy, flow cytometry; FM, fluorescence microscopy; NS, NanoSIMS; ChIP, ChIP–seq; Hu, used for human samples; Ms, used for mouse samples. Dilutions are indicated. Any antibody validation by manufacturers is indicated and can be found on the manufacturers' websites. Our antibody validation by knockdown (KD) and/or KO strategies as described here for relevant antibodies is indicated. Primary antibodies: ALKBH1 (Abcam, ab195376, clone EPR19215, lot GR262105-2, WB 1:1,000, Hu, Ms, KO validated by manufacturer), AMP-activated protein kinase subunit alpha (AMPKα, Cell Signaling, 2532S, lot 21, WB 1:1,000, Hu), p-AMPKα phosphorylated on Thr172 (p-AMPKα, Cell Signaling, 2535S, lot 27, WB 1:1,000, Hu), ATF2 (Abcam, ab32160, clone E243, lot GR3430555-1, WB 1:1,000, Hu), ATF2 (Proteintech, 14834-1-AP, WB 1:1,000, Ms, KD/KO validated by manufacturer), ATP7A (Santa Cruz Biotechnology, sc-376467, clone D-9, lot J2821, WB 1:200, Hu), ATP7A (Novus Biologicals, NBP2-59376, clone S60-4, WB 1:1,000, Ms), ATP7B (Santa Cruz Biotechnology, sc-373964, clone A-11, lot I2719, WB 1:200, Hu, Ms), catalase (Cell Signaling, 12980T, clone D4P7B, lot 3, WB 1:1,000, Hu), CCL2 (also known as Mcp1) (Proteintech, 66272-1-Ig, clone 1B9F7, WB 1:1,000, Hu), CD11b-Pacific Blue (BioLegend, 101224, clone M1/70, lot B350151, B323654 and B323653, FCy 1:800, Ms), CD14-Krome Orange (Beckman Coulter, B01175, clone RMO/52, lot 200040, FCy 1:100, Hu), CD16-Pacific Blue (Beckman Coulter, B36292, clone 3G8, lot 200029, FCy 1:100, Hu), CD3 (BioLegend, 317326, clone OKT3, lot B372352, T cell activation, 2.5 µg ml⁻¹, Hu), CD25-BV711 (BioLegend, 302636, clone BC96, lot B281779, FCy 1:100, Hu), CD28 (BioLegend, 302934, clone CD28.2, lot 374639, T cell activation, 2.5 µg ml⁻¹, Hu), CD40-APC (BioLegend, 124612, clone 3/23, lot B309981, FCy 1:200, Ms), CD40-BV510 (BioLegend, 334329, clone 5C3, lot 312131, FCy 1:100, Hu), CD44 (Abcam, ab189524, lot GR320797-13, GR3314218-16, clone EPR18668, WB 1:30,000, Hu, Ms, KO/KD validated by us and manufacturer), CD44 (Thermo Fisher Scientific, 701406, clone 19H8L4, lot 1976318, FM 1:400, Hu), CD44-AF647 (Novus Biologicals, NB500-481AF647, clone MEM-263, lot 118753, FCy 1:100, Hu), CD44-AF647 (BioLegend, 103018, clone IM7, lot B317762, FCy 1:200, Ms), CD45-BV510 (BioLegend, 103138, clone 30-F11, lot B362964, B322199, B333193, FCy 1:200, Ms), CD64-FITC (BioLegend, 399505, clone S18012C, lot 308498, FCy 1:100, Hu), CD66b-PE/Cy7 (BioLegend, 305115, clone G10F5, lot 283925, FCy 1:100, Hu), CD69-PerCP (BioLegend, 310928, clone FN50, lot B290414, FCy 1:100, Hu), CD80-AF700 (BD Biosciences, 561133, clone 307.4, lot 1060235, FCy 1:100, Hu), CD83-PE (BioLegend, 305307, clone HB15e, lot B303073, FCy 1:100, Hu), CD86-PE (BioLegend, 105007, clone GL-1, lot B318893, FCy 1:200, Ms), CD86-PE/Cy7 (BD Biosciences, 561128, clone 2331 (FUN-1), lot 1309531, FCy 1:33, Hu), CD109 (Santa Cruz Biotechnology, sc-271085, clone C-9, lot E1018, WB 1:200, Hu), CD109 (Biotechne, AF7717-SP, WB 1:1,000, Ms), CD163-PE (BD Biosciences, 556018, clone GHI/61, lot 9143793, FCy 1:100, Hu), CD170 (Siglec-F)-PEeFluor 610 (eBioscience, 61-1702-80, clone 1RNM44N, lot 2472220 and 2152352, FCy 1:200, Ms), cGAS (Cell Signaling, 15102, clone D1D3G, lot 4, WB 1:1,000, Hu), cGAS (Cell Signaling, 31659, clone D3O8O, lot 3, WB 1:1,000, Ms), CLOCK (Proteintech, 18094-1-AP, WB 1:1,000, Hu, KD/KO validated by manufacturer), CLOCK (Abcam, ab3517, WB 1:1,000, Ms), CTR1 (Abcam, ab129067, clone EPR7936, lot GR3414582-4 and GR81444-2, WB 1:1,000, Hu, KD validated by us), CTR2 (Novus Biologicals, NBP1-05199SS, WB 1:1,000, Hu), CTR2 (Novus Biologicals, NBP1-85512, lot R05901, FM 1:400, Hu), CTR2 (Biorybt, orb182668, lot BR2373, WB 1:1,000, Hu) COX IV (Abcam, ab16056, lot GR320655-1, FM 1:400, Hu), CREM (Proteintech, 12131-1-AP, WB 1:1,000, Hu), cytochrome *c* (Cell Signaling, 12963S, clone 6H2.B4, lot 1 and 2, FM 1:400, NS 1:400, Hu), DMT1 (Abcam, ab55735, clone 4C6, lot GR3243346-1, WB 1:1,000, Hu, KD validated by us), *Drosophila* spike-in antibody (Active Motif, 61686, lot 23521010, ChIP 50 ng per condition), E-cadherin (Cell Signaling, 3195, clone 24E10, lot 15, WB 1:1,000, Hu), E-cadherin (BD Biosciences, 610181, clone 36, lot 7187865, WB 1:1,000, Ms), F4/80-BV605 (BioLegend, 123133, clone BM8, lot B362524, B309659, B331465 and B339746, FCy 1:100, Ms), F4/80-PE (TONBO, TNB50-4801-U100, clone BM8.1, lot C4801060619503, FCy 1:100, Ms), Fibronectin (Sigma-Aldrich, F0791, clone IST-3, lot 026M4781V, WB 1:1,000, Hu, Ms), FTO (Proteintech, 27226-1-AP, WB 1:1,000, Hu, Ms, KD/KO validated by manufacturer), H3 (Cell Signaling, 9715S, lot 23, FM, WB 1:1,000, Hu, Ms), H3K4me3 (Diagenode, C15410003-50, lot A8034D, FM 1:400, Hu, dot blot validation by manufacturer), H3K9ac (Cell Signaling, 9649S, clone C5B11, lot 13, FM, WB, ChIP 6 µl per 10⁶ cells, Hu, Ms, validated with SimpleChIP Enzymatic Chromatin IP by manufacturer), H3K9me2 (Cell Signaling, 4658S, clone D84B4, lot 10, FM 1:400, WB 1:1,000, ChIP 6 µl per 10⁶ cells, Hu, Ms, validated with SimpleChIP Enzymatic Chromatin IP by manufacturer), H3K9me3 (Cell Signaling, 13969S, clone D4W1U, lot 3, FM 1:400, Hu, validated with SimpleChIP Enzymatic Chromatin IP by manufacturer), H3K14ac (Cell Signaling, 7627S, clone D4B90, lot 6, FM, WB 1:1,000, ChIP 6 µl per 10⁶ cells, Hu, Ms, validated with SimpleChIP Enzymatic Chromatin IP by manufacturer), H3K27ac (Cell Signaling, 8173S, clone D5E4, lot 8, FM, WB 1:1,000, ChIP 6 µl per 10⁶ cells, Hu, Ms, validated with SimpleChIP Enzymatic Chromatin IP by manufacturer), H3K27me3 (Cell Signaling, 9733S, clone C36B11, lot 19, FM 1:400, WB 1:1,000, ChIP 6 µl per 10⁶ cells, Hu, Ms, validated with SimpleChIP Enzymatic Chromatin IP by manufacturer), H3K36me2 (Abcam, ab9049, lot GR3258133-1, FM 1:400, Hu), hyaluronan synthase 1 (HAS1, Novus Biologicals, NBP1-51635, clone 3E10, lot 141031, WB 1:1,000, Hu), HAS1 (Sigma-Aldrich, SAB4300848, lot 492637613, WB 1:1,000, Ms), hyaluronan synthase 2 (HAS2, Abcam, ab140671, clone

4E7, lot GR3212928-2, WB 1:1,000, Hu), HAS2 (Santa Cruz Biotechnology, sc-514737, clone A-7, WB 1:200, Ms), hyaluronan synthase 3 (HAS3, Abcam, ab154104, lot GR113715-12, WB 1:1,000, Hu), HAS3 (Proteintech, 15609-1-AP, WB 1:1,000, Ms, KO/KD validated by us and manufacturer), HAT1 (Proteintech, 11432-1-AP, WB 1:1,000, Hu, Ms, KO/KD validated by us and manufacturer), 5-hydroxymethylcytosine (5hmC, Active Motif, 39069, lot 23720003, FM 1:400, Hu, dot blot validated by manufacturer), I-A/I-E-AF700 (BioLegend, 107622, clone M5/114.15.2, lot B313251, FCγ 1:400, Ms), IRAK4 (Cell Signaling, 4363T, lot 5, WB 1:1,000, Hu, Ms), JAK2 (Cell Signaling, 3230T, clone D2E12, WB 1:1,000, lot 13, Hu, Ms), JMJD6 (Abcam, ab64575, lot GR3441511-1, WB 1:1,000, Hu, Ms), KAT2B/PCAF (Cell Signaling, 3378T, clone C14G9, lot 2, WB 1:1,000, Hu, Ms, validated with SimpleChIP Enzymatic Chromatin IP by manufacturer), KAT3A (also known as CREBBP) (Abcam, ab2832, lot GR3360262-6, WB 1:1,000, Hu, Ms), KAT5 (also known as Tip60) (Santa Cruz Biotechnology, sc-166323, clone C-7, lot 166323, WB 1:200, Hu, Ms), KAT6A (also known as MOZ) (Santa Cruz Biotechnology, sc-293283, clone 4D8, lot F0420, WB 1:200, Hu), KAT6A (Invitrogen, PA5-103467, lot XH3653004, WB 1:1,000, Ms), KAT8 (also knon as MOF) (Proteintech, 13842-1-AP, KD/KO validated by manufacturer, WB 1:1,000, Hu, Ms), KDM2A (Abcam, ab191387, clone EPR18602, lot GR3330146-4, WB 1:1,000, Hu, Ms, KO validated by manufacturer), KDM5A (also known as Jarid1a) (Cell Signaling, 3876S, clone D28B10, lot 5, WB 1:1,000, Hu, Ms), KDM5B (also known as Jarid1b) (Cell Signaling, 3273T, lot 3, WB 1:1,000, Hu), KDM5B (Abcam, ab181089, clone EPR12794, WB 1:1,000, Ms, KO validated by manufacturer), KDM5C (also known as Jarid1c) (Cell Signaling, 5361, clone D29B9, lot 1, 5361, WB 1:1,000, Hu, Ms), KDM6A (also known as UTX) (Cell Signaling, 33510S, clone D3Q1l, lot 4, WB 1:1,000, Hu, Ms), KDM6B (Abcam, ab169197, WB 1:1,000, Hu), KDM6B/JMJD3 (Cell Signaling, 3457, WB 1:1,000, Ms), KDM7A (Invitrogen, PA5-96987, lot UI2838718, WB 1:1,000, Hu, Ms), Ly6C–PerCP/Cy5.5 (BioLegend, 128012, clone HK1.4, lot B363119 and B310463, FCγ 1:200, Ms), Ly6G–PE-Cy7 (BioLegend, 127618, clone 1A8, lot B288785 and B351626, FCγ 1:200, Ms), Ly6G–AF647 (BioLegend, 127610, clone 1A8, lot B2559839, FCγ 1:200, Ms), lysosome-associated membrane protein 2 (LAMP2) (Abcam, ab25631, clone H4B4, FM 1:400, Hu), NOS2–APC (eBioscience, 17-5920-82, clone CXNFT, lot 2154045, FCγ 1:100, Ms), SLC25A3 (Santa Cruz Biotechnology, sc-376742, clone F-1, lot H2313, WB 1:200, FM 1:100, Hu, KO validated by us), SLC25A3 (Abcam, ab89117, lot 1015892-1, FM 1:400, Hu), SLC25A37 (MyBiosource, MBS9210193, clone ID: RB24153, lot SA100524AR, WB 1:1,000, FM 1:400, Hu, KD validated by us), Slug (Cell Signaling, 9585S, clone C19G7, lot 6, WB 1:1,000, Hu, Ms), STAT1 (Cell Signaling, 14994T, clone D1K9Y, lot 8, WB 1:1,000, Hu, Ms, validated with SimpleChIP Enzymatic Chromatin IP by manufacturer), STAT2 (Abcam, ab32367, clone Y141, lot GR3294792-5, WB 1:1,000, Hu, Ms, KO validated by manufacturer), SOD2 (Abcam, ab13534, lot GR3345921-12 and GR33618-52, FM 1:400, WB 1:1,000, Hu, Ms, KD validated by us), Tet methylcytosine dioxygenase 2 (TET2, Abcam, ab94580, lot GR3243631-2, WB 1:1,000, Hu, Ms), Tet methylcytosine dioxygenase 3 (TET3, Abcam, ab139311, lot GR3314447-1, WB 1:1,000, Hu, Ms), TRAF2 (Cell Signaling, 4724T, clone C192, lot 2, WB 1:1,000, Hu, Ms), TFR1 (Invitrogen, 13-6800, clone H68.4, lot VJ313549, WB 1:1,000, Hu, KD validated by us), TFR1–APC–AF750 (Beckman Coulter, A89313, clone YDJ1.2.2, lot 200060, FCγ 1:100, Hu), γ-tubulin (Sigma-Aldrich, T5326, clone GTU-88, Source 0000128065, WB 1:1,000, Hu, Ms, enhanced validation by manufacturer), Twist (Santa Cruz Biotechnology, sc-81417, clone Twist2C1a, lot J2213, WB 1:200, Ms), vimentin (Cell Signaling, 5741S, clone D21H3, lot 8, WB 1:1,000, Hu, Ms, KO validated by previous users according to manufacturer's website). Secondary antibodies: Alexa Fluor 488 anti-rabbit (Invitrogen, A-11070, lot 2161039, FM 1:1,000, Hu), Alexa Fluor 594 anti-mouse (Invitrogen, A-11032, lot 1826426, FM 1:1,000, Hu), Alexa Fluor 594 anti-rabbit (Invitrogen, A11072, lot 1985650, FM 1:1,000, Hu), Alexa Fluor 647 anti-mouse (Invitrogen, A21237, lot 1743738, FM 1:1,000, Hu), Alexa Fluor 647 anti-rabbit (Invitrogen, A21246, lot 2418503, FM 1:1,000, Hu), donkey anti-rabbit IgG-h+l HRP-conjugated (Bethyl Laboratories, A120-108P, lot 12 and 13), goat anti-mouse IgG h+l HRP-conjugated (Bethyl Laboratories, A90-116P, lot 41 and 44), 10 nM gold-nanoparticle-loaded anti-mouse (Abcam, ab27241, lot GR274015-2, NS 1:200, Hu).

## Primary cells

Peripheral blood samples were collected from 128 healthy donors at EFS. The use of EFS blood samples from anonymous donors was approved by the committee of INSERM (Institut National de la Santé et de la Recherche Médicale). Written consent was obtained from all the donors. Pan monocytes were isolated by negative magnetic sorting using microbeads according to the manufacturer's instructions (Miltenyi Biotec, 130-096-537) and cultured immediately in the presence of cytokines to trigger in vitro differentiation as described in 'Cell culture'. Cells were used fresh without prior freezing. Typically, cells were collected by incubation with 1× PBS with 10 mM EDTA at 37 °C and then scraped, unless stated otherwise. Primary non-small cell lung circulating cancer cells were obtained from Celprogen (36107-34CTC) and cultured as described in 'Cell culture'. Primary macrophages from in vivo mouse models (LPS, CLP and SARS-CoV-2) were isolated and processed as described in 'LPS-induced sepsis model', 'CLP-induced sepsis model' and 'SARS-CoV-2-induced acute inflammation model'.

## Cell culture

This study was carried out using primary monocytes obtained from peripheral blood samples of 128 distinct human donors from EFS. Pan monocytes were isolated by negative magnetic sorting using microbeads according to the manufacturer's instructions (Miltenyi Biotec, 130-096-537), and cultured in RPMI 1640 supplemented with GlutaMAX (Gibco, 61870010), 10% fetal bovine serum (FBS, Eurobio Scientific, CVFSVF00-01). To generate MDMs, pan monocytes were treated with granulocyte-macrophage colony-stimulating factor (GM-CSF, Miltenyi Biotec, 130-093-866, 100 ng ml$^{-1}$) for 5 days. Subsequently, LPS (InvivoGen, tlrl-3pelps, 100 ng ml$^{-1}$) and interferon-γ (IFNγ, Miltenyi Biotec, 130-096-484, 20 ng ml$^{-1}$) were added to the medium for 24 h to generate aMDMs. CD4$^+$ T cells were isolated from peripheral blood samples by negative magnetic sorting using microbeads according to the manufacturer's instructions (Miltenyi Biotec, 130-096-533) and cultured in RPMI 1640 supplemented with glutamine and 10% fetal bovine serum. Subsequently, CD3/CD28 antibodies were added to the medium for 48 h to generate activated CD4$^+$ T cells. CD8$^+$ T cells were isolated from peripheral blood samples by negative magnetic sorting using microbeads according to the manufacturer's instructions (Miltenyi Biotec, 130-096-495) and cultured in RPMI 1640 supplemented with glutamine and 10% fetal bovine serum. Subsequently, CD3/CD28 antibodies were added to the medium for 48 h to generate activated CD4$^+$ T cells. To generate anti-inflammatory macrophages, pan monocytes were treated with macrophage colony-stimulating factor (M-CSF, Miltenyi Biotec, 130-096-492, 100 ng ml$^{-1}$) for 5 days and subsequently, IL-4 (Miltenyi Biotec, 130-093-921, 20 ng ml$^{-1}$) was added to the medium for 24 h. Neutrophils were isolated from peripheral blood samples. Red cells in whole blood samples were lysed (eBioscience RBC lysis buffer, 00-4300-54). The remaining cells were cultured in RPMI 1640 supplemented with glutamine, and 2% human serum. Subsequently, LPS (2 μg ml$^{-1}$) was added for 1 h to generate activated neutrophils. The neutrophil population was determined by flow cytometry using FSC, SSC and CD15 surface staining. To generate dendritic cells, pan monocytes were treated with GM-CSF (100 ng ml$^{-1}$) and IL-4 (10 ng ml$^{-1}$) for 5 days. Subsequently, LPS (100 ng ml$^{-1}$) was added to the medium to generate activated dendritic cells (aDC). FC1242 mouse pancreatic cancer cells were a generous gift from the Tuveson laboratory (Cold Spring Harbor Laboratory) and were cultured in Dulbecco's Modified Eagle Medium GlutaMAX (DMEM, Gibco, 61965059) supplemented with 10% FBS (Gibco, 10270-106) and penicillin–streptomycin mixture (BioWhittaker/Lonza, DE17-602E). Cells were treated with TGF-β

(Miltenyi Biotec, 130-095-066, 10 ng ml$^{-1}$) for 6 days. Primary non-small cell lung circulating cancer cells (Celprogen, 36107-34CTC, lot 219411, sex: female) were grown using stem cell complete media (Celprogen, M36102-29PS) until the third passage. These cells were grown in stem cell ECM T75-flasks (Celprogen, E36102-29-T75) and ECM 6-well plates (Celprogen, E36102-29-6Well) and treated with TGF-β for 3 days.

MDMs were activated with LPS and IFNγ and co-treated with ATTM (Sigma-Aldrich, 323446, 10 μM) or EDTA (Euromedex, EU0007, 500 μM), hyaluronate (Carbosynth, FH45321, 600-1000 kDa, 1 mg ml$^{-1}$), methylated hyaluronate (in-house, 1 mg ml$^{-1}$), LCC-12 (in-house, 10 μM), LCC-4,4 (in-house, 10 μM), metformin (1,1-dimethylbiguanide hydrochloride, Alfa Aesar, J63361, 10 mM), D-penicillamine (Sigma-Aldrich, P4875, 250 μM), trientine hydrochloride (Trien, Sigma-Aldrich, PHR1495-500MG, 200 μM) or anti-human CD44 therapeutic antibody (RG7356, Creative Biolabs, TAB-128CL, 10 μg ml$^{-1}$) for 24 h. aMDMs were treated with CCCP (Sigma-Aldrich, C2759, 10 μM), $^{15}$N,$^{13}$C-LCC-12 (in-house, 10 μM), LCC-12,4 (in-house, 100 nM) or trientine alkyne (in-house, 10 μM) for 3 h, by adding the reagents directly to the media. Dendritic cells were co-treated with LPS and LCC-12 (10 μM) for 24 h. CD4$^{+}$ and CD8$^{+}$ T cells were co-treated with CD3/CD28 and LCC-12 (50 μM) for 48 h. Anti-inflammatory macrophages were co-treated with IL-4 and LCC-12 (10 μM) for 24 h. Neutrophils were co-treated with LPS and LCC-12 (10 μM) for 1 h. Primary non-small cell lung circulating cancer cells and FC1242 cells were co-treated with TGF-β and LCC-12 (1 μM) for 3 days and 6 days, respectively.

## Flow cytometry

For human immune cells: cells were washed with ice-cold 1× PBS, incubated with Fc block (Human TruStain FcX, BioLegend, 422302, 1:20) for 15 min, subsequently incubated with antibodies for 20 min at 4 °C in 1× PBS/ 0.5% bovine serum albumin (BSA) and then washed before analysis using a flow cytometer (BD LSR Fortessa X-20). Cells were analysed with the corresponding antibody panels. Primary non-small cell lung circulating cancer cells and FC1245 cells were collected by trypsinization using Trypsin/EDTA (GIBCO, TRYPGIB01), washed with 1× PBS and antibody incubation was performed for 20 min at 4 °C in 1× PBS/10% FBS. Cells were washed and analysed using a flow cytometer (BD Accurie C6). For flow cytometry on cells from the LPS murine model, see the LPS-induced sepsis model paragraph. Flow cytometry analysis of mitochondrial H$_2$O$_2$ content: MDMs were activated with LPS and IFNγ, in the presence of MitoPY1 (R&D Systems, 4428/10, 5 μM) for 24 h. Fluorescence was analysed by flow cytometry. Flow cytometry analysis with the mitochondrial Cu(II) probe M$_{Cu}$-2$^{39}$: naMDMs, aMDMs and MDMs that were activated with LPS and IFNγ and co-treated with LCC-12 (10 μM) were incubated with M$_{Cu}$-2 (5 μM) for 1 h. Fluorescence was analysed by flow cytometry. The data were analysed with FlowJo software v. 10.8.2.

## ICP-MS

Glass vials equipped with Teflon septa were cleaned with nitric acid 65% (VWR, Suprapur, 1.00441.0250), washed with ultrapure water (Sigma-Aldrich, 1012620500) and dried. Cells were collected and washed twice with 1× PBS. Cells were then counted using an automated cell counter (Entek) and transferred in 200 μl 1× PBS or ultrapure water to the cleaned glass vials. The same volume of 1× PBS or ultrapure water was transferred into separate vials for the background subtraction, at least in duplicate per experiment. Mitochondria, nuclei and endoplasmic reticula were extracted as described in 'Isolation of mitochondria' from a pre-counted population of cells. Samples were lyophilized using a freeze dryer (CHRIST, 2–4 LDplus). Samples were subsequently mixed with nitric acid 65% and heated at 80 °C overnight in the same glass vials closed with a lid carrying a Teflon septum. Samples were then cooled to room temperature and diluted with ultrapure water to a final concentration of 0.475 N nitric acid and transferred to metal-free centrifuge vials (VWR, 89049-172) for subsequent mass spectrometry analyses. Amounts of metals were measured using an Agilent 7900 ICP-QMS in low-resolution mode, taking natural isotope distribution into account. Sample introduction was achieved with a micro-nebulizer (MicroMist, 0.2 ml min$^{-1}$) through a Scott spray chamber. Isotopes were measured using a collision-reaction interface with helium gas (5 ml min$^{-1}$) to remove polyatomic interferences. Scandium and indium internal standards were injected after inline mixing with the samples to control the absence of signal drift and matrix effects. A mix of certified standards was measured at concentrations spanning those of the samples to convert count measurements to concentrations in the solution. Values were normalized against cell number.

## Western blotting

MDMs were treated as indicated and then washed with 1× PBS. For MDMs, proteins were solubilized in 2× Laemmli buffer containing benzonase (VWR, 70664-3, 1:100). Extracts were incubated at 37 °C for 1 h and heated at 94 °C for 10 min, and quantified using a NanoDrop 2000 spectrophotometer (Thermo Fisher Scientific). SPMs and AMs from the LPS, CLP and SARS-CoV-2 mouse models were isolated by flow cytometry as described in the relevant flow cytometry section. Due to low cell number count, SPMs were collected in 1× PBS, which was subsequently freeze dried in Eppendorf tubes. Dried material was solubilized in 2× Laemmli buffer containing benzonase. AMs were pelleted and the cell pellets were solubilized in 2× Laemmli buffer containing benzonase. Extracts from SPMs and AMs were then incubated at 37 °C for 1 h and heated at 94 °C for 10 min. Proteins extracts from SPMs and AMs were quantified using a Qubit (Invitrogen) and a Qubit protein quantification assay (Invitrogen, Q33212). For the LPS model, SPMs were pooled for 8 sham mice, for 4 LPS-treated mice and for 6 LPS and LCC-12-treated mice. For the CLP model, SPMs were pooled from 8 sham mice and 7 mice subjected to CLP. For the SARS-CoV-2 model, AMs were pooled for 10 sham mice and for 10 SARS-CoV-2-infected mice. Protein lysates were resolved by SDS–PAGE (Invitrogen sure-lock system and NuPAGE 4–12% Bis-Tris precast gels). In a typical experiment from MDMs or cancer cells 10–20 μg of total protein extract was loaded per lane in 2× Laemmli buffer containing bromophenol blue. For in vivo isolated SPM and AMs where protein amounts were limited, typically 1 μg of total protein extract was loaded (more protein lysate was loaded in a new experiment if the antibody could not recognize a specific band at these protein amounts). On each gel a size marker was run (3 μl PageRuler or PageRuler plus, Thermo Scientific, 26616 or 26620 and 17 μl 2× Laemmli buffer) in parallel. Proteins were then transferred onto nitrocellulose membranes (Amersham Protran 0.45 μm) using a Trans-Blot SD semi-dry electrophoretic transfer cell (Bio-rad) using 1× NuPage transfer buffer (Invitrogen, NP00061) with 10% methanol. Membranes were blocked with 5% non-fat skimmed milk powder (Régilait) in 0.1% Tween-20/1× PBS for 20 min. Membranes were cut at the appropriate marker size to allow for the probing of several antibodies on the same membrane. Blots were then probed with the relevant primary antibodies in 5% BSA, 0.1% Tween-20/1× PBS or in 5% non-fat skimmed milk powder in 0.1% Tween-20/1× PBS at 4 °C overnight with gentle motion in a hand-sealed transparent plastic bag. Membranes were washed with 0.1% Tween-20/1× PBS three times and incubated with horseradish peroxidase conjugated secondary antibodies (Jackson Laboratories) in 5% non-fat skimmed milk powder, 0.1% Tween-20/1× PBS for 1 h at room temperature and washed three times with 0.1% Tween-20/1× PBS. Antigens were detected using the SuperSignal West Pico PLUS (Thermo Scientific, 34580) and SuperSignal West Femto (Thermo Scientific, 34096) chemiluminescent detection kits. For blotting proteins on the same membranes, stripping buffer (0.1 M TRIS pH 6.8, 2% SDS w/v, 0.1 M β-mercaptoethanol) was used for 30 min and membranes were washed with 0.1% Tween-20/1× PBS and reblotted subsequently as described above. Signals were recorded using a Fusion Solo S Imaging System (Vilber). For histone marks, H3 was run as a sample processing control on a separate gel in parallel and is displayed in the respective panels.

γ-tubulin served as loading control on the same gels and is not displayed in the respective panels. Band quantifications were performed with FIJI 2.0.0-rc-69/1.52n using pixel intensity normalized against the signal of γ-tubulin. All full scans of blots are displayed in the Supplementary Information.

## Fluorescence microscopy

Isolated monocytes were plated on coverslips, differentiated and activated as described in 'Cell culture'. For fluorescent detection of hyaluronate and Lys-Cu, live cells were treated with hyaluronate–FITC (800 kDa, Carbosynth, YH45321, 0.1 mg ml$^{-1}$) and Lys-Cu (in-house, 20 µM, 1 h)[26] for 1 h in the presence or absence of hyaluronidase (Sigma-Aldrich, H3884, 0.1 mg ml$^{-1}$). Hyaluronate–FITC and hyaluronidase were solubilized together in medium for 2 h at 37 °C before adding to the cells. Cells were then washed three times with 1× PBS, fixed with 2% paraformaldehyde in 1× PBS for 12 min and then washed 3 times with 1× PBS. For antibody staining, cells were then permeabilized with 0.1% Triton X-100 in 1× PBS for 5 min and washed 3 times with 1× PBS. Subsequently, cells were blocked in 2% BSA, 0.2% Tween-20/1× PBS (blocking buffer) for 20 min at room temperature. Cells were incubated with the relevant antibody in blocking buffer for 1 h at room temperature, washed 3 times with 1× PBS and were incubated with secondary antibodies for 1 h. Finally, coverslips were washed 3 times with 1× PBS and mounted using VECTASHIELD containing DAPI (Vector Laboratories, H-1200-10). Fluorescence images were acquired using a Deltavision real-time microscope (Applied Precision). 40×/1.4NA, 60×/1.4NA and 100×/1.4NA objectives were used for acquisitions and all images were acquired as z-stacks. Images were deconvoluted with SoftWorx (Ratio conservative–15 iterations, Applied Precision) and processed with FIJI 2.0.0-rc-69/1.52n. Fluorescence intensity is displayed as arbitrary units (AU) and is not comparable between different panels. Colocalization quantification was calculated using FIJI 2.0.0-rc-69/1.52n. Histone quantification was performed using FIJI 2.0.0-rc-69/1.52n by delineating the nuclei using DAPI fluorescence, and calculating the mean fluorescence intensity normalized by area.

## Small molecule labelling using click chemistry

aMDMs on coverslips were treated with LCC-12,4 (in-house, 100 nM, 3 h) in absence or presence of CCCP or LCC-12 competitor (10 µM, 3 h), fixed and permeabilized as indicated in the fluorescence microscopy paragraph. Mitotracker (Invitrogen, M22426) was added to live cells for 45 min before fixation. For in-cell-labelling of trientine, live cells were incubated with trientine alkyne (in-house, 10 µM, 3 h). The click reaction cocktail was prepared using the Click-iT EdU Imaging kit (Invitrogen, C10337) according to the manufacturer's protocol. In a typical experiment we mixed 50 µl of 10× Click-iT reaction buffer with 20 µl of CuSO$_4$ solution, 1 µl Alexa Fluor–azide, 50 µl reaction buffer additive (sodium ascorbate) and 379 µl ultrapure water to reach a final volume of 500 µl. For variations as indicated in the figures, reactions were performed with or without CuSO$_4$ and ascorbate. Coverslips were incubated with the click reaction cocktail in the dark at room temperature for 30 min, then washed three times with 1× PBS. Immunofluorescence was then performed as described in 'Fluorescence microscopy'.

## NanoSIMS imaging

MDMs were grown on coverslips and activated to obtain aMDMs as described in 'Cell culture'. Cells were treated with 10 µM $^{15}$N,$^{13}$C-LCC-12 for 3 h. Subsequently, cells were washed twice with 1× PBS, once with 0.1 M cacodylate buffer (LFG Distribution, 11653) and then fixed with 2% paraformaldehyde in 0.1 M cacodylate buffer for 20 min. Then, cells were washed three times with 0.1 M cacodylate buffer for 5 min and permeabilized with 0.1% Triton X-100 in 0.1 M cacodylate buffer for 5 min. Subsequently, cells were washed three times with 0.1 M cacodylate buffer and blocking buffer (2% BSA, 0.1% Tween in 0.1 M cacodylate buffer) was added for 20 min. Primary antibody (1:400) was added for 1 h in blocking buffer. Then, cells were washed three times with 0.1 M cacodylate buffer and the 10 nM gold-nanoparticle-loaded secondary antibody (1:50) was added in blocking buffer for 1 h. Cells were washed three times with 0.1 M cacodylate buffer and treated with 1% OsO$_4$ (Electron Microscopy Sciences, 19152) in 0.1 M cacodylate buffer for 1 h. Coverslips with samples were washed three times for 10 min with Milli-Q water. Subsequently, cells were dehydrated sequentially with ethanol solutions for 10 min each: 50%, 70%, 2× 90%, 3× 100% (dried over molecular sieves, Sigma-Aldrich, 69833). Samples were then coated with a 1:1 mixture of resin (Electron Microscopy Sciences, dodecenylsuccinic anhydride, 13710, methyl-5-norbornene-2, 3-dicarboxylic anhydride, 19000, DMP-30, 13600 and LADD research industries: LX112 resin, 21310) and dry ethanol for 1 h. Then, samples were embedded in pure resin for 1 h. Embedding capsules (Electron Microscopy Sciences, 69910-10) were filled with resin, inverted onto the cover slides and placed in an oven at 56 °C for 24 h. Sections 0.2 µm in thickness were prepared using a Leica Ultracut UCT microtome. Sample sections were deposited onto a clean silicon chip (Institute for Electronic Fundamentals/CNRS and University Paris Sud) and dried upon exposure to air before being introduced into the NanoSIMS-50 ion microprobe (Cameca). A Cs$^+$ primary ion was employed to generate negative secondary ion from the sample surface. The probe steps over the image field and the signal of selected secondary ion species were recorded pixel-by-pixel to create 2D images. Image of $^{12}$C$^{14}$N$^-$ was recorded to provide the anatomic structure of the cells, while the one of $^{31}$P$^-$ highlights the location of cell nucleus. The cellular distribution of $^{15}$N label was imaged by measuring the excess in $^{12}$C$^{15}$N$^-$ to $^{12}$C$^{14}$N$^-$ ratio with respect to the natural abundance level (0.0037), and the one for antibody with gold staining targeting mitochondria was performed by detecting directly $^{197}$Au$^-$ ions. When detecting $^{12}$C$^{15}$N$^-$ ion, appropriate mass resolution power was required to discriminate abundant $^{13}$C$^{14}$N$^-$ isobaric ions (with an M/ΔM of 4,272). For each image recording process, multiframe acquisition mode was applied and hundreds of image planes were recorded. The overall acquisition time corresponding to the $^{15}$N image was 12 h and 6 h 30 min for the $^{197}$Au image. During image processing with FIJI (2.0.0-rc-69/1.52n), the successive image planes were properly aligned using TomoJ plugin[51], so as to correct the slight primary beam shift during long hours of acquisition. A summed image was then obtained with improved statistics. Further, for the $^{12}$C$^{15}$N$^-$ to $^{12}$C$^{14}$N$^-$ ratio map, an HSI (Hue-Saturation-Intensity) colour image was generated using OpenMIMS for display with increased significance[52]. The hue corresponds to the absolute $^{15}$N/$^{14}$N ratio value, and the intensity at a given hue is an index of the statistical reliability.

## RNA interference

Human primary monocytes were transfected with Human Monocyte Nucleofector kit (Lonza, VPA-1007) according to the manufacturer's instructions. 5× 10$^6$ monocytes were resuspended in 100 µl nucleofector solution with 200 pmol of ON-TARGETplus SMARTpool siRNA or negative control siRNA (Qiagen, 1027310) before nucleofection with Nucleofector II (Lonza). Cells were then immediately removed and incubated overnight with 5 ml of prewarmed complete RPMI medium (Gibco). The following day, GM-CSF was added to the medium. The sequences of the SMARTpools used are detailed in the Supplementary Information.

## Genome editing

CRISPR knockout was performed using the following strategy. Human primary monocytes were transfected with Human Monocyte Nucleofector kit (Lonza, VPA-1007). Five million monocytes were resuspended in 100 µl nucleofector solution with a 100 pmol CAS9 (Dharmacon, CAS12206)/200 pmol CD44 (Dharmacon, SQ-009999-01-0010) single guide RNA (sgRNA) mix. The Cas9–CD44 mix was incubated for 10 min at 37 °C before nucleofection with Nucleofector II (Lonza). Cells were then immediately removed and incubated overnight with 5 ml of

prewarmed complete RPMI medium (Gibco) and the following day, GM-CSF was added to the medium. At day 5, the cells were activated with LPS (100 ng ml$^{-1}$, 24 h) and IFNγ (20 ng ml$^{-1}$, 24 h). At day 6, cells were sorted for CD44$^-$ versus CD44$^+$ populations with BD FACSAria. The sorting strategy and the sequences used for CD44 sgRNA (Edit-R Human Synthetic CD44, set of 3, target sequences) are detailed in the Supplementary Information.

## Bright-field microscopy and digital photographs
Bright-field images were acquired using a CKX41 microscope (Olympus) and cellSens Entry imaging software (Olympus). Digital images were taken with an iPhone 11 Pro (Apple).

## Isolation of mitochondria
Mitochondria were isolated using the Qproteome Mitochondria Isolation Kit (Qiagen, 37612) according to the manufacturer's protocol. Cells were washed and centrifuged at 500$g$ for 10 min and the supernatant was removed. Cells were then washed with a solution of 0.9% NaCl (Sigma-Aldrich, S7653-250G) and resuspended in ice-cold lysis buffer and incubated at 4 °C for 10 min. The lysate was then centrifuged at 1,000$g$ for 10 min at 4 °C and the supernatant carefully removed. Subsequently, the cell pellet was resuspended in disruption buffer. Complete cell disruption was obtained by using a dounce homogenizer (mitochondria for ICP-MS) or a blunt-ended needle and a syringe (mitochondria for metabolomics). The lysate was then centrifuged at 1,000$g$ for 10 min at 4 °C and the supernatant transferred to a clean tube. The supernatant was then centrifuged at 6,000$g$ for 10 min at 4 °C to obtain mitochondrial pellets.

## Isolation of nuclei
Nuclei were isolated using the Nuclei EZ Prep (Sigma-Aldrich, NUC101-1KT) according to the manufacturer's instructions. In brief, cells were treated as indicated and collected upon scraping and counted. Subsequently, cells were washed twice with 1× PBS and lysed with 1 ml ice-cold Nuclei EZ lysis buffer for 5 min on ice. The suspension was centrifuged at 500$g$ for 5 min at 4 °C. Resulting nuclei were washed with Nuclei EZ lysis buffer and centrifuged to generate a pellet of isolated cell nuclei.

## Isolation of endoplasmic reticula
Endoplasmic reticula were isolated using the Endoplasmic Reticulum Enrichment Extraction Kit (Novus Biologicals, NBP2-29482) according to the manufacturer's instructions. In brief, 500 µl of 1× isosmotic homogenization buffer followed by 5 µl of 100× PIC were added to a pellet of 10$^6$ cells. The resulting suspension was centrifuged at 1,000$g$ for 10 min at 4 °C. The supernatant was transferred to a clean centrifuge tube and centrifuged at 12,000$g$ for 15 min at 4 °C. The floating lipid layer was discarded. The supernatant was centrifuged in a clean centrifuge tube using an ultracentrifuge at 90,000$g$ for 1 h. The resulting pellet contained the total endoplasmic reticulum fraction (rough and smooth).

## NMR spectroscopy
$^1$H NMR, $^{13}$C NMR and $^{15}$N NMR spectra were recorded on a 500 MHz Bruker spectrometer at 298 K or 310 K, and chemical shifts δ are expressed in ppm using the residual solvent signals as internal standard. To measure the interaction of hyaluronate tetrasaccharide with copper, portions of 0.25 mol equivalent of a solution of CuCl$_2$ in D$_2$O (8.6 mg in 599 µl D$_2$O) were added to a 2 mM solution of hyaluronate tetrasaccharide (TCI Chemicals, H1284) in D$_2$O (1.0 mg hyaluronate tetrasaccharide in 600 µl D$_2$O) up to 1 mol equivalent into an NMR tube. Then, a drop of trifluoroacetic acid (TFA, Alfa Aesar, A12198) was added. In a separate NMR tube, a drop of TFA was added to a copper-free 2 mM solution of hyaluronate tetrasaccharide in D$_2$O. To measure oxidation of NADH into NAD$^+$, $^1$H NMR were recorded and tubes were prepared as follows: NADH (Sigma-Aldrich, N4505-100MG, 200 µM) or NAD$^+$ (Sigma-Aldrich, N0632-1G, 200 µM), imidazole (10 mM), CuSO$_4$ (10 µM) were added as indicated to an NMR tube containing sodium phosphate buffer in D$_2$O (10 mM, pD 8.4, 981 µl) or D$_2$O. $^1$H NMR were recorded at $t_0$. H$_2$O$_2$ (19 µl of 100× diluted solution of 32.3% (w/w%) in H$_2$O) was added to the NMR tubes and $^1$H NMR were recorded after 1 h.

## Chemical synthesis
Products were purified on a preparative HPLC Quaternary Gradient 2545 equipped with a Photodiode Array detector (Waters) fitted with a reverse phase column (XBridge BEH C18 OBD Prep column 5 µm 30×150 mm). NMR Spectra were run in DMSO-$d_6$, methylene chloride-$d_2$ or methanol-$d_4$ at 298 K unless stated otherwise. $^1$H NMR spectra were recorded on Bruker spectrometers at 400 or 500 MHz. Chemical shifts δ are expressed in ppm using residual non-deuterated solvent signals as internal standard. The following abbreviations are used: ex, exchangeable; s, singlet; d, doublet; t, triplet; td, triplet of doublets; m, multiplet. The $^{13}$C NMR spectra were recorded at 100.6 or 125.8 MHz, and chemical shifts δ are expressed in ppm using deuterated solvent signal as internal standard. The purity of final compounds, determined to be >98% by UPLC-MS, and low-resolution mass spectra (LRMS) were recorded on a Waters Acquity H-class equipped with a Photodiode array detector and SQ Detector 2 (UPLC-MS) fitted with a reverse phase column (Acquity UPLC BEH C18 1.7 µm, 2.1x50 mm). HRMS were recorded on a Thermo Scientific Q-Exactive Plus equipped with a Robotic TriVersa NanoMate Advion.

**LCC-12.** Dicyandiamide (Alfa Aesar, A10451, 500 mg, 5.94 mmol), 1,12-diaminododecane (Alfa Aesar, A04258, 500 mg, 2.50 mmol) and CuCl$_2$ (Sigma-Aldrich, 22.201-1, 249 mg, 1.85 mmol) were suspended in water (6 ml) in a sealed tube and stirred for 1 h at room temperature, then heated at 80 °C for 48 h. The resulting pink mixture was filtered and the solid was resuspended in water (10 ml). H$_2$S, generated from dropwise addition of 37% aqueous (aq.) HCl (Supelco, 1.00317.100) on FeS (-100 mesh powder, Alfa Aesar, 17422), was passed into the mixture until it turned black. The black mixture was filtered, and the filtrate was acidified to pH 5 with a 1 M aq. solution of HCl. The solvent was evaporated under reduced pressure. LCC-12 was purified by preparative HPLC (H$_2$O:acetonitrile:formic acid, 95:5:0.1 to 0:100:0.1) to afford LCC-12 di-formic acid salt as a white powder (280 mg, 24%). $^1$H NMR (500 MHz, DMSO-$d_6$) δ: 8.80–8.08 (m, 2H, ex), 8.47 (s, 2H, formate), 7.60–6.78 (m, 12H, ex), 3.14–2.93 (m, 4H), 1.55–1.34 (m, 4H), 1.32–1.18 (m, 16H) ppm. $^{13}$C NMR (125.8 MHz, DMSO-$d_6$) δ: 167.2 (formate), 160.3, 159.2, 41.3, 29.5 (3C), 29.2, 26.8 ppm. HRMS (ESI+) $m/z$: calculated for C$_{16}$H$_{38}$N$_{10}$ [M+2H]$^{2+}$ 185.1635, found 185.1637.

**LCC-4,4.** Bis-(cyanoguanidino)butane (200 mg, 0.90 mmol) and butylamine hydrochloride (TCI Chemical, B0710, 197 mg, 1.80 mmol) were mixed together in a sealed tube and heated at 150 °C without solvent for 4 h. After cooling to room temperature, the mixture was taken up in ethanol and a large excess of ethyl acetate was added. The white precipitate was filtered and purified by preparative HPLC (H$_2$O:acetonitrile:formic acid, 100:0:0.1 to 50:50:0.1) to give the LCC-4,4 di-formic acid salt as a white powder (130 mg, 31%). $^1$H NMR (400 MHz, DMSO-$d_6$) δ: 8.82–7.77 (m, 4H, ex), 8.46 (s, 2H, formate), 7.36–6.80 (m, 8H, ex), 3.16–2.99 (m, 8H), 1.58–1.37 (m, 8H), 1.37–1.20 (m, 4H), 0.86 (t, $J$ = 6.8 Hz, 6H) ppm. $^{13}$C NMR (100.6 MHz, DMSO-$d_6$) δ: 167.3 (formate), 159.1 (2C), 40.9 (2C), 31.6, 26.8, 20.0, 14.1 ppm. HRMS (ESI+) $m/z$: calculated for C$_{16}$H$_{38}$N$_{10}$ [M+2H]$^{2+}$ 185.1635, found 185.1636.

The synthesis of bis-(cyanoguanidino)butane was adapted from a previously published procedure[53]. 1,4-diaminobutane (Acros Organics, 112120250, 554 mg, 5,68 mmol) was dissolved in water and stirred with 37% aq. HCl for 10 min at room temperature. The solvent was evaporated under reduced pressure and the resulting salt was suspended in butanol (5 ml) with sodium dicyanamide (Sigma-Aldrich, 178322, 1.09 g, 11.4 mmol) and stirred at 140 °C overnight. After filtration, the solid was washed with butanol and cold water and recrystallized from water

to give the bis-(cyanoguanidino)butane as a white powder (350 mg, 27%). [1]H NMR (500 MHz, DMSO-$d_6$) δ: 7.62–6.12 (m, 6H, ex), 3.23–2.85 (m, 4H), 1.71–1.15 (m, 4H) ppm. [13]C NMR (125.8 MHz, DMSO-$d_6$) δ: 161.6, 118.8, 40.9, 26.8 ppm.

**LCC-12,4.** Bis-(cyanoguanidino)dodecane (227 mg, 0.60 mmol) and but-3-yne-1-amine hydrochloride (Enamine, EN300-76524, 126 mg, 1.20 mmol) were mixed together in a sealed tube and heated at 150 °C without solvent for 4 h. After cooling to room temperature, the mixture was taken up in ethanol and a large excess of ethyl acetate was added slowly. The white precipitate was filtered and purified by preparative HPLC ($H_2O$:acetonitrile:formic acid, 95:5:0.1 to 40:60:0.1) to give LCC-12,4 di-formic acid salt as a white powder (102 mg, 30%). [1]H NMR (500 MHz, DMSO-$d_6$) δ: 9.05–7.95 (m, 4H, ex), 8.47 (s, 2H, formate), 7.60–6.60 (m, 8H, ex), 3.29–3.16 (m, 4H), 3.11–3.00 (m, 4H), 2.84 (s, 2H), 2.39–2.29 (m, 4H), 1.49–1.38 (m, 4H), 1.33–1.18 (m, 16H) ppm. [13]C NMR (125.8 MHz, DMSO-$d_6$) δ: 167.6 (formate), 159.7, 158.5, 82.6, 72.7, 41.3, 40.3, 29.5 (3C), 29.2, 26.8, 19.4 ppm. HRMS (ESI+) $m/z$: calculated for $C_{24}H_{46}N_{10}$ [M+2H]$^{2+}$ 237.1948, found 237.1947.

The synthesis of bis-(cyanoguanidino)dodecane was adapted from a previously published procedure[53]. 1,12-diaminododecane (500 mg, 2.5 mmol) was dissolved in a mixture of water and methanol and stirred with 37% aq. HCl for 10 min at room temperature. The solvent was evaporated under reduced pressure and the resulting salt was suspended in butanol (2.5 ml) with sodium dicyanamide (444 mg, 5.0 mmol) and stirred at 140 °C overnight. After filtration the solid was washed with butanol and cold water and recrystallized from a mixture of water:ethoxyethanol (2:1) to give the bis-(cyanoguanidino)dodecane as a white powder (365 mg, 44%). [1]H NMR (400 MHz, DMSO-$d_6$) δ: 7.21–6.18 (m, 6H, ex), 3.11–2.94 (m, 4H), 1.49–1.34 (m, 4H), 1.33–1.14 (m, 16H) ppm. [13]C NMR (100.6 MHz, DMSO-$d_6$) δ: 161.6, 118.8, 41.0, 29.4 (3C), 29.2, 26.7 ppm.

**Isotopically labelled LCC-12.** [15]N- and [13]C-labelled dicyandiamide (Eurisotop, CNLM-9324-PK, 50 mg, 0.55 mmol), 1,12-diaminododecane (46.8 mg, 0.23 mmol) and CuCl$_2$ (31.4 mg, 0.23 mmol) were suspended in a sealed tube in water (0.6 ml) and stirred for 1 h at room temperature, then heated at 80 °C for 48 h. The resulting pink mixture was acidified with an aq. solution of HCl (2 M, 1 ml) until complete dissolution of the precipitate. The mixture was concentrated under reduced pressure and isotopically labelled LCC-12 was purified by preparative HPLC ($H_2O$:acetonitrile:formic acid, 95:5:0.1 to 73:27:0.1) to give the isotopically labelled LCC-12 di-formic acid salt as a white powder (35 mg, 32%). [1]H NMR (500 MHz, DMSO-$d_6$) δ: 8.60–7.76 (m, 2H, ex), 8.48 (s, 2H, formate), 7.70–6.30 (m, 12H, ex), 3.12–2.95 (m, 4H), 1.53–1.35 (m, 4H), 1.33–1.18 (m, 16H) ppm. [13]C NMR (125.8 MHz, DMSO-$d_6$) δ: 167.0 (formate), 160.2, 159.2, 41.3, 29.5 (3C), 29.2, 26.8 ppm. [15]N NMR (50.7 MHz, DMSO-$d_6$) δ: 156.2, 84.1 (2N), 81.2 ppm. HRMS (ESI+) $m/z$: calculated for $C_{12}{}^{13}C_4H_{37}N_2{}^{15}N_8$ [M+H]$^+$ 381.3095, found 381.3093.

**Trientine alkyne.** Under argon atmosphere, trientine dihydrochloride (Santa Cruz Biotechnology, sc-216009, 0.050 g, 0.228 mmol) was dissolved in anhydrous methanol at 0 °C followed by the addition of 4-(prop-2-ynyloxy)-benzaldehyde[54] (0.073 mg, 0.456 mmol) and molecular sieves 4 Å. The mixture was stirred for 3 h at room temperature, prior to the addition of NaBH$_3$CN (0.026 g, 0.684 mmol) and stirred overnight at room temperature. Next, the reaction mixture was filtered, the filtrate was evaporated under reduced pressure and purified by preparative HPLC ($H_2O$:acetonitrile:formic acid, 95:5:0.1 to 73:27:0.1) to afford trientine alkyne as a white powder (20.7 mg, 21%). [1]H NMR (500 MHz, Methanol-$d_4$) δ: 7.40–7.33 (m, 4H), 7.06–6.99 (m, 4H), 4.74 (d, $J$ = 2.4 Hz, 4H), 3.97 (s, 4H), 3.00–2.91 (m, 10H), 2.87 (s, 4H). [13]C NMR (125.8 MHz, Methanol-$d_4$) δ: 159.4, 131.7 (2C), 128.7, 116.4 (2C), 79.6, 76.9, 56.7, 52.4, 47.7, 46.8 (2C). (ESI+) $m/z$: calculated for $C_{26}H_{35}N_4O_2$ [M+H]$^+$ 435.2755, found 435.2758.

**Lysosomal copper(II) probe.** The lysosomal copper(II) probe Lys-Cu was synthesized as previously reported and spectral data are in agreement with the literature[26]. [1]H NMR (400 MHz, Methylene chloride-$d_2$) δ: 8.69–8.59 (m, 2H), 8.31 (dd, $J$ = 8.5, 1.2 Hz, 1H), 8.04 (d, $J$ = 1.5 Hz, 1H), 7.82–7.69 (m, 2H), 7.64 (dd, $J$ = 7.8, 1.7 Hz, 1H), 7.24 (d, $J$ = 7.7 Hz, 1H), 6.60 (d, $J$ = 8.8 Hz, 2H), 6.45 (d, $J$ = 2.5 Hz, 2H), 6.38 (dd, $J$ = 8.9, 2.6 Hz, 2H), 4.33 (t, $J$ = 6.8 Hz, 2H), 3.74 (s, 2H), 3.65–3.62 (m, 4H), 3.40–3.34 (m, 9H), 2.69 (t, $J$ = 6.9 Hz, 2H), 2.56 (s, 3H), 1.19 (t, $J$ = 7.1 Hz, 12H).

**Mitochondrial copper(II) probe.** The mitochondrial copper(II) probe M$_{Cu}$-2 was synthesized as previously reported and spectral data are in agreement with the literature[39]. [1]H NMR (400 MHz, DMSO-$d_6$) δ: 9.89 (s, 1H), 8.07–7.67 (m, 16H), 7.58–7.45 (m, 2H), 7.02–6.93 (m, 1H), 6.80 (d, $J$ = 2.4 Hz, 1H), 6.62 (d, $J$ = 2.3 Hz, 1H), 6.58–6.36 (m, 4H), 4.44 (s, 2H), 4.08 (t, $J$ = 6.0 Hz, 2H), 3.75–3.58 (m, 2H), 2.02–1.83 (m, 2H), 1.83–1.60 (m, 2H).

**Methylated hyaluronate.** Methylated hyaluronate (meth-HA) was synthesized using modified published methods[25,55]. In brief, 1 g of sodium hyaluronate (600–1,000 kDa) was dissolved overnight in 200 ml of distilled water at room temperature. Then amberlite cation exchange resin (H$^+$) (10 g) was added to the solution and stirred for one day at room temperature. The resin was subsequently filtered off the solution. The resulting solution was then neutralized using tetrabutylammonium hydroxide (TBAOH) to obtain (tetrabutylammonium) TBA–hyaluronate as follows: TBAOH, diluted fivefold with water was added dropwise to the previously prepared hyaluronic acid solution until the pH reached 8. The solution was then freeze dried and the resulting TBA–hyaluronate was kept in a freezer until further use. 0.530 g of TBA–hyaluronate were dissolved in 10 ml of DMSO at 30 °C, then 1.5 ml of methyl iodide were added and the solution was kept at 30 °C overnight. The resulting mixture was slowly poured into 200 ml of ethyl acetate under constant agitation. The white precipitate obtained was filtered and washed four times with 100 ml of ethyl acetate and finally vacuum dried for 24 h at room temperature. Methylation of hyaluronate was confirmed by [1]H NMR spectroscopy and size-exclusion chromatography. [1]H NMR spectra were acquired on a Bruker 400 MHz spectrometer in D$_2$O containing 0.125 M sodium deuteroxide to increase hyaluronate proton mobility, thus improving spectra resolution. [1]H NMR spectrum displays peaks of methyl NHCOMe at 1.79–1.86 ppm, as well as peaks of methyl groups OMe at 2.61, 2.78 ppm and COOMe at 3.22 ppm.

## Computational structural characterization of metformin-based Cu complexes

The starting structure for Cu(Met)$_2$ was based on the published X-ray structure[56]. The starting geometries for the other copper(II) complexes were obtained via molecular dynamics conformation search (Gabedit[57], amber99 (ref. 58) potential). For each complex, the ten geometries with the lowest energies resulting from the molecular dynamics search were reoptimized using MOPAC2016 (PM7 (ref. 59), COSMO[60] water model). The geometries with the lowest energy for each complex were optimized at TPSSh/D3BJ/Def2-TZVP level using Orca 4.2.1 (ref. 61). We performed a benchmark study based on the structure of Cu(Met)$_2$ using B3LYP[62], M062X[63], TPSSh, BHLYP[64,65] functionals with the Def2-TZVP[66] basis set using D3BJ[67] dispersion correction and the CPCM water solvation model. We have also used the BHLYP functional with the SVP basis set and the SMD water solvation model, which was recommended in the literature[68] for copper(II) complexes.

## Energy calculation of copper(II)-catalysed hydride transfer to $H_2O_2$ from NADH

The UBHLYP functional[64,65] was used associated with the SVP basis set[69,70] and the SMD solvation model[71,72] to represent an adequate method to describe the [Cu(H$_2$O)$_6$]$^{2+}$ species[68]. Thus, all structures (minima and transition states) were optimized using the Gaussian 16 set

of programs at the UBHLYP/SVP level for all atoms (doublet spin state). The SMD solvation model (water) was applied during the optimization process. Thermal correction to the Gibbs free energy was computed at 310.15 K. Single points at the UMP2/SVP level were performed. The results presented are $\Delta G_{298}$ in kcal mol$^{-1}$. MDHNA was chosen as a NADH model to study the copper(II)-catalysed hydride transfer to $H_2O_2$.

## HRMS of biguanide–metal complexes

HRMS solutions were prepared and injected without further dilution. Stock solution of LCC-12:$K_2CO_3$ (1:2) (100 µM) or metformin:$K_2CO_3$ (1:1) (200 µM) were prepared in analytical grade methanol. Stock solutions of metals, $CuCl_2.2H_2O$ (Acros Organics, 405840050, 100 µM), $MnCl_2$ (Sigma-Aldrich, 244589, 100 µM), $CaCl_2.2H_2O$ (Acros Organics, 423520250, 100 µM), $FeCl_2.xH_2O$ (Alfa Aesar, 12357, 100 µM), $MgCl_2.6H_2O$ (Prolabo, 25 108.295, 100 µM), $NiCl_2$ (Alfa Aesar, 53131, 100 µM) and $ZnCl_2$ (Sigma-Aldrich, 429430, 100 µM) were prepared in milliQ water. The HRMS solutions were prepared in methanol (80 µl) with LCC-12:$K_2CO_3$ (10 µl) and the relevant metal (10 µl) with a 1:1 ratio, or with metformin:$K_2CO_3$ (10 µl) and $CuCl_2.2H_2O$ (10 µl) with a 2:1 ratio.

## UV titration experiments

To a solution of LCC-12 (5 µM) in HEPES (10 mM), portions of 0.1 mol equivalent of a solution of $CuCl_2$ in HEPES (10 mM) were added up to 3 mol equivalent. UV spectra were recorded on an Analytik Jena UV/VIS spectrophotometer specord 205 system at room temperature in the 200–1000 nm range using a micro cuvette (quartz Excellence Q 10 mm). All spectra were blanked against HEPES buffer.

## Copper-catalysed oxidation of NADH

The oxidation kinetics of NADH (Sigma-Aldrich, N4505) were followed by measuring the absorbance at 340 nm using a Cary 300 UV-Vis spectrometer. The measurements were recorded at 37 °C controlled with a Pelletier Cary temperature controller (Agilent Technologies). Stock solutions of NADH (1 mM), imidazole (Sigma-Aldrich, 56750, 100 mM), $CuSO_4$ (Sigma-Aldrich, 451657, 500 µM), LCC-12 (10 mM or 1 mM), metformin·HCl (Alfa Aesar, J63361, 100 mM or 10 mM) and LCC-4,4 (10 mM) were prepared in a 10 mM sodium phosphate buffer adjusted to pH 8.0. The concentration of $H_2O_2$ (Sigma-Aldrich, 16911, 32.3% wt in $H_2O$) was determined by titration with $KMnO_4$ and diluted 100 times in the phosphate buffer. In disposable cuvettes, 1 ml of experimental solutions were prepared using sodium phosphate buffer and respective stock solutions to attain the final concentrations described Fig. 3f, by adding: NADH (200 µl), imidazole (100 µl), $CuSO_4$ (20 µl), LCC-12 (20 µl), LCC-4,4 (20 µl) or metformin (40 µl) as indicated, and $H_2O_2$ solution (19 µl) at $t_0$. The concentration of NADH was calculated from the measured absorbance at 340 nm and its molar extinction coefficient.

## Measurement of NADH concentrations

NADH absolute concentrations were measured using a fluorometric assay (Abcam, ab176723) according to the manufacturer's protocol. At least 500,000 cells were collected per condition. Floating cells were collected and adherent cells were washed with 1× PBS. Adherent cells were incubated with 1× PBS with 10 mM EDTA and then scraped and pooled together with the collected floating cells. Cells were subsequently washed with ice-cold 1× PBS and counted, then centrifuged at 1,500 rpm for 5 min and the supernatant discarded. The pellet was then resuspended in 100 µl lysis buffer (kit component) and incubated at 37 °C for 15 min. NAD$^+$ and NADH extraction solutions as well as NAD$^+$/NADH control solutions (kit components) were added and incubated at 37 °C for 15 min at a volume of 15 µl sample to 15 µl of the respective buffers (kit components). The reactions were stopped using 15 µl of respective buffers (kit components). Finally, 75 µl of NAD$^+$/NADH reaction mixture (NAD$^+$/NADH recycling enzyme mixture and sensor buffer, kit components) were added and the resulting mixtures incubated for

1 h at room temperature. Fluorescence intensities (excitation 540 nm; emission 590 nm) were recorded using a Perkin Elmer Wallac 1420 Victor2 Microplate Reader. Values were derived from the standard curve of each experiment and compared to the data obtained by mass spectrometry-based metabolomics, to calculate total and mitochondrial NADH concentrations.

## Quantitative metabolomics

In a typical experiment, 1.5 million cells were used for total extracts and 15 million cells for mitochondrial extracts. Cells were collected and the supernatant removed to generate the corresponding cell pellets. Subsequently, pellets were dried and supplemented with 300 µl methanol, vortexed 5 min and centrifuged (10 min at 15,000$g$, 4 °C). Then, the upper phase of the supernatant was split into two parts: 150 µl were used for gas chromatography–mass spectrometry (GC–MS) experiment in microtubes and the remaining 150 µl were used for ultra high pressure liquid chromatography–mass spectrometry (UHPLC–MS). For the GC–MS aliquots, supernatants were completely evaporated from the sample. 50 µl of methoxyamine (20 mg ml$^{-1}$ in pyridine) were added to the dried extracts, then stored at room temperature in the dark for 16 h. The following day, 80 µl of $N$-methyl-$N$-(trimethylsilyl) trifluoroacetamide was added and final derivatization occurred at 40 °C for 30 min. Samples were then transferred into vials and directly injected for GC–MS analysis. For the UHPLC–MS aliquots, 150 µl were dried in microtubes at 40 °C in a pneumatically-assisted concentrator (Techne DB3). The dried UHPLC–MS extracts were solubilized with 200 µl of MilliQ water. Aliquots for analysis were transferred into liquid chromatography vials and injected into UHPLC–MS or kept at −80 °C until injection. Widely-targeted analysis of intracellular metabolites gas chromatography coupled to a triple-quadrupole mass spectrometer (QQQGC–MS): the GC–MS/MS method was performed on a 7890A gas chromatography (Agilent Technologies) coupled to a triple-quadrupole 7000C (Agilent Technologies) equipped with a High sensitivity electronic impact source operating in positive mode[73]. Peak detection and integration of the analytes were performed using the Agilent Mass Hunter quantitative software (B.07.01). Targeted analysis of nucleotides and cofactors by ion pairing ultra high performance liquid chromatography (UHPLC) coupled to a Triple Quadrupole (QQQ) mass spectrometer: targeted analysis was performed on a RRLC 1290 system (Agilent Technologies) coupled to a Triple Quadrupole 6470 (Agilent Technologies) equipped with an electrospray source operating in both negative and positive modes. Gas temperature was set to 350 °C with a gas flow of 12 l min$^{-1}$. Capillary voltage was set to 5 kV in positive mode and 4.5 kV in negative mode. Ten microlitres of sample were injected on a Column Zorbax Eclipse XDB-C18 (100 mm × 2.1 mm particle size 1.8 µm) from Agilent technologies, protected by a guard column XDB-C18 (5 mm × 2.1 mm particle size 1.8 µm) and heated at 40 °C by a pelletier oven. The gradient mobile phase consisted of water with 2 mM of dibutylamine acetate concentrate (DBAA) (A) and acetonitrile (B). Flow rate was set to 0.4 ml min$^{-1}$ and an initial gradient of 90% phase A and 10% phase B, which was maintained for 3 min. Molecules were then eluted using a gradient from 10% to 95% phase B over 1 min. The column was washed using 95% mobile phase B for 2 min and equilibrated using 10% mobile phase B for 1 min and the autosampler was kept at 4 °C. Scan mode used was the MRM for biological samples. Peak detection and integration of the analytes were performed using Agilent Mass Hunter quantitative software (B.10.1). Pseudo-targeted analysis of intracellular metabolites by UHPLC coupled to a Q-Exactive mass spectrometer. Reversed phase acetonitrile method: the profiling experiment was performed with a Dionex Ultimate 3000 UHPLC system (Thermo Fisher Scientific) coupled to a Q-Exactive (Thermo Fisher Scientific) equipped with an electrospray source operating in both positive and negative modes and full scan mode from 100 to 1,200 $m/z$. The Q-Exactive parameters were: sheath gas flow rate 55 au, auxiliary gas flow rate 15 au, spray voltage 3.3 kV, capillary temperature 300 °C,

S-Lens RF level 55 V. The mass spectrometer was calibrated with sodium acetate solution dedicated to low mass calibration. 10 µl of sample were injected on a SB-Aq column (100 mm × 2.1 mm particle size 1.8 µm) from Agilent Technologies, protected by a guard column XDB-C18 (5 mm × 2.1 mm particle size 1.8 µm) and heated at 40 °C by a pelletier oven. The gradient mobile phase consisted of water with 0.2% acetic acid (A) and acetonitrile (B). The flow rate was set to 0.3 ml min⁻¹. The initial condition was 98% phase A and 2% phase B. Molecules were then eluted using a gradient from 2% to 95% phase B for 22 min. The column was washed using 95% mobile phase B for 2 min and equilibrated using 2% mobile phase B for 4 min. The autosampler was kept at 4 °C. Peak detection and integration were performed using the Thermo Xcalibur quantitative software (2.1.)[73].

## Quantitative proteomics

Cells were grown and treated as indicated. Whole-cell extracts were collected by scraping after incubation with 1× PBS with 10 mM EDTA at 37 °C. After centrifugation at 1,500$g$ for 5 min at 4 °C, cells were washed twice with ice-cold 1× PBS and lysed using lysis buffer (8 M urea, 200 mM NH$_4$HCO$_3$, cOmplete) for 1 h at 4 °C on a rotary wheel. After centrifugation at 20,000$g$, 4 °C for 20 min, supernatants that contain proteins were used for the global proteome analysis. In brief, the global proteome was quantitatively analysed with a Orbitrap Eclipse mass spectrometer using a label-free approach. About 10 µg of total protein cell lysate were reduced by incubation with 5 mM dithiothreitol (DTT) at 57 °C for 30 min and then alkylated with 10 mM iodoacetamide for 30 min at room temperature in the dark. The samples were then diluted with 100 mM ammonium bicarbonate to reach a final concentration of 1 M urea and digested overnight at 37 °C with Trypsin:Lys-C (Promega, V5071) at a ratio of 1:50. Samples were then loaded onto a homemade C18 StageTips for desalting. Peptides were eluted from beads by incubation with 40:60 acetonitrile:water with 0.1% formic acid. Peptides were dried in a Speedvac and reconstituted in 10 µl 0.3% TFA prior to liquid chromatography-tandem mass spectrometry (LC–MS/MS) analysis. Samples of 4 µl were chromatographically separated using an RSLC-nano system (Ultimate 3000, Thermo Fisher Scientific) coupled online to an Orbitrap Eclipse mass spectrometer (Thermo Fisher Scientific). Peptides were first loaded onto a C18-trapped column (75 µm inner diameter × 2 cm; nanoViper Acclaim PepMap 100, Thermo Fisher Scientific), with buffer A (2:98 MeCN:H$_2$O with 0.1% formic acid) at a flow rate of 3 µl min⁻¹ over 4 min and then switched for separation to a C18 column (75 µm inner diameter × 50 cm; nanoViper C18, 2 µm, 100 Å, Acclaim PepMap RSLC, Thermo Fisher Scientific) regulated to a temperature of 50 °C with a linear gradient of 2 to 30% buffer B (100% MeCN and 0.1% formic acid) at a flow rate of 300 nl min⁻¹ over 211 min. MS1 data were collected in the Orbitrap (120,000 resolution; maximum injection time 60 ms; AGC 4 × 10⁵). Charges states between 2 and 5 were required for MS2 analysis, and a 45 s dynamic exclusion window was used. MS2 scans were performed in the ion trap in rapid mode with HCD fragmentation (isolation window 1.2 Da; NCE 30%; maximum injection time 60 ms; AGC 10⁴). The identity of proteins was established from the UniProt human canonical database (UP000005640_9606) using Sequest HT through proteome discoverer (version 2.4) (Thermo Scientific). Enzyme specificity was set to trypsin and a maximum of two missed cleavage sites were allowed. Oxidized methionine, Methionine-loss, Methionine-loss-acetyl and N-terminal acetylation were set as variable modifications. Carbamidomethylation of cysteins were set as fixed modification. Maximum allowed mass deviation was set to 10 ppm for monoisotopic precursor ions and 0.6 Da for MS/MS peaks. The resulting files were further processed using myProMS v3.9.3 (https://github.com/bioinfo-pf-curie/myproms)[74]. For the false discovery rate (FDR) calculation we used Percolator[75] and, this was set to 1% at the peptide level for the whole study. The label-free quantification was performed by peptide Extracted Ion Chromatograms (XICs) computed with MassChroQ version 2.2.21 (ref. 76). For protein quantification, XICs from proteotypic peptides shared between compared conditions (TopN matching) with up to two missed cleavages and carbamidomethyl modifications were used. Median and scale normalization was applied on the total signal to correct the XICs for each biological replicate. To estimate the significance of the change in protein abundance, a linear model (adjusted on peptides and biological replicates) was performed and $P$ values were adjusted with a Benjamini–Hochberg FDR procedure with a control threshold set to 0.05 and the proteins should have at least 3 peptides[75].

## Lactate quantification

Extracellular lactate was quantified using a fluorometric lactate assay (Abcam, ab65330) according to the manufacturer's instructions. The culture media of cells was collected and centrifuged at 500$g$ for 5 min. The supernatant was subsequently centrifuged at 20,000$g$ for 10 min. The supernatant was then deproteinized using 10 kD spin columns (Abcam, ab93349). A lactate standard was prepared by adding 5 µl of the 100 nmol µl⁻¹ lactate standard to 495 µl of lactate assay buffer. Subsequently, 1 ml of 0.01 nmol µl⁻¹ lactate standard was produced by diluting 10 µl of 1 nmol µl⁻¹ standard to 990 µl of lactate assay buffer. In a 96-well plate, standard samples of 0–0.1 nmol per well were added. A reaction mix of 46 µl lactate assay buffer, 2 µl probe and 2 µl enzyme mix was prepared for each well. For background measurements, 48 µl lactate assay buffer were mixed with 2 µl probe to obtain a background reaction mix. Fifty microlitres of each sample was added to a 96-well plate and either the reaction mix or the background reaction mix was added. Samples were incubated for 30 min at room temperature. Fluorescence intensities (excitation 540 nm; emission 590 nm) were recorded using a Perkin Elmer Wallac 1420 Victor2 Microplate Reader. Values were derived from the standard curve.

## Glyceraldehyde 3-phosphate quantification

Glyceraldehyde 3-phosphate (GA3P) was quantified using a fluorometric glyceraldehyde 3-phosphate assay kit (Abcam, ab273344) adapting the manufacturer's instructions. Cells were washed twice with 1× PBS and then collected into a centrifugation tube in 100 µl GA3P Assay Buffer. Samples were kept on ice for 10 min and then centrifuged at 10,000× g for 10 min. The supernatant was collected and then deproteinized using 10 kD spin columns (Abcam, ab93349). For each test sample, 10 µl of sample were added into three parallel wells in a white, flat bottom 96-well plate. A sample background control, an un-spiked and spiked sample were added to these three wells. The spiked sample contained 200 pmol of GA3P standard. 50 µl of GA3P assay buffer was added per well. For the assay blank, 50 µl GA3P assay buffer was added per well. To each background well, 50 µl of a reaction mix consisting of 46 µl GA3P assay buffer, 2 µl GA3P enzyme mix and 2 µl GA3P probe was added. To each remaining well 50 µl of a reaction mix was added, which consisted of 44 µl GA3P assay buffer, 2 µl GA3P developer, 2 µl GA3P enzyme mix and 2 µl GA3P probe. Fluorescence intensities (excitation 540 nm; emission 590 nm) were recorded using a Perkin Elmer Wallac 1420 Victor2 Microplate Reader. Several readings were performed at 1 min intervals. The final GA3P concentrations were calculated by subtracting background sample values.

## Luminex immunoassay

Cytokine levels were measured in cell culture supernatants using the V-Plex pro-inflammatory panel (MSD, K15049D-1). The kit was run according to the manufacturer's protocol and the chemiluminescence signal was measured on a Sector Imager 2400 (MSD).

## RNA-seq

RNAs were extracted from MDMs using the RNeasy mini kit (Qiagen, 74104). RNA sequencing libraries were prepared from 1 µg total RNA using the Illumina TruSeq Stranded mRNA library preparation kit (Illumina, 20020594), which allows strand-specific sequencing.

A first step of polyA selection using magnetic beads was performed to allow sequencing of polyadenylated transcripts. After fragmentation, cDNA synthesis was performed and resulting fragments were used for dA-tailing followed by ligation of TruSeq indexed adapters (Illumina, 20020492). Subsequently, polymerase chain reaction amplification was performed to generate the final barcoded cDNA libraries. Sequencing was carried out on a NovaSeq 6000 instrument from Illumina based on a 2×100 cycles mode (paired-end reads, 100 bases). For RNA-seq on cells from the in vivo murine models, see the respective paragraphs. Raw sequencing reads were first checked for quality with Fastqc (0.11.8) and trimmed for adapter sequences with the trimGalore (0.6.2) software. Trimmed reads were then aligned on the human hg38 reference genome using the STAR mapper (2.6.1b), up to the generation of a raw count table per gene (GENCODE annotation v29). The bioinformatics pipelines used for these tasks are available online (rawqc v2.1.0: https://github.com/bioinfo-pf-curie/raw-qc, RNA-seq v3.1.4: https://github.com/bioinfo-pf-curie/RNA-seq). The downstream analysis was then restricted to protein-coding genes. Data from the literature[77] were converted into bulk by keeping cells annotated as macrophages and then summing the counts for each sample. Counts data from the literature[44] were downloaded from GEO under accession number GSE73502. Raw data from the literature[45,46] were downloaded from the NCBI Short Read Archive under records PRJNA528433 and PRJNA290995 and processed as described above. For *L. major*, we used data at 4 h post-infection; for *A. fumigatus*, we used data at 2 h post-infection. Counts were normalized using TMM normalization from edgeR (v 3.30.3)[78]. Differential expression was assessed with the limma/voom framework (v 3.44.3)[79]. The intra-donor correlation was controlled by using the duplicateCorrelation from limma. Genes with an adjusted *P* value < 0.05 were labelled significant. Enrichment analysis from differentially expressed genes has been performed using the enrichGO function from clusterProfiler package v3.16.1.

## ChIP–seq

Cells were grown and treated as described. Cells were centrifuged at 1,500*g* for 5 min at room temperature. The pelleted cells were resuspended in medium, counted and crosslinked with 1% formaldehyde for 10 min at room temperature. Then, 2.5 M glycine was added to a final concentration of 0.125 M and incubated for 5 min at room temperature followed by centrifugation at 1,500*g* for 5 min at 4 °C. The pelleted cells were washed twice with ice-cold 1× PBS and collected by centrifugation at 1,500*g* for 5 min at 4 °C. Pellets were resuspended in lysis buffer A (50 mM Tris-HCl pH 8, 10 mM EDTA, 1% SDS, cOmplete) and incubated for 30 min on a rotating wheel at 4 °C. Next, lysates were centrifuged at 1,500*g* for 15 min at 8 °C to prevent SDS precipitation, and the supernatants were discarded. Pellets were then sheared in buffer B (25 mM Tris-HCl pH 8, 3 mM EDTA, 0.1% SDS, 1% Triton X-100, 150 mM NaCl, cOmplete) to approximately 200–600 bp average size using a Bioruptor Pico (Diagenode). After centrifugation at 20,000*g* at 4 °C for 15 min, supernatants containing sheared chromatin were used for immunoprecipitation. Twenty-five microlitres (10%) of sheared chromatin was used as input DNA to normalize sequencing data. As an additional control for normalization, spike-in chromatin from *Drosophila* (Active Motif, 53083) and a spike-in antibody were used. Chromatin immunoprecipitation (1 million cells per ChIP condition) was carried out using sheared chromatin and antibodies against specific histone marks, which were subsequently complexed to either Dynabeads Protein G-coated magnetic beads (Invitrogen, 10003D) for H3K27ac, H3K14ac, H3K9ac, H3K27me3 and H3K9me2. In brief, each antibody was mixed with 1 µg of spike-in antibody. Then, 22 µl magnetic beads were washed three times in ice-cold buffer C (20 mM Tris-HCl pH 8, 2 mM EDTA, 0.1% SDS, 1% Triton X-100, 150 mM NaCl, cOmplete) and incubated with the mixture of antibodies for 4 h, at room temperature on a rotating wheel in buffer C (494 µl). After spinning and removal of supernatants, beads were resuspended in 50 µl buffer C.

This suspension was subsequently incubated with 250 µl sheared chromatin previously mixed with 50 ng of spike-in chromatin from *Drosophila* (250 µl chromatin of interest: 2.5 µl spike-in chromatin) at 4 °C on a rotating wheel overnight (16 h). After a spinning, supernatants were discarded and beads were successively washed in buffer C (twice), buffer D (20 mM Tris-HCl pH 8, 2 mM EDTA, 0.1% SDS, 1% Triton X-100, 500 mM NaCl), buffer E (10 mM Tris-HCl pH 8, 0.25 M LiCl, 0.5% NP-40, 0.5% sodium deoxycholate, 1 mM EDTA) and once in buffer F (10 mM Tris-HCl pH 8, 1 mM EDTA, 50 mM NaCl). Finally, input and immunoprecipitated chromatin samples were resuspended in a solution containing TE buffer/1% SDS, de-crosslinked by heating at 65 °C overnight and subjected to both RNase A (Invitrogen, 12091-039, 1 mg ml$^{-1}$) and Proteinase K (Thermo Scientific, EO0491, 20 mg ml$^{-1}$) treatments. Input and immunoprecipitated DNA extraction: after reverse crosslinking, input and immunoprecipitated chromatin samples were treated with RNase A and proteinase K, and glycogen (Thermo Scientific, R0561, 20 mg ml$^{-1}$) was added. Samples were incubated at 37 °C for 2 h. DNA precipitation was carried out using 8 M LiCl (final concentration 0.44 M) and phenol:chloroform:isoamyl alcohol. Samples were vortexed and centrifuged at 20,000*g* for 15 min at 4 °C. The upper phase was mixed with chloroform by vortexing. After centrifugation at 20,000*g* for 15 min at 4 °C, the upper phase was mixed with −20 °C absolute ethanol by vortexing and stored at −80 °C for 2 h. Next, samples were pelleted at 20,000*g* for 20 min at 4 °C. The pellets were washed with ice-cold 70% ethanol and centrifuged at 20,000*g* for 15 min at 4 °C. The supernatants were discarded and pellets were dried at room temperature, dissolved in nuclease-free water and quantified using a Qubit fluorometric assay (Invitrogen) according to the manufacturer's protocol. Library preparation and sequencing: Illumina compatible libraries were prepared from input and immunoprecipitated DNAs using the Illumina TruSeq ChIP library preparation kit according to the manufacturer's protocol (IP-202-1012). In brief, 4 to 10 nanograms of DNA were subjected to end-repair, dA-tailing and ligation of TruSeq indexed Illumina adapters. After a final PCR amplification step (with 15 cycles), the resulting barcoded libraries were equimolarly pooled and quantified by quantitative PCR using the KAPA library quantification kit (Roche, 07960336001). Sequencing was performed on the NovaSeq 6000 (Illumina), targeting 75 million clusters per sample and using paired-end 2×100 bp. ChIP–seq data processing and quality controls have been performed with the Institut Curie ChIP–seq Nextflow pipeline (1.0.6) available at https://github.com/bioinfo-pf-curie/ChIP-seq. In brief, reads were trimmed for adapter content, and aligned on the Human reference genome hg38 with BWA-mem. Low-quality mapped reads, reads aligned on ENCODE blacklist regions, reads aligned on the spike-in genome and reads marked as duplicates were discarded from the analysis. Bigwig tracks were then generated with deeptools and normalized to 1 million reads to account for differences in sequencing depth. In order to integrate the histone mark enrichments with the gene expression data (RNA-seq), the ChIP–seq signal has been counted either at the transcription start site level (±2 kb) for permissive histone marks or at the gene body for repressive histone marks. Coding genes from Gencode v34 have been used for the annotations. ChIP–seq counts data have then been filtered to remove low counts, and normalized using the TMM methods (edgeR R package). Fold-changes have been then calculated for all genes and donors.

## In vivo animal studies

Survival assessment using the LPS mouse model was conducted at Fidelta (now Selvita) according to 2010/63/EU and National legislation regulating the use of laboratory animals in scientific research and for other purposes (Official Gazette 55/13). An institutional committee on animal research ethics (CARE-Zg) monitored animal-related procedures to ensure they were not compromising animal welfare. Experiments were performed on eight-week-old male BALB/c mice. Other experiments involving LPS and CLP mouse models were performed

in accordance with French laws concerning animal experimentation (#2021072216346511) and approved by the Institutional Animal Care and Use Committee of Université de Saint-Quentin-en-Yvelines (C2EA-47). These LPS experiments were performed on eight-week-old male BALB/c mice and five-week-old male SWISS mice and experiments involving the CLP model were performed on nine-week-old male BALB/c mice. Mice were housed in a state-of-the-art animal care facility (2CARE, prefectural number agreement: A78-322-3, France). For experiments involving SARS-CoV-2, 8-week-old male K18-human ACE2-expressing C57BL/6 mice were used. A SARS-CoV-2 mouse model was used within the biosafety level 3 facility of the Institut Pasteur de Lille, after validation of the protocols by the local committee for the evaluation of the biological risks and complied with current national and institutional regulations and ethical guidelines (Institut Pasteur de Lille/B59-350009). The experimental protocols using animals were approved by the institutional ethical committee 'Comité d'Ethique en Experimentation Animale (CEEA) 75, Nord-Pas-de-Calais'. The animal study was authorized by the 'Education, Research and Innovation Ministry' under registration number APAFIS#25517-2020052608325772v3. A SARS-CoV-2 mouse model was used within the biosafety level 3 facility of the University of Toulouse. This work was overseen by an Institutional Committee on Animal Research Ethics (License APAFIS#27729-2020101616517580 v3, Minister of Research, France (CEEA-001)), to ensure that animal-related procedures were not compromising the animal welfare. Four mice per cage were housed in in a ventilated rack with a media enrichment element. Mice in all animal facilities were housed in ventilated cages (temperature 22 °C ± 2 °C, humidity 55% ± 10%) with free access to water and food on a 12 h light/dark cycle. Male littermates were randomly assigned to experimental groups throughout.

## LPS-induced sepsis model

Survival assessment was conducted at Fidelta (now Selvita). LPS (Sigma-Aldrich, L2630, 20 mg kg$^{-1}$) was injected intraperitoneally to male BALB/c mice (8 weeks old). LCC-12 (0.3 mg kg$^{-1}$, intraperitoneal injection, $n = 10$) or vehicle (0.9% NaCl, 10 ml kg$^{-1}$, intraperitoneal injection, $n = 10$) were injected 2 h prior LPS challenge, then 24 h, 48 h, 72 h and 96 h after challenge. Dexamethasone (10 mg kg$^{-1}$, oral gavage, $n = 10$) was given 1 h before LPS challenge. Incidence of mortality was monitored every 4 h up to 48 h, then twice daily. The cytometry work, ICP-MS, western blotting and RNA-seq work were performed on eight-week-old male BALB/c mice and five-week-old male SWISS mice. The experimental endotoxemic model was induced by intraperitoneal injection of LPS (5 mg kg$^{-1}$ in BALB/c mice (*Escherichia coli* O111:B4, Sigma-Aldrich, L2630) or 20 mg kg$^{-1}$ in SWISS mice (*E. coli* O55:B5, Sigma-Aldrich, L2880)). All the mice were resuscitated with 30 ml kg$^{-1}$ body weight of saline administered subcutaneously 6 h after LPS administration. LCC-12 (0.3 mg kg$^{-1}$) was injected intraperitoneally. Mice were killed 22 h post LPS challenge. Body temperature was measured at 0 h, 6 h and 22 h post LPS challenge. Flow cytometry: after euthanasia at 22 h post LPS challenge, the organs were perfused with PBS/EDTA (1 ml g$^{-1}$, 2 mM, pH 7.4) and 10 ml of 1× PBS were injected in the peritoneum. The peritoneal liquid was then collected and centrifugated for 5 min at 1,500 rpm. The pellet was resuspended in RPMI medium containing 2% fetal calf serum (Dutscher, S181H-100). The peritoneal liquid was then washed in 96-well plates at 2,000 rpm for 2 min, and pellets were suspended in complete RPMI. For intracellular protein staining, samples were incubated for 2 h in brefeldin A (BFA, 5 ng ml$^{-1}$, Invitrogen, 00-4506-51) before surface staining with Fixable Viability Dye eFluor 780 (Invitrogen, 65-0865-14) followed by fluorochrome-conjugated antibodies (35 min at 4 °C). Samples incubated with BFA were fixed (Foxp3/Transcription factor Fix/Perm 4X, Cell Signaling, 44931S) for 20 min at 4 °C and permeabilized (Flow Cytometry Perm Buffer 10X, TONBO, TNB-1213-L150) before intracellular staining. The antibodies used were as follows: CD11b-Pacific Blue (BioLegend, 101224),

CD40-APC (BioLegend, 124612), CD45-BV510 (BioLegend, 103138), CD86-PE (BioLegend, 105007), CD170 (Siglec-F)-PEeFluor 610 (eBioscience, 61-1702-80), F4/80-BV605 (BioLegend, 123133), I-A/I-E-AF700 (BioLegend, 107622), Ly6C–PerCP/Cy5.5 (BioLegend, 128012) and Ly6G–PE/CY7 (BioLegend, 127618). For intracellular staining, NOS2–APC (eBioscience, 17-5920-82) was used. SPMs correspond to CD45$^+$ IA-IE$^+$CD11b$^+$F4/80$^{int}$SiglecF$^-$ cells. After washing with PBS or with Perm Buffer, data were acquired using an LSR Fortessa flow cytometer (BD Biosciences) and analysed with FlowJo. software v. 10.8.2. ICP-MS experiments were conducted using SPMs as described in 'ICP-MS'. Tissue-specific data were normalized against dry weight. Sorting of SPMs was performed using the following antibodies: CD11b-Pacific Blue (BioLegend, 101224), F4/80-PE (TONBO, TNB50-4801-U100), Ly6C–PerCP/Cy5.5 (BioLegend, 128012) and Ly6G–AF647 (BioLegend, 127610). The sorted SPMs corresponded to CD11b$^+$F4/80$^{int}$Ly6C$^-$Ly6G$^-$ cells and were isolated on a BD FACSAria IIIu. RNA-seq: the CD11b$^+$F4/80$^{int}$Ly6C$^-$Ly6G$^-$ sorted SPMs from 7 LPS-treated mice (102283 SPM) and from 8 LPS and LCC-12-treated SWISS mice (66967 SPM) were centrifuged, then resuspended in 350 μl of TCL (Qiagen, 1031576) and 1% β-mercaptoethanol. Total RNA was extracted with the Norgen Single Cell RNA Purification kit (Norgen, 51800). RNA sequencing libraries were prepared using the SMARTer Stranded Total RNA-seq Kit v2−Pico Input Mammalian (Clontech/Takara, 634419). The input quantity of total RNA was 10 ng for each condition. A first step of RNA fragmentation, was applied using a proprietary fragmentation mix at 94 °C for 4 min. After fragmentation, indexed cDNA synthesis was performed. Then the ribodepletion step was performed using probes specific to mammalian rRNA. PCR amplification (12 cycles) was done to amplify the cDNA libraries. Library quantification and quality assessment was performed using a Qubit fluorometric assay (Invitrogen, Q32854) with dsDNA HS (High Sensitivity) Assay Kit and LabChip GX Touch using a High Sensitivity DNA chip (Perkin Elmer, 760517, CLS760672). Libraries were then equimolarly pooled and quantified by quantitative PCR using the KAPA library quantification kit (Roche, 07960336001). Sequencing was performed on a NovaSeq 6000 (Illumina) targeting 100 million clusters per sample and using paired-end 2× 100 bp. For the analysis, see 'RNA-seq'. Trimmed reads were aligned on the mouse mm10 reference genome.

## CLP-induced sepsis model

Nine-week-old male BALB/c mice were used for these experiments. Animals were anaesthetized by isoflurane (Forene). After abdominal incision, the cecum was ligated, punctured with a gauge needle (25G), and a small amount of fecal matter was released. For the sham group, after abdominal incision, the cecum was manipulated but was neither ligated nor punctured. After the cecum was returned to the abdomen, the abdominal cavity was closed in two layers and the mice were resuscitated with 30 ml kg$^{-1}$ body weight of saline (0.9% NaCl) administered subcutaneously. LCC-12 (0.3 mg kg$^{-1}$, intraperitoneal injection) was administered at 4 h, 24 h, 48 h, 72 h and 96 h following CLP creation. Mortality incidence was monitored every 2 h up to 120 h (except from 10 pm to 6 am) after CLP creation. Dexamethasone was administered intraperitoneally at 1 mg kg$^{-1}$ 5 min before CLP creation. ICP-MS experiments were conducted on SPMs as described in 'ICP-MS'. Tissue-specific data were normalized against dry weight. The sorting of SPMs was done using the following antibodies: CD11b-Pacific Blue (BioLegend, 101224), F4/80-PE (TONBO, TNB50-4801-U100), Ly6C–PerCP/Cy5.5 (BioLegend, 128012) and Ly6G–AF647 (BioLegend, 127610). The sorted SPMs corresponded to CD11b$^+$F4/80$^{int}$Ly6C$^-$Ly6G$^-$ cells.

## SARS-CoV-2-induced acute inflammation model

Eight-week-old male K18-human ACE2-expressing C57BL/6 mice (B6.Cg-Tg(K18-hACE2)2Prlmn/J) were purchased from Jackson Laboratory. The mice were anaesthetized by intraperitoneal injection of ketamine (100 mg kg$^{-1}$) and xylazine (10 mg kg$^{-1}$) and then intranasally infected

with 50 μl of DMEM containing $5 \times 10^2$ $TCID_{50}$ of hCoV-19_IPL_France strain of SARS-CoV-2 (NCBI MW575140). LCC-12 was inoculated intranasally (0.5 mg ml$^{-1}$, 50 μl) 6 h, 24 h and 48 h post-infection. Mice were killed at day 4 post-infection. Cell sorting: K18-hACE2 mice (Jackson Laboratory, male, 10 mice per group) were infected under a short anaesthesia (isoflurane 4%), by the intranasal route with $10^5$ PFU of SARS-CoV-2 (strain Be-taCoV/France/IDF0372/2020). During the 4 days of infection, mice were monitored daily for disease progression. At day 4 post-infection, after a terminal anaesthesia (ketamine 100 mg kg$^{-1}$ and xylazine 10 mg kg$^{-1}$, intraperitoneal injection), lung tissues were collected and homogenized in gentleMACS C Tubes (Miltenyi Biotec) containing 2.5 ml of RPMI 1640 medium (Gibco) and collagenase (2.5 mg ml$^{-1}$, Roche) using a gentleMACS Dissociator (Miltenyi Biotec). Lung tissues were further dissociated for 30 min at 37 °C under shaking, passed through a 70 μm cell strainer, and proceeded for red blood cell lysis before staining for cell sorting. Cells were stained with Fixable Viability Dye eFluor 780 (Invitrogen, 65-0865-14) followed by fluorochrome-conjugated antibodies (35 min at 4 °C). The antibodies used were as follows: CD11b-Pacific Blue (BioLegend, 101224), CD45-BV510 (BioLegend, 103138), CD170 (Siglec-F)-PEeFluor 610 (eBioscience, 61-1702-80), F4/80-BV605 (BioLegend, 123133). AMs correspond to CD45$^+$CD11b$^{int}$F4/80$^+$SiglecF$^+$ cells and were isolated on a BD FACSAria Fusion. ICP-MS experiments were conducted in AMs as described in 'ICP-MS'. RNA-seq: half of the right lobe was homogenized in 1 ml of RA1 buffer from the NucleoSpin RNA kit (Macherey Nagel, 740955.250) containing 20 mM of tris(2-carboxyethyl)phosphine (TCEP). Total RNAs in the tissue homogenate were extracted with the NucleoSpin RNA kit. RNA was eluted with 60 μl of water. For the following steps, see 'RNA-seq'. Trimmed reads were aligned on the mouse mm10 reference genome.

## Software for illustrations

Illustrations were created using FIJI 2.0.0-rc-69/1.52n, Prism 8.2.0 and Adobe Illustrator 26.0.2 and BioRender.com. BioRender.com was used for Figs. 1a and 5a and Extended Data Fig. 10g.

## Quantification, statistical analysis and reproducibility

Results are presented as mean ± s.e.m or mean ± s.d. as indicated. In box plots, boxes represent interquartile range and median, and whiskers indicate the minimum and maximum values. A specific colour of dots on a box plot represents a distinct donor only within a given figure panel. Each donor or mouse represents an independent biological sample. Prism 8.2.0 software was used to calculate $P$ values using a two-sided Mann–Whitney test, two-sided unpaired $t$-test, Kruskal–Wallis test with Dunn's post test, two-way ANOVA or Mantel–Cox log-rank test as indicated. Prism 8.2.0 software or the R programming language was used to generate graphical representations of quantitative data unless stated otherwise. Exact $P$ values are indicated in the figures. Sample sizes ($n$) are indicated in the figure legends. All immunofluorescence experiments were repeated with at least $n = 3$ donors with similar results. Western blotting on macrophages isolated from mice was performed on pooled samples and performed once per pool. Morphological changes observed between naMDMs and aMDMs were observed in $n = 128$ donors and representative images of $n = 1$ donor are displayed in Extended Data Fig. 1c. NanoSIMS imaging was performed on $n = 1$ donor and a representative image is displayed in Extended Data Fig. 5b.

## Materials availability

In-house reagents can be made available under a material transfer agreement with Institut Curie. Inquiries should be addressed to R.R.

## Reporting summary

Further information on research design is available in the Nature Portfolio Reporting Summary linked to this article.

## Data availability

RNA-seq and ChIP–seq data are available at the Gene Expression Omnibus with accession reference GSE160864. The mass spectrometry proteomics raw data have been deposited to the ProteomeXchange Consortium via the PRIDE[80] partner repository with the dataset identifier PXD038612. The donor number corresponds to the order of blood collection. Source data are provided with this paper.

## Code availability

Analysis scripts for RNA-seq and ChIP–seq data are available at https://github.com/bioinfo-pf-curie/MDMmetals.

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

**Acknowledgements** R.R. thanks J.-M. Lehn, S. Schreiber and R. Vale. This work was supported by the CNRS, INSERM, PSL Research University, Paris Saclay University, Ecole Polytechnique, University of Bath, University of Lille and Institut Pasteur (Lille). R.R. was supported by the European Research Council under the European Union's Horizon 2020 research and innovation programme (grant agreement no. 647973), Foundation Charles Defforey-Institut de France, Region IdF for NMR infrastructure, Ligue Contre le Cancer and the Ladies of Pompadour. G.K. was supported by ANR Grant Wilsonmed and LabEx Immuno-Oncology (ANR-18-IDEX-0001). D.A. was supported by FHU Sepsis and Programme d'Investissements d'avenir ANR-18-RHUS-0004-RHU records. ICP-MS platform at Institut de Physique du Globe de Paris is supported by IPGP multidisciplinary programme PARI and Region IdF (SESAME grant agreement no. 12015908). This work was granted access to the HPC resources of CINES under the allocation 2020-A0070810977 made by GENCI, the Balena High Performance Computing (HPC) Service at the University of Bath, the PICT-IbiSA@BDD Imaging Facility of Institut Curie, member of the France-BioImaging national research infrastructure (ANR-10-INBS-04), the ICGex NGS platform of Institut Curie (ANR-10-INBS-09-08, INCa-DGOS-465, INCa-DGOS-Inserm_12554), GenoToul ANEXPLO Animal Level 3 and TRI Facilities of the IPBS (Investissement d'Avenir and Foundation Bettencourt), the mass spectrometry platform supported by Région Ile-de-France (N°EX061034) and ITMO Cancer of Aviesan and INCa on funds administered by INSERM (N°21CQ016-00), and the cytometry platform of Institut Curie. We thank P. Benaroch, J.-L. Guerquin-Kern, M. Plays, J. Sampaio Lopes, D. Guillemot, O. Delattre and O. Neyrolles for support.

**Author contributions** R.R. conceptualized the study and directed the research. R.R., S.M., S.S. and T.C. designed the experiments and analysed the data. S.M. and S.S. performed the experiments unless stated otherwise. T.C., A.V., L.B. and C.G. performed chemical synthesis and analytical chemistry. G.D.P. and V.G. assisted with molecular modelling. T.-D.W. assisted with NanoSIMS imaging. S.D. and G.K. assisted with mass spectrometry-based metabolomics. F.S., F.D. and D.L. performed mass spectrometry-based proteomics. F.S., P.G., S.B. and N.S. performed Solexa/Illumina sequencing and bioinformatics. A.M., L.E. and D.A. established the preclinical models of sepsis and performed SPM isolation. V.S., C.R., F.T., D.P., E.N., C.C. and E.M. performed SARS-CoV-2 infection and alveolar macrophage isolation. R.R. and S.M. wrote the article. All authors contributed to drafts of the article.

**Competing interests** Institut Curie and the CNRS have filed patents on the LCC family of compounds and their therapeutic use. Patents: WO 2019/233982, filed on 4 June 2019; PCT/EP2021/082073, filed on 18 November 2021, WO 2021/233962, filed on 19 May 2021.

### Additional information
**Correspondence and requests for materials** should be addressed to Raphaël Rodriguez.

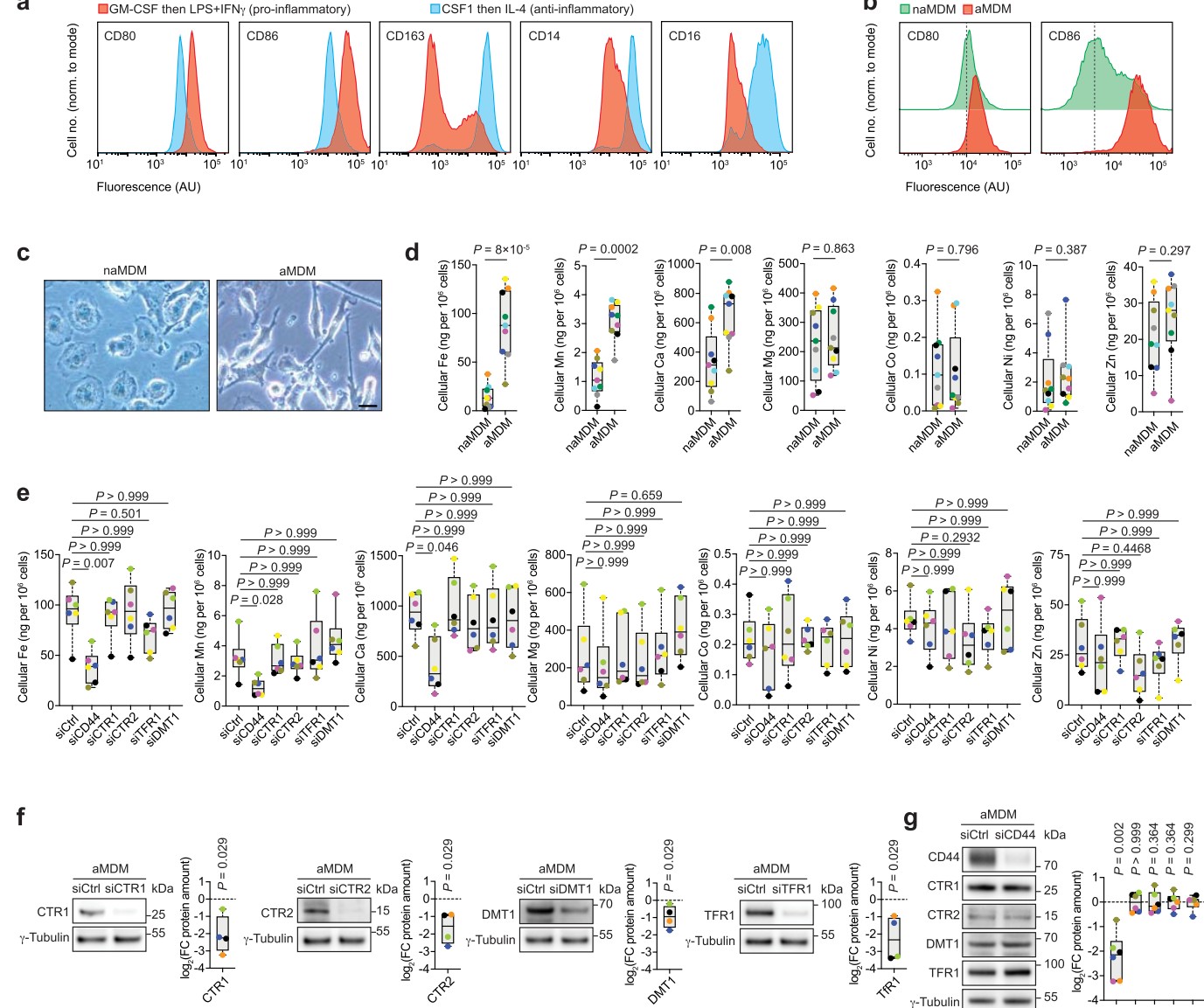

**Extended Data Fig. 1 | CD44 mediates the uptake of metals in inflammatory macrophages. a**, Flow cytometry of cell surface markers in MDM. Monocytes treated as indicated to obtain pro-inflammatory or anti-inflammatory states. **b**, Flow cytometry of cell surface markers in MDM. Data representative of *n* = 13 donors. **c**, Bright field microscopy images of MDM. Scale bar, 20 μm. Morphological observations representative of *n* = 128 donors. **d**, ICP-MS of cellular metals in MDM (*n* = 9 donors). **e**, ICP-MS of cellular metals in aMDM under knockdown conditions of indicated genes (*n* = 6 donors). **f**, Western blots of cellular metal transporters in aMDM under knockdown conditions of indicated genes (*n* = 4 donors). **g**, Western blots of cellular metal transporters in aMDM under CD44 knockdown conditions (*n* = 6 donors). For **d**, **f** and **g** two-sided Mann-Whitney test. For **e**, Kruskal-Wallis test with Dunn's post-test. Box plots: boxes represent interquartile range and median and whiskers indicate the minimum and maximum values. Each colored dot represents a distinct donor for a given panel.

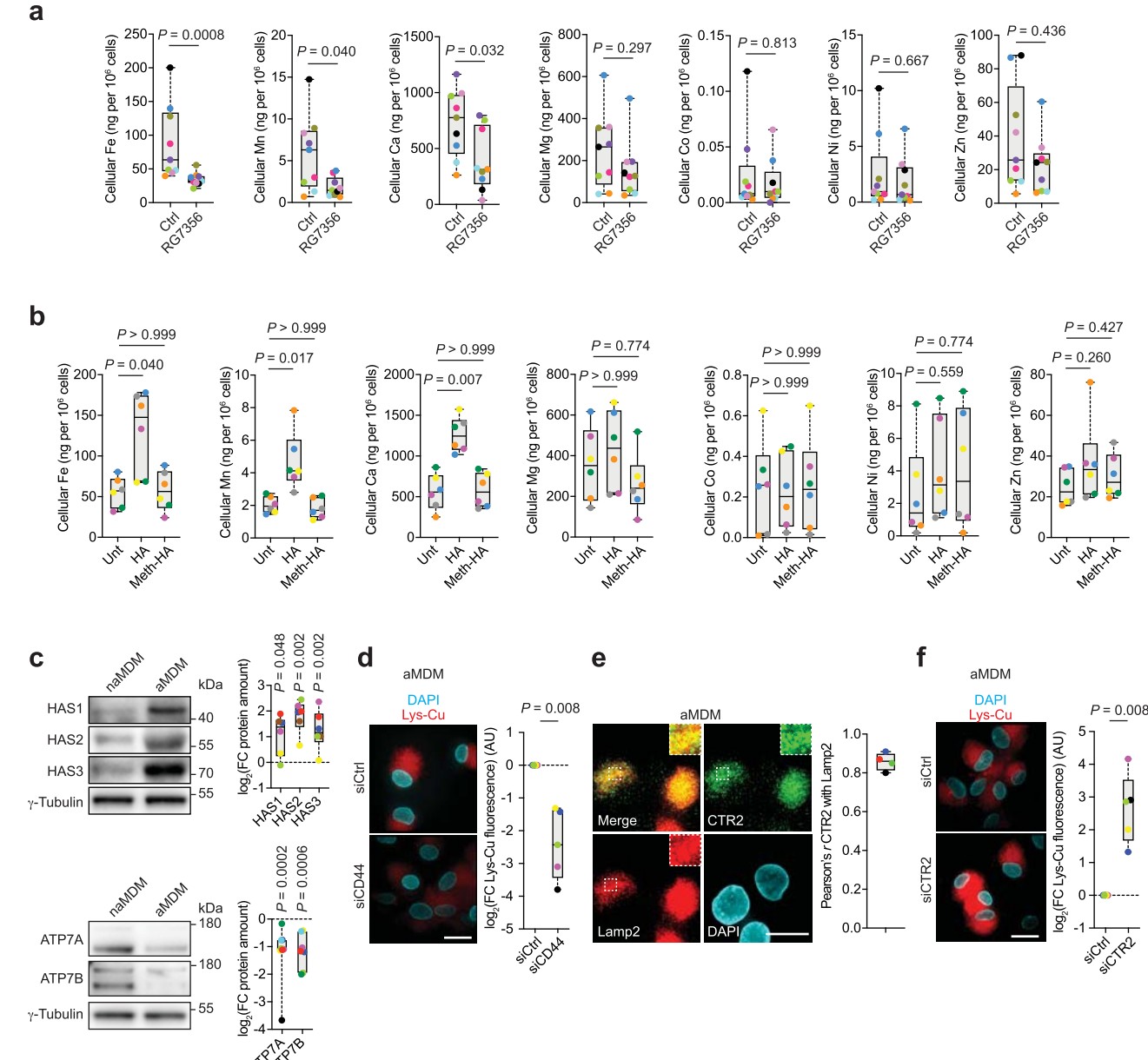

**Extended Data Fig. 2 | CD44 mediates the uptake of metals in inflammatory macrophages. a**, ICP-MS of cellular metals in aMDM treated with an anti-CD44 antibody RG7356 during activation (*n* = 7 donors). **b**, ICP-MS of cellular metals in aMDM supplemented with HA (0.6-1 MDa) or permethylated (meth-) HA during activation (*n* = 6 donors). **c**, Western blots of hyaluronan synthases (HAS) (*n* = 6 donors) and ATP7A/B (*n* = 8 donors) in MDM. **d**, Fluorescence microscopy of a lysosomal copper(II) probe (Lys-Cu) in aMDM under CD44 knockdown conditions (*n* = 5 donors). **e**, Fluorescence microscopy of CTR2 and Lamp2 in

aMDM (*n* = 4 donors). **f**, Fluorescence microscopy of a lysosomal copper(II) probe (Lys-Cu) in aMDM under CTR2 knockdown conditions (*n* = 5 donors). For **d**–**f**, Scale bars, 10 μm. For **a**, **c**, **d** and **f** two-sided Mann-Whitney test. For **b**, Kruskal-Wallis test with Dunn's post-test. Box plots: boxes represent interquartile range and median, and whiskers indicate the minimum and maximum values. Each colored dot represents a distinct donor for a given panel.

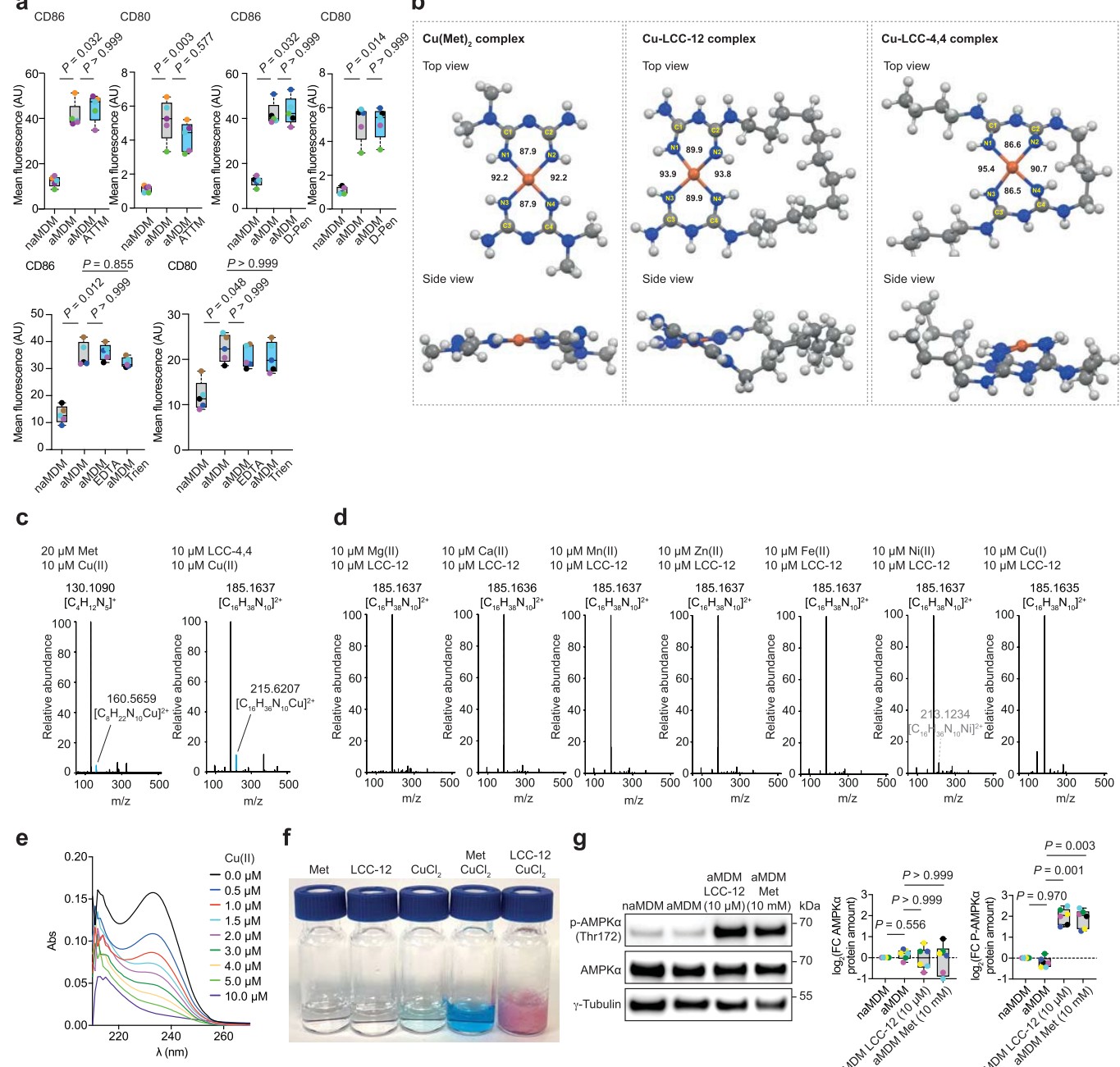

**Extended Data Fig. 3 | Development of a small molecule inactivator of mitochondrial copper(II). a**, Flow cytometry of MDM treated with ATTM (10 μM, $n = 5$ donors), D-Pen (250 μM, $n = 5$ donors), EDTA (500 μM, $n = 5$ donors) or Trien (200 μM, $n = 5$ donors). **b**, Structural analysis of biguanide-based copper(II) complexes by molecular modeling. Top and side views highlight distinct geometries of Cu(Met)$_2$, Cu–LCC-12 and Cu–LCC-4,4. **c**, HRMS of Cu(Met)$_2$ and Cu–LCC-4,4. **d**, HRMS of LCC-12 in the presence of metals as indicated. **e**, UV absorbance spectra of LCC-12 (5 μM) titrated with a solution of copper(II). **f**, Picture of aq. solutions of Met, LCC-12, CuCl$_2$ and corresponding mixtures. **g**, Western blots of AMPKα and phosphorylated AMPKα (p-AMPKα) in MDM treated with LCC-12 or Met ($n = 6$ donors). For **a** and **g** Kruskal-Wallis test with Dunn's post-test. Box plots: boxes represent interquartile range and median, and whiskers indicate the minimum and maximum values. Each colored dot represents a distinct donor for a given panel.

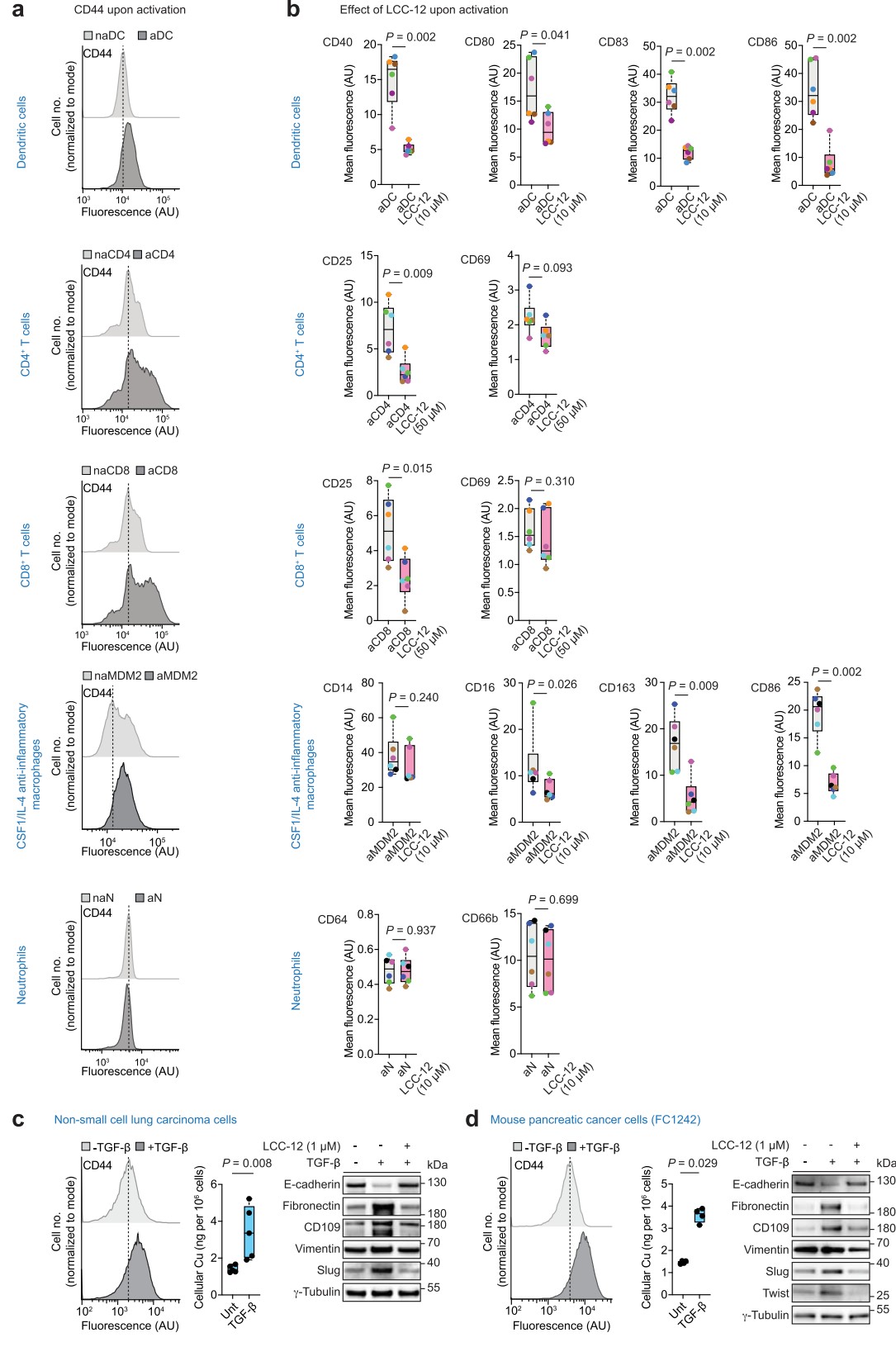

**Extended Data Fig. 4** | See next page for caption.

**Extended Data Fig. 4 | Targeting mitochondrial copper(II) interferes with cell plasticity. a**, Flow cytometry of CD44 in dendritic cells (DC) ($n = 6$ donors), non-activated (naDC) and activated (aDC), CD4$^+$ T cells (CD4) ($n = 6$ donors), non-activated (naCD4) and activated (aCD4), CD8$^+$ T cells (CD8) ($n = 6$ donors), non-activated (naCD8) and activated (aCD8), CSF1/IL-4 anti-inflammatory macrophages ($n = 6$ donors), non-activated (naMDM2) and activated (aMDM2), neutrophils (N) ($n = 6$ donors), non-activated (naN) and activated (aN). **b**, Flow cytometry of cell surface markers in immune cells treated with LCC-12 during activation. **c**, Primary human non-small cell lung circulating cancer cells treated as indicated. Left: Flow cytometry of CD44. Middle: ICP-MS of cellular copper ($n = 5$ independent biological experiments). Right: Western blots of EMT markers. **d**, Murine pancreatic cancer cells treated as indicated. Left: Flow cytometry of CD44. Middle: ICP-MS of cellular copper ($n = 4$ independent biological experiments). Right: Western blots of EMT markers. For **b**–**d** two-sided Mann-Whitney test. Box plots: boxes represent interquartile range and median, and whiskers indicate the minimum and maximum values. Each colored dot represents a distinct donor for a given cell type.

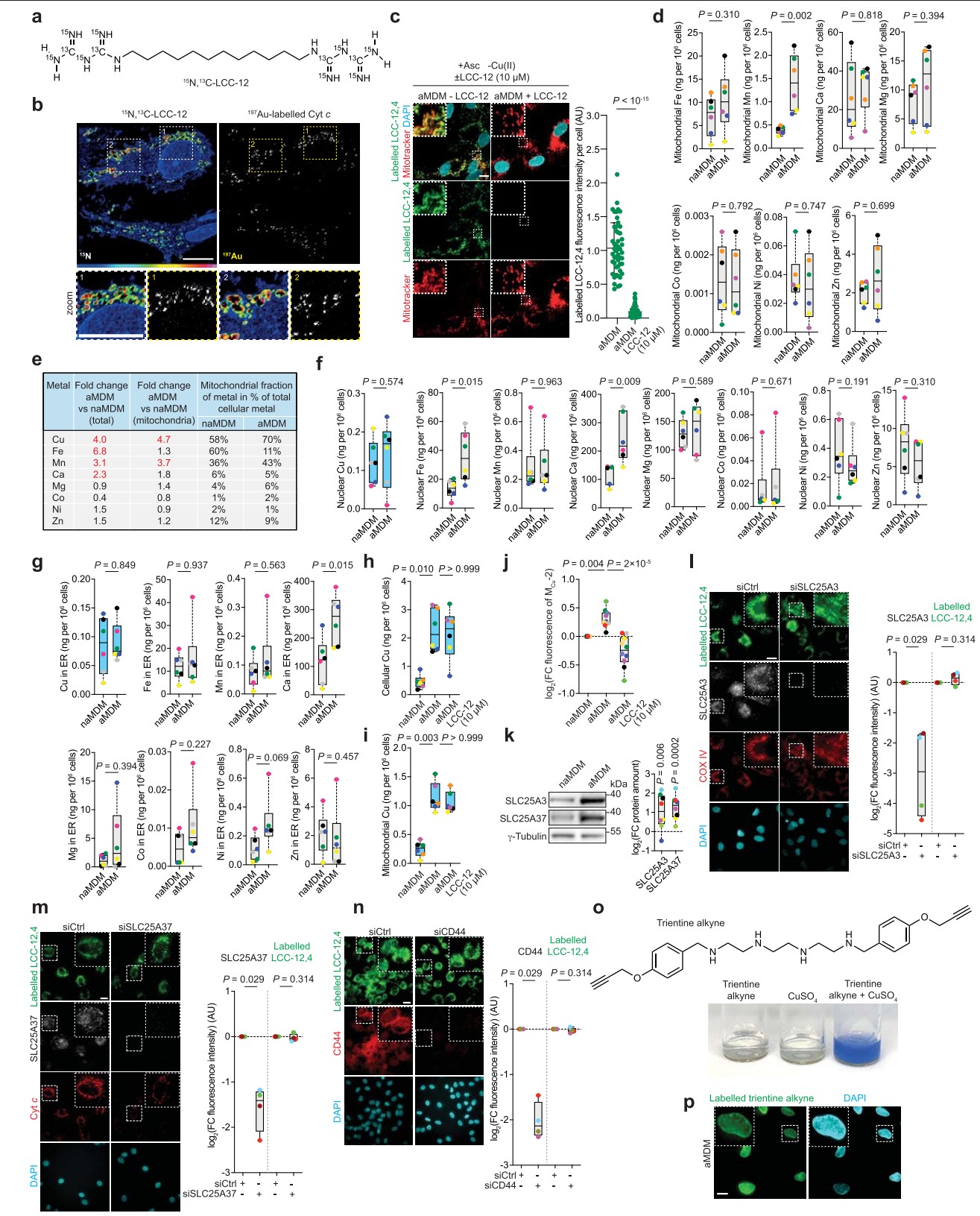

**Extended Data Fig. 5 | See next page for caption.**

**Extended Data Fig. 5 | Detection of a druggable pool of copper(II) in mitochondria. a**, Molecular structure of isotopologue $^{15}N,^{13}C$-LCC-12. **b**, NanoSIMS image of $^{15}N$ and $^{197}Au$ in aMDM of $n = 1$ donor. **c**, Fluorescence microscopy of labelled LCC-12,4 (100 nM) in aMDM. In-cell-labelling performed without added copper(II) and using LCC-12 as a competitor. Representative of $n = 3$ donors. **d**, ICP-MS of metals in mitochondria of MDM ($n = 6$ donors). **e**, Comparison of the total metal contents in cells and mitochondria of MDM determined by ICP-MS. **f**, ICP-MS of metals in nuclei isolated from MDM ($n = 6$ donors). **g**, ICP-MS of metals in endoplasmic reticula (ER) isolated from MDM ($n = 6$ donors). **h**, ICP-MS of the total cellular copper content in MDM treated with LCC-12 ($n = 6$ donors). **i**, ICP-MS of mitochondrial copper in MDM treated with LCC-12 ($n = 6$ donors). **j**, Flow cytometry of a mitochondrial copper(II) probe ($M_{Cu}$-2) in MDM treated with LCC-12 ($n = 10$ donors). **k**, Western blots of mitochondrial metal transporters in MDM ($n = 8$ donors). **l, m, n**, Fluorescence microscopy of labelled LCC-12,4 in aMDM under gene knockdown conditions as indicated ($n = 4$ donors). **o**, Top: Structure of trientine alkyne. Bottom: Picture of aq. solutions of trientine alkyne, $CuSO_4$ and corresponding mixtures. **p**, Fluorescence microscopy of labelled trientine alkyne in aMDM. For **b, c, l – n, p** scale bar, 10 µm. For **c** two-sided unpaired t-test, representative of $n = 3$ donors. Mean ± s.d. For **d, f, g, k, l – n**, two-sided Mann-Whitney test. For **h – j** Kruskal-Wallis test with Dunn's post-test. Box plots: boxes represent interquartile range and median, and whiskers indicate the minimum and maximum values. Each colored dot represents a distinct donor for a given panel.

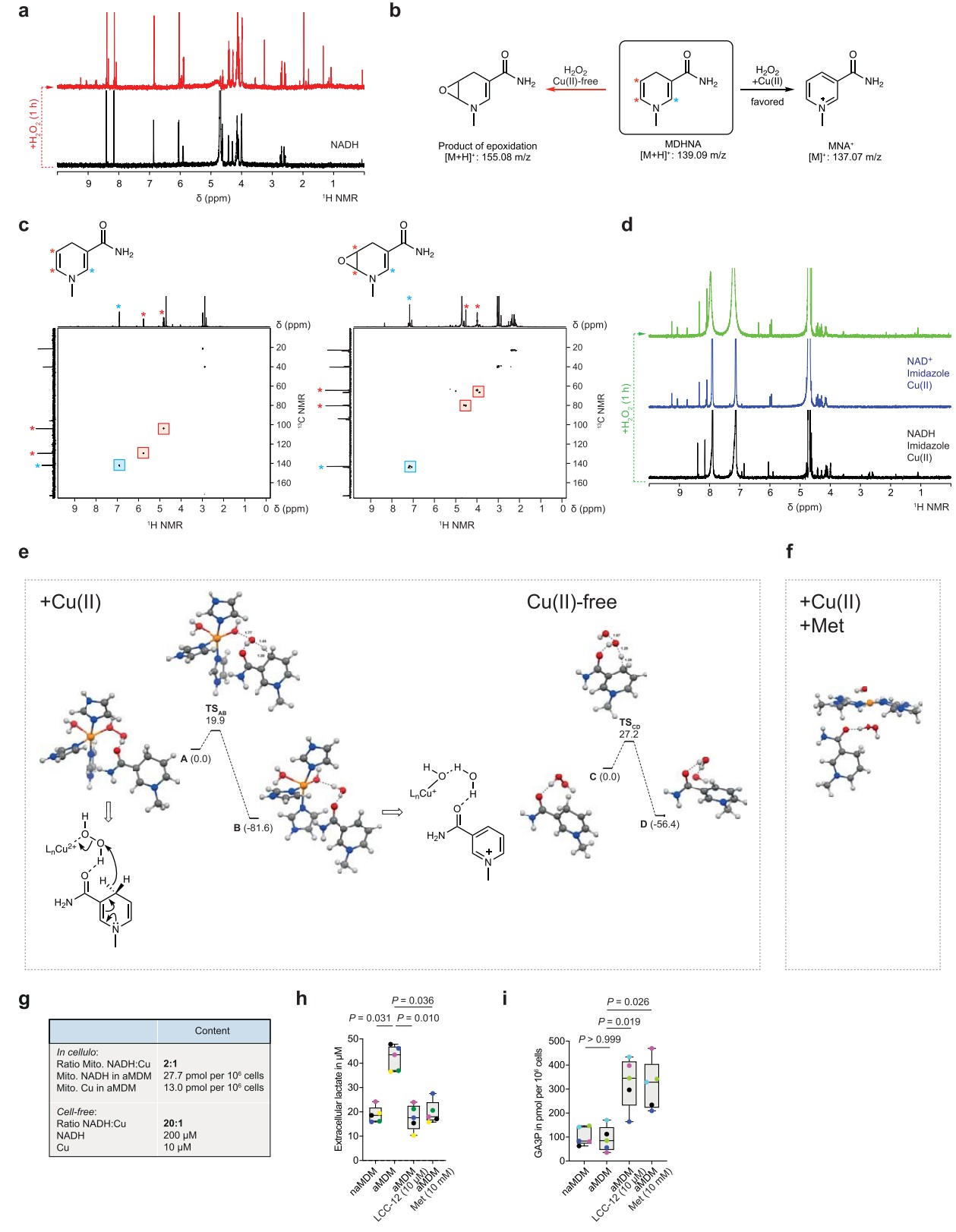

**Extended Data Fig. 6** | See next page for caption.

**Extended Data Fig. 6 | Copper(II) regulates NAD(H) redox cycling. a**, $^1$H NMR spectra of NADH (black) and the reaction products of NADH with $H_2O_2$ after 1 h at 37 °C, spectra recorded at 298 K in $D_2O$ (red). **b**, Reaction of MDHNA with $H_2O_2$ under copper(II)-catalyzed and copper-free conditions to afford either MNA$^+$ or a product of epoxidation, respectively. The mass of molecular ions detected by mass spectrometry are indicated. **c**, Heteronuclear single quantum coherence (HSQC) NMR spectra of MDHNA and its product of epoxidation. Red stars mark $^1$H and $^{13}$C NMR signals of the most reactive double bond towards $H_2O_2$ and that of the corresponding epoxide product. Blue stars mark $^1$H and $^{13}$C NMR signals of the least reactive double bond towards $H_2O_2$. Blue boxes show $^1$H-$^{13}$C HSQC correlations of the least reactive double bond. Red boxes show $^1$H-$^{13}$C HSQC correlations of the most reactive double bond and corresponding epoxide. **d**, $^1$H NMR spectra of NADH, imidazole and copper(II) (black), NAD$^+$ in the presence of imidazole and copper(II) (blue), the reaction product of NADH with $H_2O_2$ in the presence of imidazole and copper(II) after 1 h at 25 °C, spectra recorded at 298 K in buffered $D_2O$ (pD 8.4) (green). **e**, Free energy profile ($\Delta G_{298}$, kcal/mol) of the [(Imidazole)$_3$Cu(H$_2$O)](II)-mediated H-transfer reaction from MDHNA to $H_2O_2$. Selected distances in Å. Free energy profile ($\Delta G_{298}$, kcal/mol) of the copper-free H-transfer reaction from MDHNA to $H_2O_2$. Selected distances in Å. **f**, Optimized Cu(Met)$_2$, MDHNA, $H_2O_2$ and $H_2O$. **g**, Concentrations of copper and NADH in cells, and in the cell-free system used in Fig. 3f. **h**, Quantification of extracellular lactate produced by MDM treated with LCC-12 or Met ($n$ = 5 donors). **i**, Quantification of glyceraldehyde 3-phosphate (GA3P) in MDM treated with LCC-12 or Met ($n$ = 5 donors). For **h** and **i** Kruskal-Wallis test with Dunn's post-test. Box plots: boxes represent interquartile range and median, and whiskers indicate the minimum and maximum values. Each colored dot represents a distinct donor for a given panel.

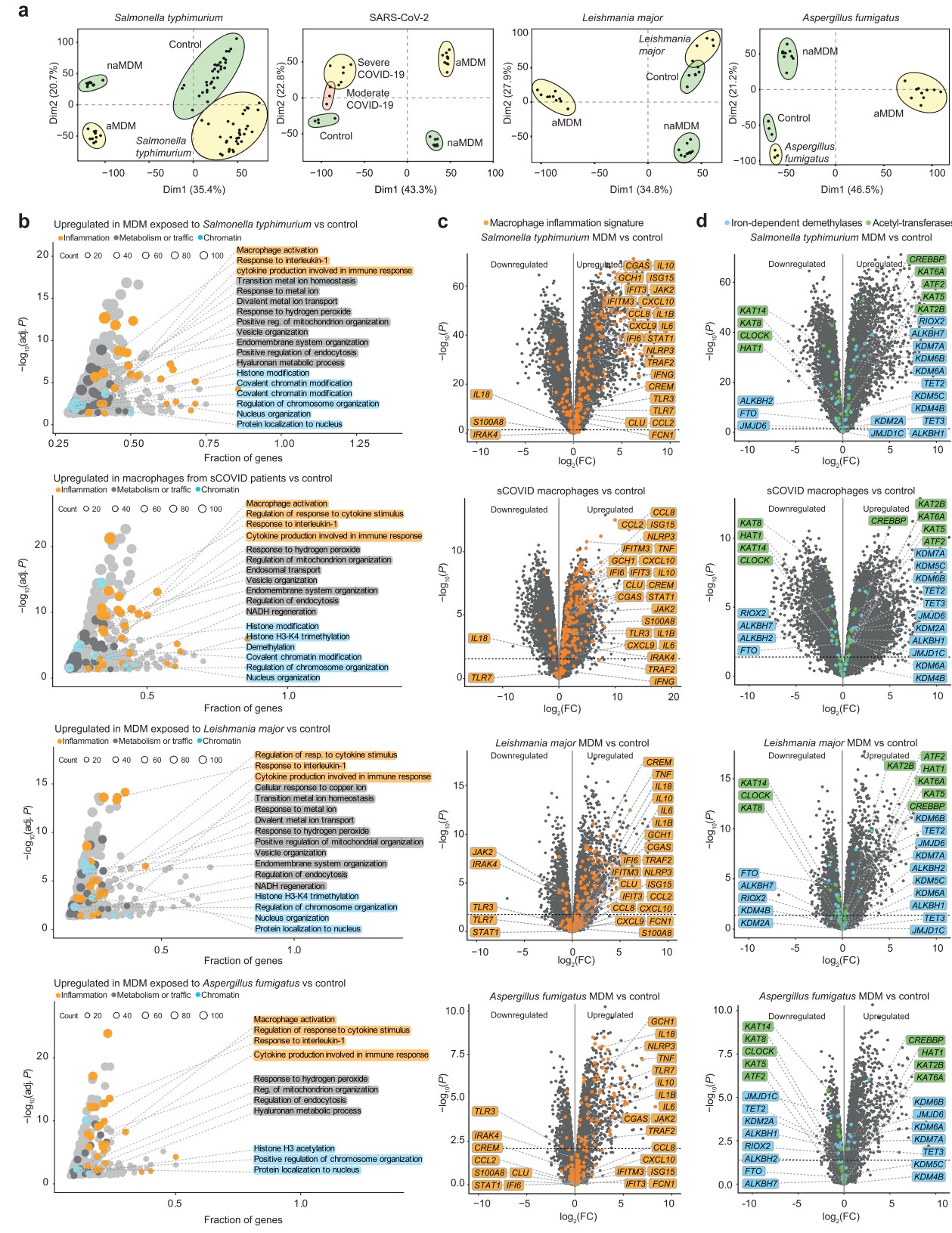

**Extended Data Fig. 7 |** See next page for caption.

**Extended Data Fig. 7 | Comparative analysis of transcriptomes of inflammatory macrophages. a**, Principal Component Analysis (PCA) of RNA-seq comparing naMDM ($n$ = 10 donors) and aMDM ($n$ = 10 donors) with MDM exposed to *Salmonella typhimurium* ($n$ = 32 donors) vs control ($n$ = 32 donors), macrophages from bronchoalveolar fluids of moderate ($n$ = 3 donors) and severe COVID-19 individuals (sCOVID, $n$ = 6 donors) vs control ($n$ = 4 donors), MDM exposed to *Leishmania major* ($n$ = 5 donors) vs control ($n$ = 6 donors) and MDM exposed to *Aspergillus fumigatus* ($n$ = 3 donors) vs control ($n$ = 3 donors). **b**, GO term analyses of upregulated genes in MDM exposed to *Salmonella typhimurium* ($n$ = 32 donors) vs control ($n$ = 32 donors), sCOVID ($n$ = 6 donors) vs control ($n$ = 4 donors), *Leishmania major* ($n$ = 5 donors) vs control ($n$ = 6 donors) and *Aspergillus fumigatus* ($n$ = 3 donors) vs control ($n$ = 3 donors). **c**, RNA-seq analyses of gene expression in MDM exposed to *Salmonella typhimurium* ($n$ = 32 donors) vs control ($n$ = 32 donors), sCOVID ($n$ = 6 donors) vs control ($n$ = 4 donors), *Leishmania major* ($n$ = 5 donors) vs control ($n$ = 6 donors) and *Aspergillus fumigatus* ($n$ = 3 donors) vs control ($n$ = 3 donors). Inflammatory signature genes are highlighted. Dashed lines, adjusted *P* values = 0.05. **d**, RNA-seq analyses of gene expression in MDM exposed to *Salmonella typhimurium* ($n$ = 32 donors) vs control ($n$ = 32 donors), sCOVID ($n$ = 6 donors) vs control ($n$ = 4 donors), *Leishmania major* ($n$ = 5 donors) vs control ($n$ = 6 donors) and *Aspergillus fumigatus* ($n$ = 3 donors) vs control ($n$ = 3 donors). Genes encoding iron-dependent demethylases and acetyl-transferases are highlighted. Dashed lines, adjusted *P* values = 0.05. For **b** – **d** differential gene expression was assessed with the limma/voom framework. GO enrichment was assessed with the enrichGO method from clusterProfiler. *P* values were corrected for multiple testing with the Benjamini-Hochberg procedure.

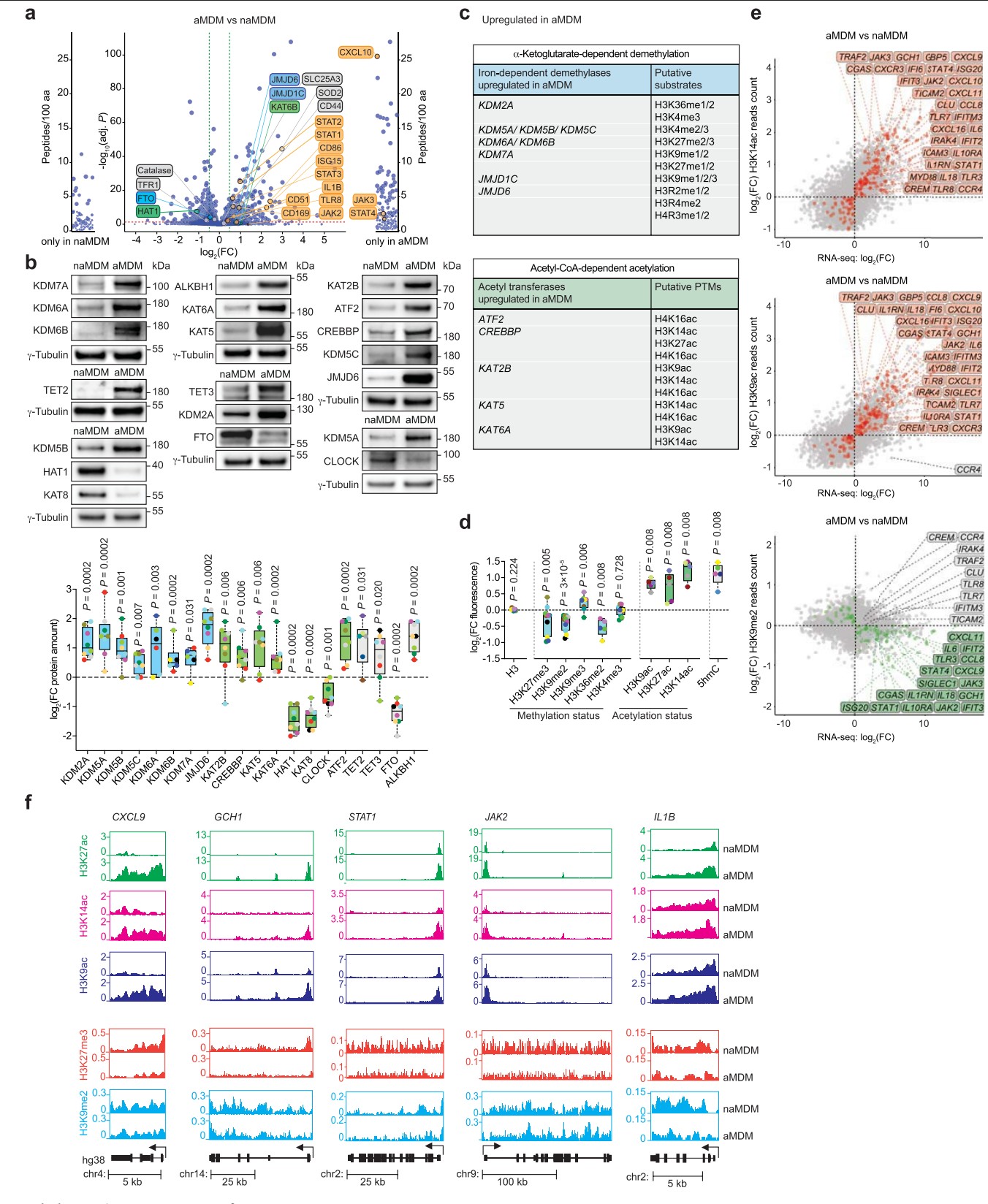

**Extended Data Fig. 8** | See next page for caption.

**Extended Data Fig. 8 | Mitochondrial copper(II) regulates epigenetic states and transcriptional programs of inflammatory macrophages.** **a**, Quantitative mass-spectrometry-based proteomics of MDM ($n$ = 8 donors). **b**, Representative western blots (top) of epigenetic modifiers identified by RNA-seq in aMDM and corresponding quantifications (bottom) ($n$ = 6–8 donors). **c**, Genes encoding iron-dependent demethylases and acetyl-transferases found to be upregulated in aMDM are listed together with putative substrates and post-translational modifications (PTMs) products. **d**, Fluorescence microscopy quantifications of histone H3 methyl and acetyl marks in MDM. Quantifications represent aMDM normalized against naMDM. At least 50 cells were quantified per donor per condition ($n$ = 5–11 donors). **e**, Scatter plot correlation of a representative donor of ChIP-seq reads count of histone marks in genes against RNA-seq of gene transcripts in MDM ($n$ = 10 donors). **f**, ChIP-seq tracks of selected genes involved in inflammation in MDM. For **b** and **d**, two-sided Mann-Whitney test. Box plots: boxes represent interquartile range and median, and whiskers indicate the minimum and maximum values. Each colored dot represents a distinct donor for a given panel.

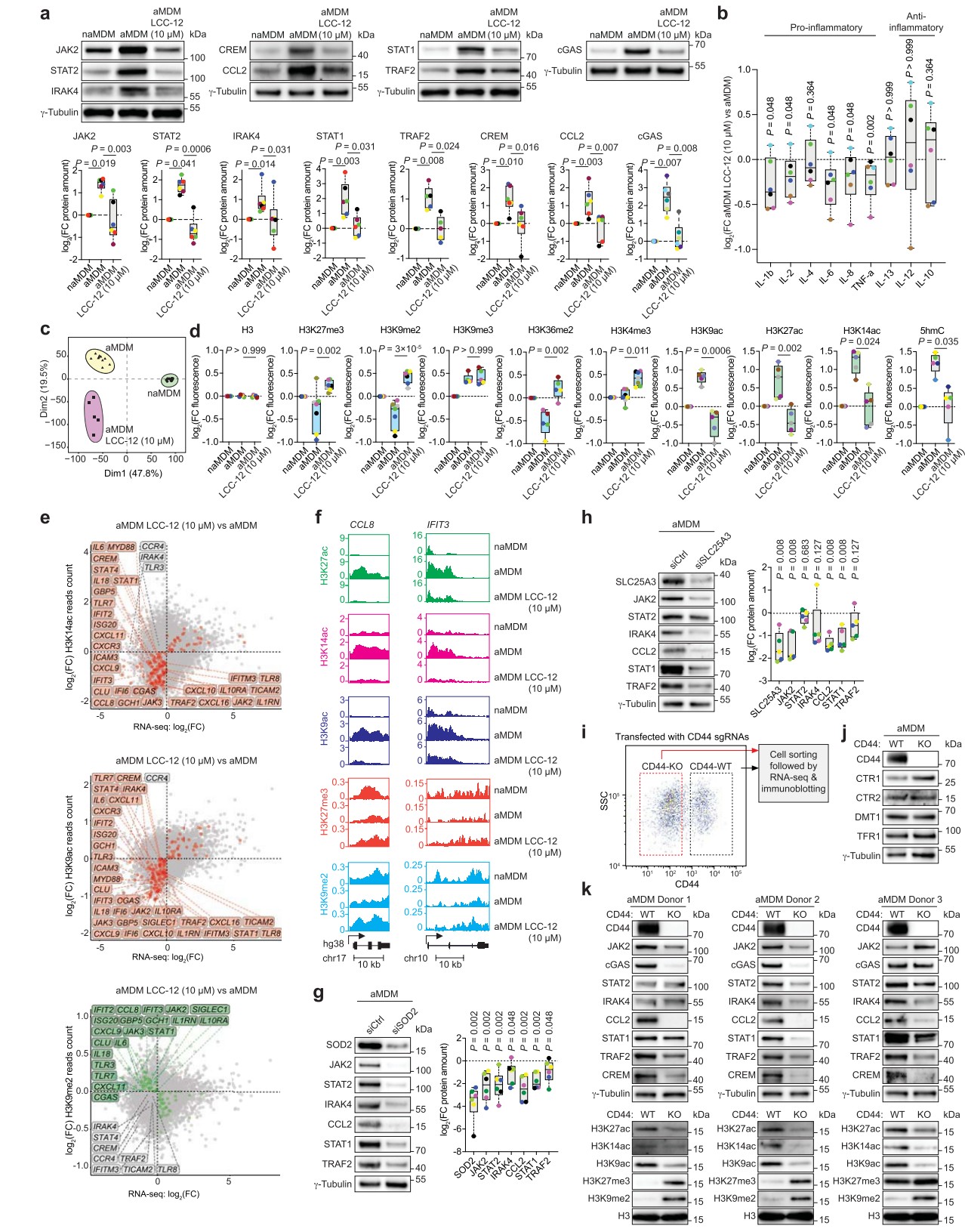

**Extended Data Fig. 9** | See next page for caption.

**Extended Data Fig. 9 | Mitochondrial copper(II) regulates epigenetic states and transcriptional programs of inflammatory macrophages.**
**a**, Representative western blots (top) of proteins involved in inflammation in MDM treated with LCC-12 and corresponding quantifications (bottom) ($n = 8$ donors). **b**, Immunoassay of cytokines secreted by MDM treated with LCC-12 during activation ($n = 6$ donors). **c**, PCA of RNA-seq comparing naMDM ($n = 10$ donors), aMDM ($n = 10$ donors) and MDM treated with LCC-12 during activation ($n = 5$ donors). **d**, Fluorescence microscopy quantifications of histone H3 methyl and acetyl marks in MDM treated with LCC-12. Quantifications were normalized against naMDM. At least 50 cells were quantified per donor per condition ($n = 5$–7 donors). **e**, Scatter plot correlation of a representative donor of ChIP-seq reads count of histone marks in genes against RNA-seq of gene transcripts in aMDM ($n = 10$ donors) and MDM treated with LCC-12 during activation ($n = 5$ donors). **f**, ChIP-seq tracks of selected genes involved in inflammation in MDM. **g**, Western blots of proteins involved in inflammation in aMDM under SOD2 knockdown conditions. **h**, Western blots of proteins involved in inflammation in aMDM under SLC25A3 knockdown conditions. **i**, Flow cytometry of wild-type (WT) and CD44 knockout (KO) aMDM. Gating strategy see Supplementary Information. **j**, Western blots of metal transporter proteins in WT and CD44-KO aMDM in $n = 1$ donor. **k**, Western blots of proteins involved in inflammation and histone marks in WT aMDM and CD44-KO aMDM for $n = 3$ donors. H3 is a sample processing control. For **a** and **d** Kruskal-Wallis test with Dunn's post-test. For **b**, **g** and **h** two-sided Mann-Whitney test. Box plots: boxes represent interquartile range and median, and whiskers indicate the minimum and maximum values. Each colored dot represents a distinct donor for a given panel.

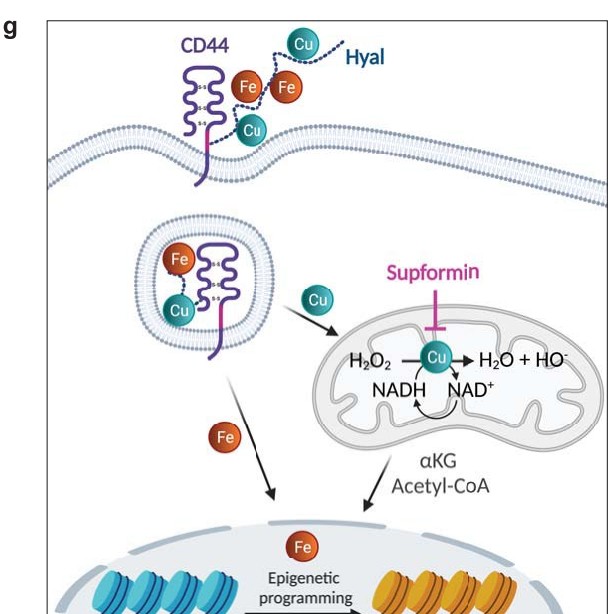

**a** LPS-induced sepsis model

**b** CLP-induced sepsis model

**c** SARS-CoV-2-induced acute inflammation

**d**

**e** Downregulated in SARS-CoV-2 + LCC-12 vs SARS-CoV-2

Positive regulation of cytokine production
Cytokine-mediated signaling pathway
Cytokine production involved in immune response
Reg. of cytokine prod. involved in immune response
Cytokine secretion
Regulation of cytokine secretion
Response to interleukin-1
Regulation of response to cytokine stimulus
Cytokine production involved in inflammatory response

Covalent chromatin modification
Histone modification
Regulation of chromatin organization
Chromatin remodeling
Regulation of gene expression, epigenetic

**f** Inflammation signature in lung tissue
SARS-CoV-2 + LCC-12 vs SARS-CoV-2

**g**

**Extended Data Fig. 10** | See next page for caption.

**Extended Data Fig. 10 | Pharmacological inactivation of mitochondrial copper(II) attenuates inflammation in vivo. a**, Western blots of copper-signalling effectors in SPMs from mice treated with LPS. Macrophages of several mice were pooled (4–7 mice per condition). **b**, Western blots of copper-signalling effectors in SPMs from mice subjected to CLP. Macrophages of several mice were pooled (7–8 mice per condition). H3 is a sample processing control. **c**, Western blots of copper-signalling effectors in AMs from K18-hACE2 mice infected with SARS-CoV-2. Macrophages of several mice were pooled (10 mice per condition). H3 is a sample processing control. **d**, Average body temperature of mice treated as indicated ($n$ = 6–9 mice per group). **e**, GO term analysis of downregulated genes in lung tissues of SARS-CoV-2 infected K18-hACE2 mice treated with LCC-12 (0.5 mg/kg). **f**, RNA-seq analysis of gene expression in lung tissues of SARS-CoV-2-infected K18-hACE2 mice treated with LCC-12 (0.5 mg/kg) ($n$ = 8 mice per group). Inflammatory signature genes highlighted. Dashed lines, adjusted $P$ value = 0.05. **g**, Illustration of copper-signalling. Cell plasticity involves upregulation of the cell surface marker CD44, which mediates endocytosis of metal-bound hyaluronates. In the presence of copper(II), NADH reacts with $H_2O_2$ to replenish $NAD^+$ in mitochondria, an enzyme cofactor involved in the biosynthesis of αKG and acetyl-CoA. These co-substrates of iron-dependent demethylases and acetyl-transferases are required for epigenetic and transcriptional programming of inflammation and the regulation of cell plasticity. Pharmacological inactivation of mitochondrial copper(II) blocks NAD(H) redox cycling, leading to distinct epigenetic states and transcriptional profiles. Targeting copper(II) interferes with cell plasticity in immune and cancer cells. For **a** – **c** gating strategy of SPMs and AMs see Methods and Supplementary Information. For **d** 2-way ANOVA. Mean values ± s.e.m. For **e** and **f** differential gene expression was assessed with the limma/voom framework. GO enrichment was assessed with the enrichGO method from clusterProfiler. $P$-values were corrected for multiple testing with the Benjamini-Hochberg procedure.

# Reporting Summary

## Statistics

For all statistical analyses, confirm that the following items are present in the figure legend, table legend, main text, or Methods section.

| n/a | Confirmed | |
|---|---|---|
| ☐ | ☒ | The exact sample size (*n*) for each experimental group/condition, given as a discrete number and unit of measurement |
| ☐ | ☒ | A statement on whether measurements were taken from distinct samples or whether the same sample was measured repeatedly |
| ☐ | ☒ | The statistical test(s) used AND whether they are one- or two-sided<br>*Only common tests should be described solely by name; describe more complex techniques in the Methods section.* |
| ☐ | ☒ | A description of all covariates tested |
| ☐ | ☒ | A description of any assumptions or corrections, such as tests of normality and adjustment for multiple comparisons |
| ☐ | ☒ | A full description of the statistical parameters including central tendency (e.g. means) or other basic estimates (e.g. regression coefficient) AND variation (e.g. standard deviation) or associated estimates of uncertainty (e.g. confidence intervals) |
| ☐ | ☒ | For null hypothesis testing, the test statistic (e.g. *F*, *t*, *r*) with confidence intervals, effect sizes, degrees of freedom and *P* value noted<br>*Give P values as exact values whenever suitable.* |
| ☒ | ☐ | For Bayesian analysis, information on the choice of priors and Markov chain Monte Carlo settings |
| ☒ | ☐ | For hierarchical and complex designs, identification of the appropriate level for tests and full reporting of outcomes |
| ☐ | ☒ | Estimates of effect sizes (e.g. Cohen's *d*, Pearson's *r*), indicating how they were calculated |

*Our web collection on statistics for biologists contains articles on many of the points above.*

## Software and code

Policy information about availability of computer code

| Data collection | No software was used |
|---|---|
| Data analysis | Code availability:<br>Analysis scripts for RNA-seq and ChIP-seq data are available at https://github.com/bioinfo-pf-curie/MDMmetals.<br><br>Softwares:<br>Flow cytometry<br>*FlowJo software v 10.8.2<br><br>Imaging<br>*Prism 8.2.0<br><br>Quantitative metabolomics<br>*Agilent Mass Hunter quantitative software B.07.01<br>*Agilent Mass Hunter quantitative software B.10.1<br><br>Quantitative proteomics<br>*Proteome Discoverer (version 2.4), Thermo Scientific<br>*myProMS (v3.9.3) (https://github.com/bioinfo-pf-curie/myproms), Poullet P et al., 2007<br><br>RNA-seq |

*Raw sequencing reads were first checked for quality with Fastqc (0.11.8) and trimmed for adapter sequences with the trimGalore (0.6.2) software
*Trimmed reads were then aligned on the human hg38 reference genome using the STAR mapper (2.6.1b), up to the generation of a raw count table per gene (GENCODE annotation v29)
*The bioinformatics pipelines used for these tasks are available online (rawqc v2.1.0: https://github.com/bioinfo-pf-curie/raw-qc, RNA-seq v3.1.4: https://github.com/bioinfo-pf-curie/RNA-seq)
*Counts were normalized using TMM normalization from edgeR (v 3.30.3)
*Differential expression was assessed with the limma voom framework (v 3.44.3)
*Enrichment analysis from differentially expressed genes has been performed using the enrichGO function from clusterProfiler package v3.16.1.

ChIP-seq
ChIP-seq data processing and quality controls have been performed with the Institut Curie ChIP-seq Nextflow pipeline (1.0.6) available at https://github.com/bioinfo-pf-curie/ChIP-seq

Illustrations
*FIJI 2.0.0-rc-69/1.52n
*Prism 8.2.0
*Adobe Illustrator 26.0.2
*biorender.com

For manuscripts utilizing custom algorithms or software that are central to the research but not yet described in published literature, software must be made available to editors and reviewers. We strongly encourage code deposition in a community repository (e.g. GitHub). See the Nature Portfolio guidelines for submitting code & software for further information.

# Data

Policy information about availability of data

All manuscripts must include a data availability statement. This statement should provide the following information, where applicable:
- Accession codes, unique identifiers, or web links for publicly available datasets
- A description of any restrictions on data availability
- For clinical datasets or third party data, please ensure that the statement adheres to our policy

Data availability:
RNA-seq and ChIP-seq data are available on the National Center for Biotechnology Information website with accession reference GSE160864 (go to: https://www.ncbi.nlm.nih.gov/geo/query/acc.cgi?acc=GSE160864). The mass spectrometry proteomics raw data have been deposited to the ProteomeXchange Consortium via the PRIDE partner repository with the dataset identifier PXD038612. The donor number corresponds to the order of blood collection.

Publicly available datasets:
* mouse mm10 reference genome
* human hg38 reference genome

# Human research participants

Policy information about studies involving human research participants and Sex and Gender in Research.

| | |
|---|---|
| Reporting on sex and gender | Peripheral blood samples were collected from 128 anonymous donors (Etablissement Français du Sang, EFS). |
| Population characteristics | Not available for anonymous donors.<br><br>According to the 2016 activity report of "Etablissement Français du Sang", half of the donors are under 40 years old, and consist of 52% women and 48% men. |
| Recruitment | It is not applicable. The blood samples are not selected but donated on a random basis according to availability. |
| Ethics oversight | The use of EFS blood samples from anonymous donors was approved by the Institut National de la Santé et de la Recherche Médicale committee. Written consent was obtained from all the donors. |

Note that full information on the approval of the study protocol must also be provided in the manuscript.

# Field-specific reporting

Please select the one below that is the best fit for your research. If you are not sure, read the appropriate sections before making your selection.

☒ Life sciences          ☐ Behavioural & social sciences          ☐ Ecological, evolutionary & environmental sciences

For a reference copy of the document with all sections, see nature.com/documents/nr-reporting-summary-flat.pdf

# Life sciences study design

All studies must disclose on these points even when the disclosure is negative.

| | |
|---|---|
| Sample size | No prior sample size calculation was performed.<br>A minimum of three independent experiments was done to be able to perform meaningful statistical analyses. |
| Data exclusions | No data were excluded from the analyses. |
| Replication | All attemps at replication were successful as stated in the reporting summary. |
| Randomization | This applies only to experiments involving mice with experimental groups. Consequently, randomization was made for these experiments. |
| Blinding | Data were blinded for ICP-MS data collection. For other data collection, blinding was not performed. |

# Reporting for specific materials, systems and methods

We require information from authors about some types of materials, experimental systems and methods used in many studies. Here, indicate whether each material, system or method listed is relevant to your study. If you are not sure if a list item applies to your research, read the appropriate section before selecting a response.

## Materials & experimental systems

| n/a | Involved in the study |
|---|---|
| ☐ | ☒ Antibodies |
| ☐ | ☒ Eukaryotic cell lines |
| ☒ | ☐ Palaeontology and archaeology |
| ☐ | ☒ Animals and other organisms |
| ☒ | ☐ Clinical data |
| ☒ | ☐ Dual use research of concern |

## Methods

| n/a | Involved in the study |
|---|---|
| ☐ | ☒ ChIP-seq |
| ☐ | ☒ Flow cytometry |
| ☒ | ☐ MRI-based neuroimaging |

## Antibodies

| | |
|---|---|
| Antibodies used | WB: western blot, FC: flow cytometry, FM: fluorescence microscopy, NS: NanoSIMS, ChIP: ChIP-seq. Hu: used for human samples. Ms: used for mouse samples. Dilutions are indicated. Any antibody validation by manufacturers is indicated and can be found on the manufacturers' websites. Our antibody validation by knockdown (kd) and/or knockout (ko) strategies as described in the manuscript for relevant antibodies is indicated.<br><br>Primary: ALKBH1 (Abcam, ab195376, clone EPR19215, Lot GR262105-2, WB 1:1000, Hu, Ms, ko validated by manufacturer), AMP-activated protein kinase subunit alpha (AMPKalpha, Cell Signaling, 2532S, Lot 21, WB 1:1000, Hu), P-AMPKalpha phosphorylated on Thr172 (P-AMPKalpha, Cell Signaling, 2535S, Lot 27, WB 1:1000, Hu), ATF2 (Abcam, ab32160, clone E243, Lot GR3430555-1, WB 1:1000, Hu), ATF2 (Proteintech, 14834-1-AP, WB 1:1000, Ms, kd/ko validated by manufacturer), ATP7A (Santa Cruz Biotechnology, sc-376467, clone D-9, Lot J2821, WB 1:200, Hu), ATP7A (Novus Biologicals, NBP2-59376, clone S60-4, WB 1:1000, Ms), ATP7B (Santa Cruz Biotechnology, sc-373964, clone A-11, Lot I2719, WB 1:200, Hu, Ms), Catalase (Cell Signaling, 12980T, clone D4P7B, Lot 3, WB 1:1000, Hu), CCL2/Mcp1 (Proteintech, 66272-1-Ig, clone 1B9F7, WB 1:1000, Hu), CD11b-Pacific Blue (BioLegend, 101224, clone M1/70, Lot B350151, B323654 and B323653, FC 1:800, Ms), CD14-Krome Orange (Beckman Coulter, B01175, clone RMO/52, Lot 200040, FC 1:100, Hu), CD16-Pacific Blue (Beckman Coulter, B36292, clone 3G8, Lot 200029, FC 1:100, Hu), CD3 (BioLegend, 317326, clone OKT3, Lot B372352, T cell activation, 2.5 µg/mL, Hu), CD25-BV711 (BioLegend, 302636, clone BC96, Lot B281779, FC 1:100, Hu), CD28 (BioLegend, 302934, clone CD28.2, Lot 374639, T cell activation, 2.5 µg/mL, Hu), CD40-APC (BioLegend, 124612, clone 3/23, Lot B309981, FC 1:200, Ms), CD40-BV510 (BioLegend, 334329, clone 5C3, Lot 312131, FC 1:100, Hu), CD44 (Abcam, ab189524, Lot GR320797-13, GR3314218-16, clone EPR18668, WB 1:30000, Hu, Ms, ko/kd validated by us and manufacturer), CD44 (ThermoFisher Scientific, 701406, clone 19H8L4, Lot 1976318, FM 1:400, Hu), CD44-AF647 (Novus Biologicals, NB500-481AF647, clone MEM-263, Lot 118753, FC 1:100, Hu), CD44-AF647 (BioLegend, 103018, clone IM7, Lot B317762, FC 1:200, Ms), CD45-BV510 (BioLegend, 103138, clone 30-F11, Lot B362964, B322199, B333193, FC 1:200, Ms), CD64-FITC (BioLegend, 399505, clone S18012C, Lot 308498, FC 1:100, Hu), CD66b-Pe/Cy7 (BioLegend, 305115, clone G10F5, Lot 283925, FC 1:100, Hu), CD69-PerCP (BioLegend, 310928, clone FN50, Lot B290414, FC 1:100, Hu), CD80-AF700 (BD Biosciences, 561133, clone 307.4, Lot 1060235, FC 1:100, Hu), CD83-PE (BioLegend, 305307, clone HB15e, Lot B303073, FC 1:100, Hu), CD86-PE (BioLegend, 105007, clone GL-1, Lot B318893, FC 1:200, Ms), CD86-PE/Cy7 (BD Biosciences, 561128, clone 2331 (FUN-1), Lot 1309531, FC 1:33, Hu), CD109 (Santa Cruz Biotechnology, sc-271085, clone C-9, Lot E1018, WB 1:200, Hu), CD109 (Biotechne, AF7717-SP, WB 1:1000, Ms), CD163-PE (BD Biosciences, 556018, clone GHI/61, Lot 9143793, FC 1:100, Hu), CD170 (Siglec-F)-PEeFluor 610 (eBioscience, 61-1702-80, clone 1RNM44N, Lot 2472220 and 2152352, FC 1:200, Ms), cGAS (Cell Signaling, 15102, clone D1D3G, Lot 4, WB 1:1000, Hu), cGAS (Cell Signaling, 31659, clone D3O8O, Lot 3, WB 1:1000, Ms), CLOCK (Proteintech, 18094-1-AP, WB 1:1000, Hu, kd/ko validated by manufacturer), CLOCK (Abcam, ab3517, WB 1:1000, Ms), Copper transporter 1 (Ctr1, Abcam, ab129067, clone EPR7936, Lot GR3414582-4 and GR81444-2, WB 1:1000, Hu, kd validated by us), Copper transporter 2 (Ctr2, Novus Biologicals, NBP1-05199SS, WB 1:1000, Hu), Copper transporter 2 (Ctr2, Novus Biologicals, NBP1-85512, Lot R05901, FM 1:400, Hu), Copper transporter 2 (Biorybt, orb182668, Lot BR2373, WB 1:1000, Hu) COX IV (Abcam, ab16056, Lot GR320655-1, FM 1:400, Hu), CREM (Proteintech, 12131-1-AP, WB 1:1000, Hu), |

Cytochrome c (Cyt c, Cell Signaling, 12963S, clone 6H2.B4, Lot 1 and 2, FM 1:400, NS 1:400, Hu), Divalent Metal Transporter 1 (DMT1, Abcam, ab55735, clone 4C6, Lot GR3243346-1, WB 1:1000, Hu, kd validated by us), Drosophila spike-in antibody (Active Motif, 61686, Lot 23521010, ChIP 50 ng per condition), E-cadherin (Cell Signaling, 3195, clone 24E10, Lot 15, WB 1:1000, Hu), E-cadherin (BD Biosciences, 610181, clone 36, Lot 7187865, WB 1:1000, Ms), F4/80-BV605 (BioLegend, 123133, clone BM8, Lot B362524, B309659, B331465 and B339746, FC 1:100, Ms), F4/80-PE (TONBO, TNB50-4801-U100, clone BM8.1, Lot C4801060619503, FC 1:100, Ms), Fibronectin (Sigma-Aldrich, F0791, clone IST-3, Lot 026M4781V, WB 1:1000, Hu, Ms), FTO (Proteintech, 27226-1-AP, WB 1:1000, Hu, Ms, kd/ko validated by manufacturer), H3 (Cell Signaling, 9715S, Lot 23, FM, WB 1:1000, Hu, Ms), H3K4me3 (Diagenode, C15410003-50, Lot A8034D, FM 1:400, Hu, dot blot validation by manufacturer), H3K9ac (Cell Signaling, 9649S, clone C5B11, Lot 13, FM, WB, ChIP 6μL per 1×106 cells, Hu, Ms, validated with SimpleChIP Enzymatic Chromatin IP by manufacturer), H3K9me2 (Cell Signaling, 4658S, clone D84B4, Lot 10, FM 1:400, WB 1:1000, ChIP 6μL per 1×106 cells, Hu, Ms, validated with SimpleChIP Enzymatic Chromatin IP by manufacturer), H3K9me3 (Cell Signaling, 13969S, clone D4W1U, Lot 3, FM 1:400, Hu, validated with SimpleChIP Enzymatic Chromatin IP by manufacturer), H3K14ac (Cell Signaling, 7627S, clone D4B90, Lot 6, FM, WB 1:1000, ChIP 6μL per 1×106 cells, Hu, Ms, validated with SimpleChIP Enzymatic Chromatin IP by manufacturer), H3K27ac (Cell Signaling, 8173S, clone D5E4, Lot 8, FM, WB 1:1000, ChIP 6μL per 1×106 cells, Hu, Ms, validated with SimpleChIP Enzymatic Chromatin IP by manufacturer), H3K27me3 (Cell Signaling, 9733S, clone C36B11, Lot 19, FM 1:400, WB 1:1000, ChIP 6μL per 1×106 cells, Hu, Ms, validated with SimpleChIP Enzymatic Chromatin IP by manufacturer), H3K36me2 (Abcam, ab9049, Lot GR3258133-1, FM 1:400, Hu), Hyaluronan synthase 1 (HAS1, Novus Biologicals, NBP1-51635, clone 3E10, Lot 141031, WB 1:1000, Hu), HAS1 (Sigma-Aldrich, SAB4300848, Lot 492637613, WB 1:1000, Ms), Hyaluronan synthase 2 (HAS2, Abcam, ab140671, clone 4E7, Lot GR3212928-2, WB 1:1000, Hu), HAS2 (Santa Cruz Biotechnology, sc-514737, clone A-7, WB 1:200, Ms), Hyaluronan synthase 3 (HAS3, Abcam, ab154104, Lot GR113715-12, WB 1:1000, Hu), HAS3 (Proteintech, 15609-1-AP, WB 1:1000, Ms, ko/kd validated by us and manufacturer), HAT1 (Proteintech, 11432-1-AP, WB 1:1000, Hu, Ms, ko/kd validated by us and manufacturer), 5-Hydroxymethylcytosine (5hmC, Active Motif, 39069, Lot 23720003, FM 1:400, Hu, dot blot validated by manufacturer), I-A/I-E-AF700 (BioLegend, 107622, clone M5/114.15.2, Lot B313251, FC 1:400, Ms), IRAK4 (Cell Signaling, 4363T, Lot 5, WB 1:1000, Hu, Ms), JAK2 (Cell Signaling, 3230T, clone D2E12, WB 1:1000, Lot 13, Hu, Ms), JMJD6 (Abcam, ab64575, Lot GR3441511-1, WB 1:1000, Hu, Ms), KAT2B/PCAF (Cell Signaling, 3378T, clone C14G9, Lot 2, WB 1:1000, Hu, Ms, validated with SimpleChIP Enzymatic Chromatin IP by manufacturer), KAT3A/CREBBP (Abcam, ab2832, Lot GR3360262-6, WB 1:1000, Hu, Ms), KAT5/Tip60 (Santa Cruz Biotechnology, sc-166323, clone C-7, Lot 166323, WB 1:200, Hu, Ms), KAT6A/MOZ (Santa Cruz Biotechnology, sc-293283, clone 4D8, Lot F0420, WB 1:200, Hu), KAT6A/MOZ (Invitrogen, PA5-103467, Lot XH3653004, WB 1:1000, Ms), KAT8/MOF (Proteintech, 13842-1-AP, kd/ko validated by manufacturer, WB 1:1000, Hu, Ms), KDM2A (Abcam, ab191387, clone EPR18602, Lot GR3330146-4, WB 1:1000, Hu, Ms, ko validated by manufacturer), KDM5A/Jarid1a (Cell Signaling, 3876S, clone D28B10, Lot 5, WB 1:1000, Hu, Ms), KDM5B/Jarid1b (Cell Signaling, 3273T, Lot 3, WB 1:1000, Hu), KDM5B/Jarid1b (Abcam, ab181089, clone EPR12794, WB 1:1000, Ms, ko validated by manufacturer), KDM5C/Jarid1c (Cell Signaling, 5361, clone D29B9, Lot 1, 5361, WB 1:1000, Hu, Ms), KDM6A/UTX (Cell Signaling, 33510S, clone D3Q1l, Lot 4, WB 1:1000, Hu, Ms), KDM6B (Abcam, ab169197, WB 1:1000, Hu), KDM6B/JMJD3 (Cell Signaling, 3457, WB 1:1000, Ms), KDM7A (Invitrogen, PA5-96987, Lot UI2838718, WB 1:1000, Hu, Ms), Ly6C-PerCP/Cy5.5 (BioLegend, 128012, clone HK1.4, Lot B363119 and B310463, FC 1:200, Ms), Ly6G-PE-Cy7 (BioLegend, 127618, clone 1A8, Lot B288785 and B351626, FC 1:200, Ms), Ly6G-AF647 (BioLegend, 127610, clone 1A8, Lot B2559839, FC 1:200, Ms), Lysosome-associated membrane protein 2 (LAMP2, Abcam, ab25631, clone H4B4, FM 1:400, Hu), NOS2-APC (eBioscience, 17-5920-82, clone CXNFT, Lot 2154045, FC 1:100, Ms), SLC25A3 (Santa Cruz Biotechnology, sc-376742, clone F-1, Lot H2313, WB 1:200, FM 1:100, Hu, kd validated by us), SLC25A3 (Abcam, ab89117, Lot 1015892-1, FM 1:400, Hu), SLC25A37 (MyBiosource, MBS9210193, clone ID: RB24153, Lot SA100524AR, WB 1:1000, FM 1:400, Hu, kd validated by us), Slug (Cell Signaling, 9585S, clone C19G7, Lot 6, WB 1:1000, Hu, Ms), STAT1 (Cell Signaling, 14994T, clone D1K9Y, Lot 8, WB 1:1000, Hu, Ms, validated with SimpleChIP Enzymatic Chromatin IP by manufacturer), STAT2 (Abcam, ab32367, clone Y141, Lot GR3294792-5, WB 1:1000, Hu, Ms, ko validated by manufacturer), Superoxide dismutase 2 (SOD2, Abcam, ab13534, Lot GR3345921-12 and GR33618-52, FM 1:400, WB 1:1000, Hu, Ms, kd validated by us), Tet methylcytosine dioxygenase 2 (TET2, Abcam, ab94580, Lot GR3243631-2, WB 1:1000, Hu, Ms), Tet methylcytosine dioxygenase 3 (TET3, Abcam, ab139311, Lot GR3314447-1, WB 1:1000, Hu, Ms), TRAF2 (Cell Signaling, 4724T, clone C192, Lot 2, WB 1:1000, Hu, Ms), Transferrin receptor 1 (TfR1, Invitrogen, 13-6800, clone H68.4, Lot VJ313549, WB 1:1000, Hu, kd validated by us), TfR1-APC-AF750 (Beckman Coulter, A89313, clone YDJ1.2.2, Lot 200060, FC 1:100, Hu), γ-Tubulin (Sigma-Aldrich, T5326, clone GTU-88, Source 0000128065, WB 1:1000, Hu, Ms, enhanced validation by manufacturer), Twist (Santa Cruz Biotechnology, sc-81417, clone Twist2C1a, Lot J2213, WB 1:200, Ms), Vimentin (Cell Signaling, 5741S, clone D21H3, Lot 8, WB 1:1000, Hu, Ms, ko validated by previous users according to manufacturer's website).

Secondary: Alexa-Fluor-488 anti-rabbit (Invitrogen, A-11070, Lot 2161039, FM 1:1000, Hu), Alexa-Fluor-594 anti-mouse (Invitrogen, A-11032, Lot 1826426, FM 1:1000, Hu), Alexa-Fluor-594 anti-rabbit (Invitrogen, A11072, Lot 1985650, FM 1:1000, Hu), Alexa-Fluor-647 anti-mouse (Invitrogen, A21237, Lot 1743738, FM 1:1000, Hu), Alexa-Fluor-647 anti-rabbit (Invitrogen, A21246, Lot 2418503, FM 1:1000, Hu), Donkey anti-Rabbit IgG-h+l HRP conjugated (Bethyl Laboratories, A120-108P, Lot 12 and 13), Goat anti-Mouse IgG h+l HRP conjugated (Bethyl Laboratories, A90-116P, Lot 41 and 44), 10 nM gold-nanoparticle-loaded anti-mouse (Abcam, ab27241, Lot GR274015-2, NS 1:200, Hu).

| | |
|---|---|
| Validation | Any antibody validation by manufacturers is indicated and can be found on the respective websites and is specified in the antibody section. Our antibody validation by knockdown (kd) and/or knockout (ko) strategies, as described in the manuscript, is indicated. |

# Eukaryotic cell lines

Policy information about cell lines and Sex and Gender in Research

| | |
|---|---|
| Cell line source(s) | Mouse pancreatic cancer cells FC1245 were a kind gift from Prof. D. Tuveson (Cold Spring Harbor)<br>Primary non-small cell lung circulating cancer cells (Celprogen, 36107-34CTC, Lot 219411, sex: female) |
| Authentication | FC1245 cells were a gift from the Tuveson laboratory. We did not perform additional characterization.<br><br>Primary non-small cell lung circulating cancer cells were obtained from Celprogen and are primary cells. We did not perform additional characterization. |

| Mycoplasma contamination | The cells were tested negative for mycoplasma contamination. |
| Commonly misidentified lines (See ICLAC register) | No commonly misidentified cell lines were used in this study. |

# Animals and other research organisms

Policy information about studies involving animals; ARRIVE guidelines recommended for reporting animal research, and Sex and Gender in Research

| Laboratory animals | Survival assessment using the LPS mouse model was performed on 8-week-old male BALB/c mice. Other LPS experiments were performed on 8-week-old male BALB/c mice and 5-week-old male SWISS mice and experiments involving the CLP model were performed on 9-week-old male BALB/c mice. For experiments involving SARS-CoV-2, 8-week-old male K18-human ACE2 expressing C57BL/6 mice were used.

Mice were housed in state-of-the-art animal care facilities. Mice in all animal facilities were housed in ventilated cages (temperature 22 °C +/- 2 °C, humidity 55% +/- 10%) with free access to water and food on a 12 h light/dark cycle. Male littermates were randomly assigned to experimental groups throughout. |
| Wild animals | The study did not involve wild animals. |
| Reporting on sex | Experiments were performed on male mice. |
| Field-collected samples | The study did not involve samples collected from the field. |
| Ethics oversight | Survival assessment using the LPS mouse model was conducted at Fidelta Ltd (now Selvita) according to 2010/63/EU and National legislation regulating the use of laboratory animals in scientific research and for other purposes (Official Gazette 55/13). An institutional committee on animal research ethics (CARE-Zg) oversaw that animal-related procedures were not compromising the animal welfare.

Other experiments involving LPS and CLP mouse models were performed in accordance with French laws concerning animal experimentation (#2021072216346511) and approved by the Institutional Animal Care and Use Committee of Université de Saint-Quentin-en-Yvelines (C2EA-47). Mice were housed in a state-of-the-art animal care facility (2CARE, prefectural number agreement: A78-322-3, France).

A SARS-CoV-2 mouse model was used within the biosafety level 3 facility of the Institut Pasteur de Lille, after validation of the protocols by the local committee for the evaluation of the biological risks and complied with current national and institutional regulations and ethical guidelines (Institut Pasteur de Lille/B59-350009). The experimental protocols using animals were approved by the institutional ethical committee 'Comité d'Ethique en Experimentation Animale (CEEA) 75, Nord-Pas-de-Calais'. The animal study was authorized by the 'Education, Research and Innovation Ministry' under registration number APAFIS#25517-2020052608325772v3.

A SARS-CoV-2 mouse model was used within the biosafety level 3 facility of the University of Toulouse. This work was overseen by an Institutional Committee on Animal Research Ethics (License APAFIS#27729-2020101616517580 v3, Minister of Research, France (CEEA-001)), to ensure that animal-related procedures were not compromising the animal welfare. |

Note that full information on the approval of the study protocol must also be provided in the manuscript.

# ChIP-seq

## Data deposition

☒ Confirm that both raw and final processed data have been deposited in a public database such as GEO.

☒ Confirm that you have deposited or provided access to graph files (e.g. BED files) for the called peaks.

| Data access links *May remain private before publication.* | Data are available on the National Center for Biotechnology Information website with accession reference GSE160864 (https://www.ncbi.nlm.nih.gov/geo/query/acc.cgi?acc=GSE160864; enter token wvqxgwgojxcdboz into the box). |
| Files in database submission | Raw sequencing data as well as bigwig tracks have been shared on GEO. |
| Genome browser session (e.g. UCSC) | No UCSC session available. |

## Methodology

| Replicates | All ChIP experiments have been performed using several donors. For each donor, we have 3 conditions: naMDM, aMDM and aMDM LCC-12. |
| Sequencing depth | The sequencing depth varies from a type of histone mark to another. For active histone marks, an average of 50M read pairs have been sequenced. While around 150M read pairs have been sequenced for repressive histone marks. |
| Antibodies | Drosophila spike-in antibody (Active Motif, 61686, Lot 23521010, ChIP 50 ng per condition), H3K9ac (Cell Signaling, 9649S, clone |

| Antibodies | C5B11, Lot 13, FM, WB, ChIP 6μL per 1×106 cells, Hu, Ms, validated with SimpleChIP Enzymatic Chromatin IP by manufacturer), H3K9me2 (Cell Signaling, 4658S, clone D84B4, Lot 10, FM 1:400, WB 1:1000, ChIP 6μL per 1×106 cells, Hu, Ms, validated with SimpleChIP Enzymatic Chromatin IP by manufacturer), H3K14ac (Cell Signaling, 7627S, clone D4B90, Lot 6, FM, WB 1:1000, ChIP 6μL per 1×106 cells, Hu, Ms, validated with SimpleChIP Enzymatic Chromatin IP by manufacturer), H3K27ac (Cell Signaling, 8173S, clone D5E4, Lot 8, FM, WB 1:1000, ChIP 6μL per 1×106 cells, Hu, Ms, validated with SimpleChIP Enzymatic Chromatin IP by manufacturer), H3K27me3 (Cell Signaling, 9733S, clone C36B11, Lot 19, FM 1:400, WB 1:1000, ChIP 6μL per 1×106 cells, Hu, Ms, validated with SimpleChIP Enzymatic Chromatin IP by manufacturer). |
|---|---|
| Peak calling parameters | No peak calling was in this study. As the goal was mainly to integrate histone modification and gene expression, the ChIP-seq density signal has been extracted around gene TSS (+-2kb) for active histone marks and along the gene body for repressive marks. |
| Data quality | All samples passed quality checks, with more than 80% of aligned reads on the genome and a significant enrichment (fingerprint). |
| Software | The bioinformatics pipeline used to process the ChIP-seq data is available online at https://github.com/bioinfo-pf-curie/ChIP-seq. |

# Flow Cytometry

## Plots

Confirm that:

☒ The axis labels state the marker and fluorochrome used (e.g. CD4-FITC).

☒ The axis scales are clearly visible. Include numbers along axes only for bottom left plot of group (a 'group' is an analysis of identical markers).

☒ All plots are contour plots with outliers or pseudocolor plots.

☒ A numerical value for number of cells or percentage (with statistics) is provided.

## Methodology

| Sample preparation | Flow cytometry on human immune cells: cells were washed with ice-cold 1× PBS, incubated with Fc block (Human TruStain FcX, BioLegend, 422302, 1:20) for 15 min, subsequently incubated with antibodies for 20 min at 4 °C in 1× PBS/ 0.5% bovine serum albumin (BSA) and then washed before analysis using a flow cytometer (BD LSRFortessa X-20). Cells were analyzed with the corresponding antibody panels.

Flow cytometry on primary non-small cell lung circulating cancer cells and FC1245 cells: cells were harvested by trypsinization using Trypsin/EDTA (GIBCO, TRYPGIB01), washed with 1× PBS and antibody incubation was performed for 20 mins at 4 °C in 1× PBS/10% FBS. Cells were washed and analyzed using a flow cytometer (BD Accurie C6).

Flow cytometry on LPS-induced severe inflammation mouse model: after euthanasia at 22 h post LPS challenge, the organs were perfused with PBS/EDTA (1 mL/g, 2mM, pH 7.4) and 10 mL of 1× PBS were injected in the peritoneum. The peritoneal liquid was then collected and centrifugated for 5 min at 1500 rpm. The pellet was resuspended in RPMI medium containing 2% fetal calf serum (Dutscher, S181H-100). The peritoneal liquid was then washed in 96-well plates at 2000 rpm for 2 min, and pellets were suspended in complete RPMI. For intracellular protein staining, samples were incubated for 2 h in brefeldin A (BFA, 5 ng/mL, Invitrogen, 00-4506-51) before surface staining with Fixable Viability Dye eFluor 780 (Invitrogen, 65-0865-14) followed by fluorochrome-conjugated antibodies (35 min at 4 °C). Samples incubated with BFA were fixed (Foxp3/ Transcription factor Fix/Perm 4X, Cell Signaling, 44931S) for 20 min at 4 °C and permeabilized (Flow Cytometry Perm Buffer 10X, TONBO, TNB-1213-L150) before intracellular staining. The antibodies used were as follows: CD11b-Pacific Blue (BioLegend, 101224), CD40-APC (BioLegend, 124612), CD45-BV510 (BioLegend, 103138), CD86-PE (BioLegend, 105007), CD170 (Siglec-F)-PEeFluor 610 (eBioscience, 61-1702-80), F4/80-BV605 (BioLegend, 123133), I-A/I-E-AF700 (BioLegend, 107622), Ly6C-PerCP/Cy5.5 (BioLegend, 128012) and Ly6G-PE/CY7 (BioLegend, 127618). For intracellular staining, NOS2-APC (eBioscience, 17-5920-82) was used. Small peritoneal macrophages (SPM) correspond to CD45+/IA-IE+/CD11b+/F4/80int/ SiglecF- cells. After washing with PBS or with Perm Buffer, data were acquired using an LSR Fortessa flow cytometer (BD Biosciences).

Flow cytometry on the CLP murine model: the sorting of SPM was done using the following antibodies: CD11b-Pacific Blue (BioLegend, 101224), F4/80-PE (TONBO, TNB50-4801-U100), Ly6C-PerCP/Cy5.5 (BioLegend, 128012) and Ly6G-AF647 (BioLegend, 127610). The sorted SPM corresponded to CD11b+/F4/80int/Ly6C-/Ly6G- cells. |
|---|---|

Flow cytometry on SARS-CoV-2 mouse model: after a terminal anesthesia (ketamine 100 mg/kg and xylazine 10 mg/kg IP), lung tissues were harvested and homogenized in gentleMACS C Tubes (Miltenyi Biotec) containing 2.5 mL of RPMI 1640 medium (Gibco) and collagenase (2.5 mg/mL, Roche) using a gentleMACS Dissociator (Miltenyi Biotec). Lung tissues were further dissociated for 30 min at 37 °C under shaking, passed through a 70 µm cell strainer, and proceeded for red blood cell lysis before staining for cell sorting. Cells were stained with Fixable Viability Dye eFluor 780 (Invitrogen, 65-0865-14) followed by fluorochrome-conjugated antibodies (35 min at 4 °C). The antibodies used were as follows: CD11b-Pacific Blue (BioLegend, 101224), CD45-BV510 (BioLegend, 103138), CD170 (Siglec-F)-PEeFluor 610 (eBioscience, 61-1702-80), F4/80-BV605 (BioLegend, 123133). Alveolar macrophages (AM) correspond to CD45+/CD11bint/F4/80+/SiglecF+ cells and were isolated on a BD FACSAria Fusion.

| | |
|---|---|
| Instrument | flow cytometers: BD LSRFortessa X-20,  BD FACSAria and BD Accuri C6 |
| Software | FlowJo software v 10.8.2 |
| Cell population abundance | For human cells, cell sorting allowed to obtain a satisfactory purity. |
| Gating strategy | Murine small peritoneal macrophages (SPM) correspond to CD45+/IA-IE+/CD11b+/F4/80int/SiglecF- cells.<br>Alveolar macrophages (AM) correspond to CD45+/CD11bint/F4/80+/SiglecF+ cells. The gating strategies are detailed in the Supplementary Information. |

☒ Tick this box to confirm that a figure exemplifying the gating strategy is provided in the Supplementary Information.

