## [Peer Review File · Nature]

Manuscript Title: Discovery of a druggable copper-signaling pathway that drives inflammation

Reviewer Comments & Author Rebuttals

Reviewer Reports on the Initial Version:

Referee expertise:

Referee #1: Macrophage epigenetics

Referee #2: copper metabolism/biology

Referee #3: macrophage metabolism

Referees' comments:

Referee #1 (Remarks to the Author):

In this well-written and novel paper the authors report that macrophage inflammation is controlled by a CD44 mediated pathway where CD44 acts as copper transporter and increased copper in the mitochondria results in NAD⁺ that promotes epigenetic changes that serve to promote inflammation. Further, they developed a drug that regulates copper uptake and decreases inflammation in macrophages in the setting of murine sepsis and covid. Although, the paper flows logically it requires several experiments to validate the conclusions - particularly related to the loose link between changes in metabolite co-factors, alterations of demethylases/HATs and alterations in inflammatory gene expression programs. Although there are observational links, additional experiments, particularly ChIP seq would add value in mechanistically examining the regulation of these enzymes by copper uptake. I have the following specific major comments:

1. For the experiments in Figs 1 and 2, I think a control is missing - the authors use activated (with LPS/IFN γ) and non-activated macrophages, but they fail to examine macrophages activated towards an anti-inflammatory or M2-like phenotype. In this regard, I do believe it is important to see macrophages treated with IL4/IL13 to determine if it is indeed the skewing towards inflammatory/M1-like macrophages that is involved in the CD44 uptake of copper or if any skewing of macrophages can induce this mechanism.
2. In later figures in the paper, LLC-12 is used in both a murine sepsis (two models) and covid setting - For Figure 1k, only LPS induced sepsis is used to examine CD44 mediated HA uptake. Since this paper is trying to link this mechanism to both sepsis and sars infection, it would be important to show the CD44/copper levels in all of these conditions.
3. In Figure 5, it would be important to also add a control for the RNAseq that examines conditions that alter the CD44/copper pathway and compare the RNAseq to activated/ non-activated macrophages.
4. Much of the changes in co-factor levels and specific histone marks are correlative. ChIP

sequencing to examine each of these marks and the specific genes that are regulated in each condition is important to move to mechanism. Further, many of these enzymes are only examined at the gene expression level and not protein - it is important to examine that these changes in co-factors and the CD44/copper pathway actually impact protein/enzyme levels.

5. It is important to perform ChIP-PCR and examine actual gene regulation in the setting of the LCC-12 treated macrophages. Instead of just examining gross levels of a histone mark (i.e., H3K9me2), in addition the ChIP seq, it is critical to examine the specific regulation of inflammatory genes under these conditions.

6. IN the experiments in Fig 6, the in vivo data with sepsis/sars, these macrophages should be isolated and protein levels of the epigenetic enzymes should be examined, as well as ChIP PCR for these histone marks on various inflammatory genes and other pathways that were shown to be regulated in earlier figures.

Minor:

1. The introduction moves around a bit and is too vague about macrophage plasticity in vivo and the complexity of this subject.

3.

Referee #2 (Remarks to the Author):

In the manuscript by Solier, Muller, and Versini, the authors explore a molecular mechanism by which robust macrophage activation downstream of CD44 function depends on a potential novel metal uptake and specific trafficking to the mitochondrial matrix to drive NADH production. The authors provide new evidence for the significance of Cu in macrophage function that could be pertinent to normal physiology and disease. Importantly, Cu is a well established anti-microbial and the serum Cu trafficker CP is an acute phase reactant during infection. The studies presented here build on previous findings from several groups that found that metals bind HA, CD44 promotes Fe uptake, and the biguanide metformin can function as a Cu chelation. Overall, the experimental design is solid and rigorous. As noted below in the additional questions, the molecular mechanism suggested by the authors require additional data to be overwhelmingly convincing and thus this reviewer would suggest the following:

The authors provide data that suggests that CD44 mediated HA binding promotes increased Cu and Fe uptake in activate macrophages. While the data provided in Figure 1 and Extended Data Figure 1 indicate that CD44 is increased upon GM-CSF treatment while the major Cu transporter CTR1 and the major Fe transporter TfR1 remain unchanged, the authors should ensure the following to provide further evidence of the CD44 mediated Cu and Fe uptake:

1. Confirm in CD44 knockout cells that CTR1 and TfR1 mRNA and protein expression d not change.
2. Test whether other Cu and Fe transporter mRNA and protein expression levels changes.

Specifically, CTR2 and DMT1.

3. Provide a metal stripped HA to activated macrophages and confirm there is no other mechanism for internalization.

4. Provide additional information int he text to describe the experiment in Figure 1J. it is not clear

how this experiment tests whether CD44 HA binding promotes lysosomal Cu uptake. The authors would need to do this experimental in CD44 knockdown cells and HA protonated media.

5. Additional controls in which CTR1, TfR, and DMT1 are knocked out and the contribution of CD44 to intracellular Cu and Fe should be provided.

6. The authors should confirm if the studies shown in Figure 1 f and 1g are done in the presence of HA. If not the HA-mediated increase in Cu uptake doesn't seem significant from 1f and 1g.

The authors provide evidence that the cellular plasticity of macrophages is dependent on mitochondrial Cu(II). These data draw from the utilization of the biguanide metformin and novel derivatives synthesized by the group. It's clear that metformin and the analogs specifically reduce CD86 expression in Figure 2 and that LCC analog treatment is targeted to the mitochondrial. The utilization of LCC derivative with an alkyne handle that employs the endogenous Cu pool is very clever. It would be great if the authors provided an additional genetic experiment to support the importance of mitochondrial Cu to their phenotype. Specifically, the authors should do the following:

1. Test whether mitochondrial Cu elevation in activated vs. non activated macrophage is dependent on CD44.
2. The authors should confirm that the metformin and the LCC analogs reduce mitochondrial Cu levels via an additional methods.
3. The authors should test whether knockdown of a mitochondrial Cu transporter impacts the efficacy of the LCC analogs and can alter macrophage plasticity.
4. The should should test whether knockdown of the mitochondria Cu transporter prevents LCC imaging.

It's unclear why the cells would prioritize delivery to the mitochondria. The authors should test whether author Cu pools are impacted in activated macrophages to better understand the increase in that singular compartment.

Given the importance of mitochondrial Cu to electron transport chain via the most evolutionarily conserved Cu dependent enzyme cytochrome c oxidase, the authors next interrogated whether the elevated Cu levels in activated macrophages plays a role in mitochondrial metabolism. The authors provide evidence for an interaction between Cu(II) and NADH that is required for its recycling during macrophage activation. The mechanism proposed depends on elevated SOD2 levels and activity. Thus the authors should test the following:

1. Does knockdown of SOD2 or over expression of catalase alter macrophage plasticity?

Recent studies have demonstrated that increase mitochondrial Cu levels can induce cell death via cuproptosis. The authors should at a minimum speculate on why macrophages can tolerate this increase in mitochondrial Cu and test whether lipolyated TCA cycle intermediates do not aggregate in this cell type and if so provide a hypothesis for this lack of translation of these findings to these contexts.

Minor point:

1. The authors haven't provided evidence for a druggable Cu signaling pathway that is dynamic. I would consider changing the title to better describe the findings.

Referee #3 (Remarks to the Author):

This paper describes a novel, interesting role for copper in macrophage activation. The authors used an array of techniques to show that copper is acquired by cells through CD44, and that it facilitates the oxidation of mitochondrial NADH into NAD by H₂O₂. They developed a metformin-derivative that binds copper, and in so doing inhibits inflammatory macrophage activation. They show that this small molecule (LCC-12) results in altered levels of TCA cycle metabolites that are important for demethylase and acetyl-transferase enzymatic activities critical for epigenetic changes that are essential for the expression of genes downstream of a variety of inflammatory stimuli.

I have a number of specific questions about the findings:

1) The authors argue that binding of Cu to HA underlies its ability to bind to CD44, and show that hyaluronic acids (HA) potentiate Cu uptake. Physiologically, this is envisaged as occurring in the context of upregulated CD44 expression driven by activation. However, if I understand correctly, most of the experiments are performed without added HA, and without added copper. Is this the case? If it is, the authors should discuss expected copper availability under these circumstances. Are the cells making HA, is there sufficient Cu in the tissue culture medium? The ability of LCC-12 added to macrophages in regular medium to block activation is interesting. Do the authors interpret this as the drug blocking the "chemically reactive pool of copper in mitochondria" or preventing uptake of whatever Cu may be in the medium?

2) How do the authors envisage Cu crossing the membrane to enter the cell? Presumably once CD44 has delivered Cu into the endocytic/lysosomal compartment, a transporter has to take over to get the metal across the membrane. Is this mediated by Ctr1? Does Ctr1 interact with CD44 (see the recent 2022 paper by Das, Ash...et al. in *Nature Cell Biology*, 24:35)? I would like to see an attempt to phenocopy the effects of LCC-12 by knocking down expression of Ctr1.

3) Once MDMs have been activated and cellular stores of Cu are increased, is there a return to homeostasis? There is a known role for ATP7B in Cu efflux, and it might be interesting if the expression of this gene is regulated following inflammatory activation.

4) Why didn't D-Pen or ATTM have any effect on cellular activation? Does it reflect their poor ability to chelate copper, or their inability, compared to LCC-12, to enter cells and their mitochondria? Can the authors test additional copper chelators?

5) We should know whether well studied properties of metformin, such as its ability to activate AMPK, are related to its ability to bind copper, and/or whether LCC-12 shares these properties.

6) GAPDH is an NAD-dependent enzyme that is critical for inflammatory macrophage activation. The extent to which the effects of Cu on H₂O₂-mediated oxidation of NADH are important for the supply of NAD to support GAPDH activity should be defined. Does LCC-12 affect activation induced upregulation of glycolysis in inflammatory macrophages?

Minor points:

a. Extended data Fig. 2i – needs more discussion or references to link the proteins in the Western to EMP.

Author Rebuttals to Initial Comments:

Responses to the Referees:

We thank the reviewers for their insightful comments. These comments have been instrumental in the preparation of a more compelling article and considerably strengthen the manuscript. Please find below our responses to these comments, which we addressed. As per the editor’s request, we have reduced the length of the manuscript according to the Article format (4300 Words, 5 Figures, 50 References, 10 Extended Data Figures). We hope you will find this revised version suitable for publication.

Referee #1 (Remarks to the Author):

In this well-written and novel paper the authors report that macrophage inflammation is controlled by a CD44 mediated pathway where CD44 acts as copper transporter and increased copper in the mitochondria results in NAD+ that promotes epigenetic changes that serve to promote inflammation. Further, they developed a drug that regulates copper uptake and decreases inflammation in macrophages in the setting of murine sepsis and covid. Although, the paper flows logically it requires several experiments to validate the conclusions - particularly related to the loose link between changes in metabolite co-factors, alterations of demethylases/HATs and alterations in inflammatory gene expression programs. Although there are observational links, additional experiments, particularly ChIP seq would add value in mechanistically examining the regulation of these enzymes by copper uptake. I have the following specific major comments:

We thank this reviewer for her/his insightful comments, which prompted us to perform several additional experiments. The new data obtained strengthen our conclusions and extend the scope of this study.

1. For the experiments in Figs 1 and 2, I think a control is missing - the authors use activated (with LPS/IFNg) and non-activated macrophages, but they fail to examine macrophages activated towards an anti-inflammatory or M2-like phenotype. In this regard, I do believe it is important to see macrophages treated with IL4/IL13 to determine if it is indeed the skewing towards inflammatory/M1-like macrophages that is involved in the CD44 uptake of copper or if any skewing of macrophages can induce this mechanism.

Thank you for raising this important point. In fact, we had investigated activation of other immune cells (dendritic cells, T cells) but not M2-like macrophages. In this study, we have shown that immune cell activation and cancer cells undergoing epithelial-mesenchymal transition

upregulate CD44, and that the novel mitochondrial copper(II)-targeting compound LCC-12 antagonizes these processes. Now, as specifically requested by the reviewer, we have investigated activation of M2-like macrophages using IL-4.

Under these conditions, macrophages also increase CD44/copper uptake, although this is less pronounced than in activated M1-polarized macrophages. Activation of M2-like macrophages is partly blocked by LCC-12 as indicated by the quantification of specific cell surface markers (see new **Extended Data Fig. 4a,b** and data presented on the left **for Reviewers only**). **These new data validate the general nature of the copper-signaling pathway we describe, being at work in distinct cell types.**

In this current study, however, we used LCC-12 in inflammatory settings (in vitro and in vivo), where macrophages exhibit a M1-polarized phenotype. In future work it would be interesting to explore the effect of LCC-12 against M2-like macrophages that are abundant in other settings, for instance in the tumor microenvironment. Notably, we have now performed RNA-seq experiments on M2-like non-activated (naMDM2) and activated macrophages (aMDM2), comparing their transcriptomic profiles to M1-polarized macrophages. This revealed that treating activated monocyte-derived macrophages (aMDM, generated in a pro-inflammatory settings) with LCC-12 dampens the inflammatory phenotype and that these LCC-12-treated aMDM cells exhibit a different phenotype compared to M2-like macrophages (see data presented on the right **for Reviewers only**). These data support the notion that LCC-12 antagonizes M1 activation and inflammation, without promoting an M2-like anti-inflammatory polarity.

2. In later figures in the paper, LLC-12 is used in both a murine sepsis (two models) and covid setting - For Figure 1k, only LPS induced sepsis is used to examine CD44 mediated HA uptake. Since this paper is trying to link this mechanism to both sepsis and sars infection, it would be important to show the CD44/copper levels in all of these conditions.

We thank the reviewer for her/his comment. This was indeed a very important dataset we failed to provide in our previous version of the manuscript. We have now conducted additional experiments where we isolated small peritoneal macrophages (SPM) from LPS-treated mice and mice subjected to CLP, as well as lung alveolar macrophages (AM) from SARS-CoV-2 infected mice. Immunoblotting and ICP-MS revealed that CD44 and cellular copper levels are increased in inflammatory macrophages of all three preclinical models (see new **Figure 5a-c**), supporting the proposed mechanism.

3. In Figure 5, it would be important to also add a control for the RNAseq that examines conditions that alter the CD44/copper pathway and compare the RNAseq to activated/ non-activated macrophages.

This was again an important piece of information that we did not provide initially, as the role of CD44 in inflammation had been mentioned in the literature, although without mechanistic rationale (see new **ref 14**). We have now conducted knock out experiments in primary macrophages and investigated gene expression by RNA-seq as well as proteins involved in inflammation by immunoblotting. These new data indicate that knocking out CD44 antagonizes the expression of inflammatory genes we found to be up-regulated in aMDM compared to naMDM, and impact on permissive and repressive histone marks. Taken together, this further supports the mechanism that we propose (see new **Figure 4h and Extended Data Fig. 9h,j**).

4. Much of the changes in co-factor levels and specific histone marks are correlative. CHIP sequencing to examine each of these marks and the specific genes that are regulated in each condition is important to move to mechanism. Further, many of these enzymes are only examined at the gene expression level and

not protein - it is important to examine that these changes in co-factors and the CD44/copper pathway actually impact protein/enzyme levels.

We agree with the reviewer. Without next generation sequencing (NGS) experiments, the data initially provided were incomplete and correlative. We have now conducted extensive experiments based on NGS, quantitative proteomics, and immunoblotting to support and further characterize the proposed mechanism. Specifically, we conducted ChIP-seq of several permissive (H3K27ac, H3K14ac, H3K9ac) and repressive histone marks (H3K9me2, H3K27me3). We explored the status of these marks in naMDM, aMDM and LCC-12-treated aMDM. We consistently observed the same readout across donors. We found that repressive marks are reduced and permissive marks increased in inflammatory genes upon macrophage activation and that this is reversed when aMDM are treated with LCC-12 (see new **Figure 4d,g and Extended Data Fig. 8e,f and 9e,f**). Furthermore, protein levels matched RNA-seq data under these conditions according to quantitative proteomics and immunoblotting (see new **Extended Data Fig. 8a,b and 9a**). Specifically, quantitative proteomics revealed upregulation of inflammatory proteins, effectors of the described copper-signaling pathway (e.g., CD44, SOD2, SLC25A3, acetyl transferases and demethylases) in aMDM. Immunoblotting also showed an upregulation of specific epigenetic writers and erasers of the histone marks studied (see new **Extended Data Fig. 8b,c**). Using immunoblotting, we now also validate that LCC-12 treatment leads to a reduction of key inflammatory proteins found to be upregulated in aMDM such as JAK2, STAT1/2, TRAF2 and CCL2 (see new **Extended Data Fig. 9a**). These new data strengthen our proposed model where mitochondrial copper(II) regulates the production of key metabolites, which are co-substrates of epigenetic writers and erasers required for the transcriptional regulation of inflammatory genes.

5. It is important to perform ChIP-PCR and examine actual gene regulation in the setting of the LCC-12 treated macrophages. Instead of just examining gross levels of a histone mark (i.e., H3K9me2), in addition the ChIP seq, it is critical to examine the specific regulation of inflammatory genes under these conditions.

We agree. See response to point 4 raised by this reviewer. Here, we have conducted extensive ChIP-seq experiments to identify genes regulated by specific histone marks as detailed above. Our results indicate that, within genes involved in inflammation, treatment with LCC-12 leads to a reduction of histone acetylation consistent with the reduction of acetyl-CoA, and increased methylation consistent with the reduction of α KG. Accordingly, we found that LCC-12 treatment leads to a reduction of gene expression (increased repressive marks and reduced permissive marks) at the RNA and the protein levels (see new **Figure 4g and Extended Data Fig. 9a,e,f**).

6. IN the experiments in Fig 6, the in vivo data with sepsis/sars, these macrophages should be isolated and protein levels of the epigenetic enzymes should be examined, as well as ChIP PCR for these histone marks on various inflammatory genes and other pathways that were shown to be regulated in earlier figures.

The revised manuscript now contains only 5 main Figures as we have merged previous Figures 2 and 3 according to editorial requirements. As requested by this reviewer, we have now isolated small peritoneal macrophages from LPS and CLP models as well as alveolar macrophages from SARS-CoV-2-infected mice. This allowed us to monitor the increase in CD44 and copper content in inflammatory macrophages in all three models. We have also been able to isolate enough cells to evaluate the variation of key proteins in

all three models. Immunoblotting revealed an increase of effectors of the proposed copper-signaling pathway including SOD2, hyaluronan synthases (HAS), a reduction of copper exporter proteins ATP7A/B, key epigenetic writers and erasers (epigenetic enzymes) that target the permissive and repressive marks studied (see new **Extended Data Fig. 10a-c**). We have also been able to validate the loss of repressive marks and the increase of permissive marks following the same pattern observed in vitro for primary macrophages upon activation. These data confirm that the mechanism identified in vitro is also at work in vivo.

Remarkably, in LPS-treated mice, we found that inflammatory macrophages are characterized by an increase of permissive acetyl marks and a reduction of repressive methyl marks, and that this was reversed upon treatment with LCC-12 (see new data **Figure 5d**). RNA-seq experiment on isolated SPM revealed that LCC-12 treatment indeed reduced the expression of inflammatory genes previously found to be upregulated in aMDM and this was confirmed at the protein level (see new data **Figure 5e,f**). Taken together, these data corroborate that the initial finding obtained in aMDM is at work in vivo using similar conditions.

Note that we were unable to perform ChIP-seq/PCR or cut&run using inflammatory macrophages from in vivo settings, being limited by the number of macrophages that could be isolated (See **protocol and QC data in ANNEX 1 at the end of this letter**). These techniques are routinely performed in our Institute and we have been successful using low cell numbers with other cell types but not with macrophages. It would take large cohorts of animals undergoing sepsis to establish a robust protocol. Nevertheless, pooling cells from mice undergoing acute inflammation enabled us to painstakingly optimize conditions to perform RNA-seq, ICP-MS on low cell counts as well as immunoblotting with low protein loading.

Importantly, using primary human monocyte-derived macrophages (MDM), ChIP-seq of key histone marks correlated with RNA-seq data in aMDM and cells treated with LCC-12. This validated our proposal that mitochondrial metabolites are rate-limiting, and directly impact on the epigenetic landscape that governs the expression of inflammatory genes. Thus, targeting mitochondrial copper provides the means to manipulate nuclear events with therapeutic benefits. Furthermore, we found that in the three preclinical models of inflammation, CD44 is upregulated and the copper contents is increased in inflammatory macrophages. We also found that key proteins of the proposed copper-signaling pathway are dysregulated in inflammatory settings along with key epigenetic enzymes and their targeted histone marks. Together, these data provide compelling evidence that the proposed copper signaling pathway is effective in vivo.

Minor:

1. The introduction moves around a bit and is too vague about macrophage plasticity in vivo and the complexity of this subject.

Thank you for flagging this. It is indeed a complex area. We have now rewritten the Introduction listing new references.

Referee #2 (Remarks to the Author):

In the manuscript by Solier, Muller, and Versini, the authors explore a molecular mechanism by which robust macrophage activation downstream of CD44 function depends on a potential novel metal uptake and specific trafficking to the mitochondrial matrix to drive NADH production. The authors provide new evidence for the significance of Cu in macrophage function that could be pertinent to normal physiology and disease. Importantly, Cu is a well established anti-microbial and the serum Cu trafficker CP is an acute phase reactant during infection. The studies presented here build on previous findings from several groups that found that metals bind HA, CD44 promotes Fe uptake, and the biguanide metformin can function as a Cu chelation. Overall, the experimental design is solid and rigorous.

We thank this reviewer for his/her helpful suggestions, which led to a substantially improved manuscript.

As noted below in the additional questions, the molecular mechanism suggested by the authors require additional data to be overwhelmingly convincing and thus this reviewer would suggest the following:

The authors provide data that suggests that CD44 mediated HA binding promotes increased Cu and Fe uptake in activate macrophages. While the data provided in Figure 1 and Extended Data Figure 1 indicate that CD44 is increased upon GM-CSF treatment while the major Cu transporter CTR1 and the major Fe transporter TfR1 remain unchanged, the authors should ensure the following to provide further evidence of the CD44 mediated Cu and Fe uptake:

1. Confirm in CD44 knockout cells that CTR1 and TfR1 mRNA and protein expression d not change.

We have now performed knock down of CD44 and confirmed that Ctr1 and TfR1 levels are not reduced under these conditions, meaning that targeting CD44 does not impact on metal uptake through an indirect mechanism involving other proteins such as Ctr1 or TfR1 (see new **Extended Data Fig. 1g**). Similar results were obtained under CD44 knock out conditions (see new **Extended Data Fig. 9i**).

2. Test whether other Cu and Fe transporter mRNA and protein expression levels changes. Specifically, CTR2 and DMT1.

Excellent suggestion indeed! Activation of macrophages did not lead to alterations in expression of Ctr2 or DMT1 at the RNA and protein levels (see new **Figure 1e and Supplementary Table 3**).

3. Provide a metal stripped HA to activated macrophages and confirm there is no other mechanism for internalization.

This is indeed an interesting suggestion. However a stripped HA would involve lowering the pH below the pKa of a carboxylic acid (4.5), which is the main functional group of HA that interacts with metals. Under these conditions, cells cannot be maintained in culture. Nevertheless, we believe that the reviewer's suggestion is important and instead adopted an alternative strategy to address her/his question. We synthesized permethylated HA to yield a surrogate that is not amenable to metal binding. We compared the effect of supplementing aMDM with HA versus methylated HA (meth. HA). We found that HA further

increased copper uptake upon macrophage activation, whereas meth. HA had no such effect (see new **Figure 1g and Extended Fig. 2b**). We believe that this control experiment addresses the reviewer's comment and validates our proposed mechanism.

4. Provide additional information in the text to describe the experiment in Figure 1J. It is not clear how this experiment tests whether CD44 HA binding promotes lysosomal Cu uptake. The authors would need to do this experiment in CD44 knockdown cells and HA protonated media.

We apologize for the lack of clarity. In this experiment, we used a lysosomal copper(II) probe to monitor copper uptake by endocytosis. In contrast to the Ctr1 copper transporter, we propose here that CD44 can mediate copper uptake by the dynamic process of endocytosis, where this metal traffics through the endo-lysosomal compartment. We also used a fluorescently labeled HA derivative to be able to monitor HA uptake by CD44-mediated endocytosis. Strikingly, labeled HA colocalizes with the copper(II) probe demonstrating that copper is present in the lysosome (see **Figure 1i**). This indicates a Ctr1-independent copper uptake pathway involving the HA receptor CD44. We now provide new data showing that hyaluronidase (HD), an enzyme that hydrolyzes HA, not only leads to a loss of lysosomal HA, as expected, but also to a reduced staining of lysosomal copper, indicating that CD44 mediates endocytosis of copper-bound hyaluronates (see new **Figure 1i**). These data are coherent with the upregulation of hyaluronan synthases (HAS) in aMDM and inflammatory macrophages isolated from mice (see new **Extended Data Fig. 2c and 10a,c**).

See also response to question 3 raised by this reviewer using methylated HA vs HA and monitoring copper uptake by ICP-MS.

As suggested by the reviewer, we now also provide additional data showing that knocking down CD44 reduces the fluorescence of the lysosomal copper(II) probe in aMDM (**Extended Data Fig. 2d**).

5. Additional controls in which CTR1, TfR, and DMT1 are knocked out and the contribution of CD44 to intracellular Cu and Fe should be provided.

As suggested by this reviewer, we now provide a comparative analysis of the effects of knocking down CD44 versus knocking down other metal transporters. We found this to be more reliable compared to a knock out method in primary cells where not all transporters can be effectively knocked out. Knocking down CD44 led to a reduction of Cu and Fe uptake, as defined by ICP-MS. In comparison, knocking down other metal transporters has a marginal effect on metal uptake (see new **Figure 1d and Extended Fig. 1e,f**).

6. The authors should confirm if the studies shown in Figure 1f and 1g are done in the presence of HA. If not the HA-mediated increase in Cu uptake doesn't seem significant from 1f and 1g.

This reviewer is correct stating that these experiments are conducted in the presence of HA, which is abundantly found in FBS. HA is also produced by cells upon activation (e.g., upregulation of hyaluronan synthases (HAS) (see new **Extended Data Fig. 2c and 10a,c**). The new data showing that hyaluronidase (HD) reduces lysosomal copper staining strongly support the proposed mechanism (see also answer to comment 4 raised by this reviewer).

The authors provide evidence that the cellular plasticity of macrophages is dependent on mitochondrial Cu(II). These data draw from the utilization of the biguanide metformin and novel derivatives synthesized by the group. It's clear that metformin and the analogs specifically reduce CD86 expression in Figure 2 and that LCC analog treatment is targeted to the mitochondrial. The utilization of LCC derivative with an alkyne handle that employs the endogenous Cu pool is very clever. It would be great if the authors provided an additional genetic experiment to support the importance of mitochondrial Cu to their phenotype. Specifically, the authors should do the following:

1. Test whether mitochondrial Cu elevation in activated vs. non activated macrophage is dependent on CD44.

We thank this reviewer for his/her supportive comments. As suggested, we performed the CD44 knockdown and monitored mitochondrial copper levels by ICP-MS showing a reduction of this metal in aMDM, further supporting the proposed mechanism (see new **Figure 2k**).

2. The authors should confirm that the metformin and the LCC analogs reduce mitochondrial Cu levels via an additional methods.

In fact, metformin and LCC-12 do not alter levels of mitochondrial copper. Rather, these compounds accumulate in mitochondria driven by the proton gradient (see CCCP experiment presented in **Figure 2g**), bind to copper(II) and inactivate copper-mediated catalysis. As suggested by this reviewer, we now provide data showing that the fluorescence of a mitochondrial copper(II) probe (which measures the chemical reactivity of copper(II) but not its absolute abundance) is reduced in aMDM treated with LCC-12 (see new **Extended Data Fig. 5j**). This data indicate that LCC-12 indeed binds copper(II) in mitochondria, thus lowering the availability of copper(II) that can react with the probe. Furthermore, ICP-MS quantification confirmed that LCC-12 does not reduce levels of total cellular and mitochondrial copper contents (see new **Extended Data Fig. 5h,i**), indicating that biguanides do not act as cuprophore.

3. The authors should test whether knockdown of a mitochondrial Cu transporter impacts the efficacy of the LCC analogs and can alter macrophage plasticity.

In the literature, mammalian SLC25A3 and its yeast ortholog Pic2 have been reported to act as mitochondrial copper transporters (J. Biol. Chem., 293, 1887–1896, J. Biol. Chem. 2013, 288, 23884–23892). Importantly, these transporters also transport phosphate ions, thus potentially impacting on key metabolic processes such as ATP synthesis (Am. J. Hum. Genet. 2007, 80, 478–484). In addition, other proteins might contribute to mitochondrial copper transport, such as SLC25A37 and SLC25A28 (Open Biol., 2016, 6, p. 150223), pending further confirmation. As suggested by this reviewer, we knocked down SLC25A3 in aMDM and investigated the impact on the inflammatory gene signature. We found that a series of inflammatory proteins that are upregulated in aMDM were downregulated under knock down conditions (see data presented here **for Reviewers only**),

supporting the idea that mitochondrial copper transport contributes to the acquisition of the pro-inflammatory phenotype.

4. The should test whether knockdown of the mitochondria Cu transporter prevents LCC imaging.

As suggested by this reviewer, we knocked down the putative copper transporters SLC25A3 and SLC25A37, which however did not alter the accumulation of LCC-12,4 in mitochondria, thereby confirming that it is not the presence of copper(II) in mitochondria but the proton gradient (see CCCP experiment in **Figure 2g**) that drives mitochondrial uptake of biguanides (see new **Extended Data Fig. 5l,m**). Importantly, CD44 knockdown, which leads to reduced mitochondrial copper levels (**Figure 2k**) also did not alter LCC-12 imaging (**Extended Data Fig. 5n**), further confirming that the presence of copper(II) in mitochondria does not drive LCC-12 accumulation in this organelle. Interestingly, we found SLC25A3 and SLC25A37 to be upregulated in aMDM, suggesting a functional roles in inflammation (see new **Extended Data Fig. 5k** and **Extended Data Fig. 8a**).

It's unclear why the cells would prioritize delivery to the mitochondria. The authors should test whether author Cu pools are impacted in activated macrophages to better understand the increase in that singular compartment.

Thank you for raising this point. Logically, cells would have evolved mechanisms allowing them to control the distribution of specific metals in distinct organelles to optimize their biological utility. This is likely the result of differential expression of metal transport proteins, chaperones, storage and export proteins. Here, we found that metallothioneins (see RNA-seq data of aMDM) are upregulated together with the mitochondrial copper and phosphate transporter SLC25A3. As requested by this reviewer, we studied the metal contents in other subcellular compartments of aMDM such as the endoplasmic reticulum and the nucleus. ICP-MS revealed no significant difference in copper content in these organelles upon activation (see new **Extended Data Fig. 5f,g**). Interestingly, this experiment revealed an increase of nuclear iron, supporting the contention that the increase of nuclear iron is a result of upregulated CD44-mediated metal uptake, which in turn fosters the catalytic activity of iron-dependent histone demethylases.

Given the importance of mitochondrial Cu to electron transport chain via the most evolutionarily conserved Cu dependent enzyme cytochrome c oxidase, the authors next interrogated whether the elevated Cu levels in activated macrophages plays a role in mitochondrial metabolism. The authors provide evidence for an interaction between Cu(II) and NADH that is required for its recycling during macrophage activation. The mechanism proposed depends on elevated SOD2 levels and activity. Thus the authors should test the following:

1. Does knockdown of SOD2 or over expression of catalase alter macrophage plasticity?

We thank the reviewer for this pertinent question. Following this suggestion, we knocked down SOD2 and this led to a reduction of inflammatory proteins, whose genes were found to be upregulated in aMDM (see new **Extended Data Fig. 9g**).

We obtained similar results using the panSOD inhibitor diethyldithiocarbamic acid (see data presented here **for Reviewers only**). We also found that N-acetyl cysteine (NAC), which can scavenge reactive oxygen species, antagonized macrophage activation (see data presented here **for Reviewers only**). Note that increase of SOD2 is associated with reduction of catalase and increase of hydrogen peroxide, consistently suggesting a functional role for this

metabolite (see **Figures 3b-d**). We also consistently found that SOD2 is upregulated at the RNA level in aMDM, in macrophages from sCOVID patients and MDM exposed to *Salmonella typhimurium*, *Leishmania major* or *Aspergillus fumigatus* (**Supplementary Tables 3 and 4**), and at the protein level in inflammatory macrophages in preclinical inflammation models elicited by LPS, CLP and SARS-CoV-2 model in vivo (**Extended Data Fig. 10 a-c**).

Recent studies have demonstrated that increase mitochondrial Cu levels can induce cell death via cuproptosis. The authors should at a minimum speculate on why macrophages can tolerate this increase in mitochondrial Cu and test whether lipolyated TCA cycle intermediates do not aggregate in this cell type and if so provide a hypothesis for this lack of translation of these findings to these contexts.

Thank you for raising this point. The report from Tsvetkov et al. is very recent. It describes that copper overload can lead to cancer cell death characterized by increased lipoylation and aggregation of specific proteins, in particular dihydrolipoamide S-acetyltransferase (DLAT). Here, we used primary monocytes isolated from human blood. Interestingly, patients with pneumonia/Covid-19 exhibit high levels of dead alveolar macrophages (personal communication by Prof. D. Annane who is a co-author of this manuscript and expert clinician). The cell death modalities at work under these conditions (e.g., cuproptosis, ferroptosis, apoptosis, necroptosis...) remain to be determined and represent an exciting avenue for future work.

As suggested by this reviewer, we studied the occurrence of DLAT aggregates in aMDM (see data presented here **for Reviewers only**), which contain increased mitochondrial copper levels. We detected a mild increase of such aggregates in aMDM compared to naMDM, yet cells remained viable at this time point. This finding indirectly supports an increase of mitochondrial copper content in aMDM according to the Tsvetkov report, in addition to our other extensive data sets. It is possible that the abundance of hydrogen peroxide is such that the amount of free copper(II) susceptible to trigger the production of aggregates is moderate. It is also possible that

DLAT aggregates have an unknown biological function, which becomes deleterious and leads to cell death at rather high levels as observed when cells are cultured in the presence of the cuprophore elesclomol. However, we did not detect extensive cell death when we cultured naMDM and aMDM. Note also that

this increase in DLAT aggregates is moderate compared to the one found by Tsvetkov et al. We have now cited this reference describing elesclomol as a cuprophore, differing from LCC-12, which accumulates in mitochondria and inactivates copper(II) but does not promote cellular copper transport (see new Ref. 38).

Minor point:

1. The authors haven't provided evidence for a druggable Cu signaling pathway that is dynamic. I would consider changing the title to better describe the findings.

We hope that the referee agrees that we have now provided substantial data showing that there is indeed a dynamic copper signaling axis at play, including endocytosis of copper by CD44/HA, copper transport into mitochondria potentially involving SLC25A3, and SOD2 as a hydrogen peroxide-producing enzyme required for copper-catalyzed interconversion of NAD(H) in mitochondria. Importantly, we also now convincingly show that knocking out or down these players impacts on this signaling axis and consequently on macrophage plasticity. In addition, we present extensive data sets showing that the mitochondrial copper(II) chelator LCC-12 blocks this signaling axis. Taken together, this convincingly shows that we uncovered a druggable copper-signaling pathway that drives inflammation, which we describe in the title. These data also show that the process is 'dynamic' (e.g., increase of CD44 leading to increase of copper, metabolic and dynamic epigenetic reprogramming), and amenable to therapeutic intervention, which we did not incorporate in the title due to length limitations.

Referee #3 (Remarks to the Author):

This paper describes a novel, interesting role for copper in macrophage activation. The authors used an array of techniques to show that copper is acquired by cells through CD44, and that it facilitates the oxidation of mitochondrial NADH into NAD by H₂O₂. They developed a metformin-derivative that binds copper, and in so doing inhibits inflammatory macrophage activation. They show that this small molecule (LCC-12) results in altered levels of TCA cycle metabolites that are important for demethylase and acetyltransferase enzymatic activities critical for epigenetic changes that are essential for the expression of genes downstream of a variety of inflammatory stimuli.

We thank this reviewer for her/his supportive and constructive comments.

I have a number of specific questions about the findings:

1) The authors argue that binding of Cu to HA underlies its ability to be bind to CD44, and show that hyaluronic acids (HA) potentiate Cu uptake. Physiologically, this is envisaged as occurring in the context of upregulated CD44 expression driven by activation. However, if I understand correctly, most of the experiments are performed without added HA, and without added copper. Is this the case? If it is, the authors should discuss expected copper availability under these circumstances. Are the cells making HA, is there sufficient Cu in the tissue culture medium? The ability of LCC-12 added to macrophages in regular medium to block activation is interesting. Do the authors interpret this as the drug blocking the "chemically reactive pool of copper in mitochondria" or preventing uptake of whatever Cu may be in the medium?

This reviewer correctly states that most of the experiments were conducted in absence of added hyaluronates and copper because both are naturally abundant in standard culture media supplemented with serum. This reviewer is also correct in stating that aMDM upregulate hyaluronan synthases (HAS) (**Extended Data Fig. 2c and Fig. 10a,c**) suggesting an increased production of HA upon macrophage activation. We now provide new data showing that supplementing aMDM with permethylated HA (meth.HA), which is not amenable to metal binding, does not promote metal uptake, in contrast to HA, which interacts with metals and enables copper uptake in aMDM. Furthermore, we show that hyaluronidase (HD), which can hydrolyze HA, antagonizes copper endocytosis (see new **Figure 1g,i and Extended Data Fig. 2b**).

Our data indeed indicate that LCC-12 inactivates copper(II) reactivity in mitochondria but does not affect cellular uptake of copper (see **Extended Data Fig. 5c** showing that excess LCC-12 blocks endogenous copper(II) catalyzed labelling of LCC-12,4). This is further supported by a new data showing that LCC-12 reduces the fluorescence of a mitochondrial copper(II) probe (see new **Extended Data Fig. 5j**). We further measured cellular and mitochondrial copper levels in absence and presence of LCC-12 by ICP-MS, and confirmed that LCC-12 does not impact copper uptake (see new **Extended Data Fig. 5h,i**).

2) How do the authors envisage Cu crossing the membrane to enter the cell? Presumably once CD44 has delivered Cu into the endocytic/lysosomal compartment, a transporter has to take over to get the metal across the membrane. Is this mediated by Ctr1? Does Ctr1 interact with CD44 (see the recent 2022 paper by Das, Ash...et al. in Nature Cell Biology, 24:35)? I would like to see an attempt to phenocopy the effects of LCC-12 by knocking down expression of Ctr1.

This reviewer is perfectly correct in stating that CD44, which is the main receptor of HA (Ref. 8), mediates endocytosis of copper-bound HA (see new **Figure 1i**). Once copper-bound HA has reached lysosomes, acidification of the vesicle can lead to unloading of copper from HA (see **Figure 1 h**). Then, presumably a copper transporter protein can transfer the metal from the lysosomal lumen to the cytosol. The literature describes Ctr2 as being a main lysosomal copper exporter protein (Biochem. J. 2007, 407, pp. 49–59). To support this idea, we studied the localization of Ctr2 in aMDM and confirmed that it colocalized with the endolysosomal marker Lamp2 (see new **Extended Data Fig. 2e**). Furthermore, knocking down Ctr2 in aMDM leads to an increased fluorescence of the lysosomal copper(II) probe supporting the contention that Ctr2 is the endolysosomal copper exporter protein in aMDM (see new **Extended Data Fig. 2f**). In accord with this interpretation, knocking down Ctr1 or Ctr2 in aMDM did not affect the global cellular content of copper in aMDM (see new **Figure 1d**).

3) Once MDMs have been activated and cellular stores of Cu are increased, is there a return to homeostasis? There is a known role for ATP7B in Cu efflux, and it might be interesting if the expression of this gene is regulated following inflammatory activation.

Thank you for raising this point, which has given us the opportunity to perform another series of experiments with compelling outcomes. Indeed we found that ATP7A and ATP7B are downregulated in inflammatory macrophages in vitro and in vivo (see new **Extended Data Fig. 2c and 10a,c**), correlating with downregulation of corresponding mRNA levels (see **Supplementary Table 3**).

Interestingly, we found that removing LPS and IFN γ from the media of aMDM leads to upregulation of ATP7A indicating return to homeostasis following inflammation. Consistently, copper levels decreased upon removal of LPS and IFN γ (see data presented here **for Reviewers only**).

4) Why didn't D-Pen or ATTM have any effect on cellular activation? Does it reflect their poor ability to chelate copper, or their inability, compared to LCC-12, to enter cells and their mitochondria? Can the authors test additional copper chelators?

This is indeed a very interesting point and we share this reviewer's hypotheses. Some copper chelators interact with copper(I) but not copper(II) (e.g., ability to chelate copper in an oxidation-state-specific manner, like ATTM that targets copper(I)). This is linked to the geometry of ligands and electron density (Orbital Theory). The other key aspect is the sub-cellular localization of the small molecules (e.g., inability to enter mitochondria or higher propensity to accumulate in another cellular compartment). In our experience, it is not the biological targets of drugs that define their subcellular localization but their inherent physicochemical properties (discussed Ref. 34). To support this, we have conducted additional experiments showing that knocking down CD44, which reduces mitochondrial copper, does not affect the accumulation of LCC-12,4 in mitochondria, contrary to treatment with CCCP, which dissipates the proton gradient on the inner mitochondrial membrane (see new **Extended Data Fig. 5n and Figure 2g**). We have also synthesized a derivative of the copper(II) chelator trientine that is amenable to in-cell-labeling and which is not effective at blocking macrophage activation. Strikingly, trientine alkyne forms complexes with copper(II) and accumulates in the nucleus but not in mitochondria of aMDM (see **Extended Data Fig. 5o,p**). Hence, it appears important in which organelle the interaction with copper(II) occurs. Following the comments of this reviewer, we have also tested additional copper chelators, including EDTA and trientine, which did not prevent activation of macrophages (see new **Extended Data Fig. 3a**).

5) We should know whether well studied properties of metformin, such as its ability to activate AMPK, are related to its ability to bind copper, and/or whether LCC-12 shares these properties.

According to this reviewer's comment, we studied the phosphorylation of AMPK (P-AMPK) in aMDM treated with metformin or LCC-12 and found that both compounds induced AMPK phosphorylation, yet LCC-12 was active at a dose 1000-fold lower than metformin (see new **Extended Data Fig. 3g**). Given that the higher potency of LCC-12 is linked to its dimeric form and enhanced capacity to bind to and to inactivate copper, AMPK activation at such a low concentration compared to metformin suggest that it is the result of copper targeting. Since NAD⁺ levels impacts on ATP metabolism and LCC-12 reduces overall

ATP levels (**Supplementary Table 1 and 2**), there is a link between copper(II) targeting and phosphorylation events, such as AMPK activation.

6) GAPDH is an NAD-dependent enzyme that is critical for inflammatory macrophage activation. The extent to which the effects of Cu on H₂O₂-mediated oxidation of NADH are important for the supply of NAD to support GAPDH activity should be defined. Does LCC-12 affect activation induced upregulation of glycolysis in inflammatory macrophages?

We thank the referee for raising this point. We have now compared the activity of metformin and LCC-12 on the accumulation of extracellular lactate, an end product of glycolysis, and glyceraldehyde-3-phosphate (G3P), an intermediate product of glycolysis that is metabolized by GAPDH. We found that metformin and LCC-12 led to reduced extracellular lactate and increased G3P (see new **Extended Data Fig. 6h,i**). This supports the contention that biguanides inhibit glycolysis in aMDM by reducing NAD⁺ levels, either contributing to or being concomitant with inhibition of activation. Note that LCC-12 triggered the same phenotype compared to metformin at a 1000-fold lower dose, suggesting that this effect is copper-related (see also answer to comment 5 raised by this reviewer).

Minor points:

- a. Extended data Fig. 2i – needs more discussion or references to link the proteins in the Western to EMP.

As requested by this reviewer, we now provide additional text and references as in:

The text now reads: *‘Human non-small cell lung carcinoma cells and murine pancreatic adenocarcinoma cells undergoing epithelial-mesenchymal transition (EMT), a cell biology program that can promote the acquisition of the persister cancer cell state and metastasis^{12,33}, were characterized by CD44 upregulation and increased cellular copper. Consistently, LCC-12 interfered with EMT, as shown by the levels of the epithelial marker E-cadherin, mesenchymal markers Vimentin and Fibronectin, the EMT-transcription factors Slug and Twist as well as the levels of pro-metastatic protein CD109 (Extended Data Fig. 4c,d). These data support the general nature of this copper-mediated regulatory mechanism.’*

ANNEX 1:

Material and Method

The experiments with the LPS mouse model were performed in accordance with French laws concerning animal experimentation (#2021072216346511) and approved by Institutional Animal Care and Use Committee (C2EA-47). A cohort of 30 mice with two groups (LPS and LPS + LCC-12 (0.3 mg/kg)) was used and SPM were isolated by flow cytometry from peritonea as described in the methods section. Cells were centrifuged at $1500 \times g$ for 5 min at 4 °C. Pelleted cells were resuspended in 300 μ L of nuclear extraction buffer (20 mM HEPES-KOH pH7.6, 10 mM KCl, 0.1% triton X-100, 20% glycerol, 0.5 mM spermidine, cOmplete) for 15 min on ice. Then, cells were incubated with concanavalin A-coated beads. In brief, 20 μ L concanavalin A-coated beads were washed twice in binding buffer (20 mM HEPES-KOH pH7.6, 10 mM KCl, 1 mM CaCl₂, 1mM MgCl₂) and resuspended in 150 μ L of binding buffer and subsequently mixed with 300 μ L of cell suspension (~25 000 cells) at room temperature for 10 min on a rotating wheel. After spinning and removal of supernatants, cells bound to beads were resuspended in 1mL of blocking buffer (20 mM HEPES-KOH pH 7.6, 150 mM NaCl, 0.1% BSA, 2 mM EDTA, 0.5 mM spermidine, cOmplete) at room temperature for 5 minutes. After a spinning, supernatants were discarded and cells bound to beads were resuspended in 200 μ L of washing buffer (20 mM HEPES-KOH pH 7.6, 150 mM NaCl, 0.1% BSA, 0.5 mM spermidine, cOmplete). This suspension was subsequently incubated with 200 μ L buffer with antibodies against specific histone marks H3K27ac (Cell signaling, 8173S), H3K27me3 (Cell Signaling, 9733S) and diluted in washing buffer (1:200). The mixture was incubated overnight at 4 °C on a rotating wheel. After spinning and removal of supernatants, cells bound to beads and complexed to antibodies were washed twice in 1mL of washing buffer, and then resuspended in 200 μ L of the same buffer. Subsequently, this suspension was incubated with 200 μ L of pA-MNase diluted in washing buffer (1:250) for 1h at 4 °C on a rotating wheel. After spinning and removal of supernatants, the complex was washed twice in washing buffer and resuspended in 150 μ L of the same buffer, and then equilibrated at 0 °C in an ice-water bath for 5 min. Subsequently, this suspension was incubated with 3 μ L of CaCl₂ for 30 min on ice. Next, the mixture was incubated with 150 μ L of stopping buffer (200 mM NaCl, 20 mM EDTA, 4 mM EGTA, 50 μ g/mL RNase A, 40 μ g/mL glycogen, 10 pg/mL spike-in genomic DNA of *Drosophila*) at 37 °C for 20 min to allow digested fragments to be released. Next, samples were centrifuged at $20000 \times g$ for 5 min at 4 °C and 300 μ L of supernatants were used for **DNA extraction**: samples were treated with 2.5 μ L of proteinase K 20 mg/mL, 3 μ L of SDS 10% at 70 °C for 30 min. DNA precipitation was carried out using phenol: chloroform:isoamyl alcohol. Samples were vortexed and centrifuged at $20000 \times g$ for 15 min at 4 °C. The upper phase was mixed with chloroform by vortexing. After centrifugation at $20000 \times g$ for 15 min at 4 °C, the upper phase was mixed with -20 °C absolute ethanol by vortexing and stored at -80 °C overnight. Next, samples were pelleted at $20000 \times g$ for 20 min at 4 °C. The pellets were washed with ice-cold 85% ethanol in dH₂O and centrifuged at $20000 \times g$ for 15 min at 4 °C. The supernatants were discarded and pellets were dried at room temperature, dissolved in 20 μ L of 1 mM Tris-HCl pH8.0 and 0.1 mM EDTA. **Library preparation and sequencing**: Libraries were prepared according to the manufacturer's protocol (Accel-NGS 2S Unique Dual Indexing DNA library kit, Swift Biosciences). The library amplification was modified to the following program: 45 seconds on 98°C, 21 cycles of 10 seconds 98°C, 15 seconds 60 °C and 1 minute 68 °C. End hold at 4 °C. Library sizes were tested on an Agilent 4200 TapeStation with D5000 ScreenTapes (Agilent). Quantification was performed on Invitrogen QUBIT high-sensitivity DNA kit. Libraries were sequenced on Illumina NovaSeq 6000 S1 using paired-end 100-bp run. The final pool was assessed by qPCR using the KAPA library quantification kit (Roche). Sequencing was performed on the NovaSeq 6000 (Illumina), targeting 30M clusters per sample and using paired-end 2 x 100 bp.

A-Library preparation

Protocol	Swift-ACCEL NGS- 2S PLUS DNA Lib
Sequencer	NovaSeq
Read	1 Units NS (200M)
Nb of Units / runs	1 Units NS (200M)
Primers	Illumina
PhiX	
Qubit (ng/µl) < 2 ng/µl	2,88
Pool molarity (nM) < 10 nM	16,5
Volume (µl)	85 µl
Mean size of pool (bp)	442
Sample Plan	
Number of samples	6

Description	I7_Index_ID	I7_Index_sequence	I5_Index_ID	I5_Index_sequence
Biological name (should be meaningful. Used to integrate & analyse data)	Name on the tube	NNNNNN	Name on the tube	NNNNNN
SPM LPS+LCC-12-H3K27me3	U073	AGTCTGTA	U073	TCAGTAGG
SPM LPS+LCC-12-H3K27ac	U074	CCGTATCT	U074	CTCGTTCT
SPM LPS+LCC-12-IgG rabbit	U075	CGCTTCCT	U075	AACAACCG
SPM LPS-H3K27me3	U076	CAAGACCT	U076	TCCTCATG
SPM LPS-H3K27ac	U077	CCTAGTAT	U077	ACTCTCCA
SPM LPS-IgG rabbit	U078	CCACCGAT	U078	CTCTATCG

B- Tapestation with D5000 ScreenTapes (lane 1: ladder; lane 2: library)

C- General Metrics: Calculated basic metric for the raw-qc pipeline

Sample ID	Biological name	Total reads	Sample fraction (%)	Mean length	Total base	>Q20 R1 (%)	>Q20 R2 (%)	Mean length after trimming	Trimmed reads (%)	Discarded reads (%)
D1205C050	SPM LPS+LCC-12-H3K27me3	58 227 044	15.0%	101	5 880 931 444	98.9%	98.9%	100	40.5%	1.4%
D1205C051	SPM LPS+LCC-12-H3K27ac	58 485 154	15.1%	101	5 907 000 554	98.9%	98.8%	100	40.6%	1.9%
D1205C052	SPM LPS+LCC-12-IgG rabbit	70 719 628	18.3%	101	7 142 682 428	99.1%	99.0%	100	40.1%	1.2%
D1205C053	SPM LPS-H3K27me3	81 504 782	21.0%	101	8 231 982 982	99.2%	99.1%	100	40.2%	0.9%
D1205C054	SPM LPS-H3K27ac	66 397 330	17.1%	101	6 706 130 330	99.3%	99.1%	100	40.0%	0.8%
D1205C055	SPM LPS-IgG rabbit	51 968 096	13.4%	101	5 248 777 696	99.4%	99.2%	100	39.4%	0.6%

D- Adapters: detected from the trimming report

Sample ID	Detected Adapter1	Detected Adapter2
D1205C050	AGATCGGAAGAGC	AGATCGGAAGAGC
D1205C054	AGATCGGAAGAGC	AGATCGGAAGAGC
D1205C055	AGATCGGAAGAGC	AGATCGGAAGAGC
D1205C052	AGATCGGAAGAGC	AGATCGGAAGAGC
D1205C053	AGATCGGAAGAGC	AGATCGGAAGAGC
D1205C051	AGATCGGAAGAGC	AGATCGGAAGAGC

E- Trimmed Sequence Lengths: this plot shows the number of reads with certain lengths of adapter trimmed

F- FastQC Per Sequence Quality Scores: the number of reads with average quality scores

G- FastQC Per Sequence GC content: the average GC content of reads.

H- Fastq screen

Mapped Reads

Reviewer Reports on the First Revision:

Referee expertise:

Referee #1: macrophage/epigenetics

Referee #2: copper biology

Referee #3: macrophage immunometabolism

Referees' comments:

Referee #1 (Remarks to the Author):

Nice revision addressing all of my comments.

Referee #2 (Remarks to the Author):

In the manuscript by Solier, Muller, and Versini, the authors explore a molecular mechanism by which robust macrophage activation downstream of CD44 function involves metal uptake and specific trafficking to the mitochondrial matrix to drive NADH production. The authors provide new evidence for the significance of Cu in macrophage function that could be pertinent to normal physiology and disease. The studies presented here build on previous findings from several groups that found that metals bind HA, CD44 promotes Fe uptake, and the biguanide metformin can function as a Cu chelation. Overall, the experimental design is solid and rigorous.

The authors have addressed the majority of this reviewers concerns. However, a few specific questions still remain.

1. The lysosomal Cu(II) probe utilized in Figure 1 and Extended Data Fig. 2c and 10a,c is based on hydrazine reactivity and unfortunately hydrazines also oxidize and react with aldehydes so can't distinguish between Cu(II) hydrolysis and other analytes. To confirm an increase in lysosome Cu, the author suggested that the authors test whether subcellular fractionation followed by ICP-MS yields the same result. This is especially important given the proposed mechanism of Cu uptake via CD44 bound HA.

2. Although the authors are correct that the mitochondrial localized transporter SLC25A3 also transport phosphate, it is the primary route of mitochondrial matrix Cu uptake and separation of function mutants have been identified. The authors should amend the text to indicate that SLC25A3 is a mitochondrial Cu transporter. The authors should include the data on SLC25A3 knockdown impacting inflammatory signaling proteins in macrophages. It is an important piece of this study and corroborates their findings and the importance of the mitochondrial Cu pool.

3. The uptake of LCC-12-4 by the mitochondria via a proton gradient is clear. However, how do the authors interpret their results that loss of SLC25A3 doesn't impact the LCC-12-4 reactivity with Cu in the mitochondria. Is this because it's only Cu(II) responsive and SLC25A3 would be important for Cu(I)? If Cu(I) is the primary oxidation state in the mitochondria, then is it that aMDMs have more Cu(II)? If that's true, then is the SLC25A3 knockdown phenotype for macrophage inflammatory signaling proteins unrelated to the gene expression changes the authors observe? In a related line of questions, based on the potential increase in Cu(II) increase in the lysosome that is trafficked to the mitochondria presumably, can the authors test whether Cu(II) is increased in the cytosol via newly developed Cu(II) probes?

3. The reviewer still doesn't agree with the authors that they have identified a new copper signaling axis. Instead it is this reviewer's opinion that they authors have identified a novel mechanism of Cu internalization and impacts mitochondrial biology and epigenetics. I would still suggest changing the title to reflect it.

Referee #3 (Remarks to the Author):

The revisions have improved the paper. I have a few minor points that I think should be addressed:

1) I don't think that the authors have looked at appropriate genes to be able to conclude that LCC-12 interferes with macrophage alternative activation (Ext Data 4).

2) Wouldn't the authors expect NADH levels to increase when NAD levels decrease after addition of LCC-12 (Fig. 3g)?

Author Rebuttals to First Revision:

We thank the reviewers for their additional comments that considerably helped to improve the manuscript.

Referee #1 (Remarks to the Author): Nice revision addressing all of my comments. Thank you.

Referee #2 (Remarks to the Author): The authors have addressed the majority of this reviewers concerns. However, a few specific questions still remain.

1. The lysosomal Cu(II) probe utilized in Figure 1 and Extended Data Fig. 2c and 10a,c is based on hydrazine reactivity and unfortunately hydrazines also oxidize and react with aldehydes so can't distinguish between Cu(II) hydrolysis and other analytes. To confirm an increase in lysosome Cu, the author suggested that the authors test whether subcellular fractionation followed by ICP-MS yields the same result. This is especially important given the proposed mechanism of Cu uptake via CD44 bound HA. Thank you for raising this interesting point. We note that this probe has been published (Ren et al. *J. Mater. Sci. B*, **3**, 6746, 2015) and successfully used. In our hands, it does not lead to a fluorescent product upon exposure to aldehydes and we found it to be stable during the time of our experiments. This probe is not a typical hydrazine but a less reactive acyl-hydrazine (see scheme below left panel). In our set up, cells are treated and then fixed with a large excess of formaldehyde, which can be considered a suitable control. Under these experimental conditions, the fluorescence of the probe does not increase upon fixation, indicating that this probe either does not react with aldehydes or that the product of the reaction with aldehydes does not change its fluorescence properties. From a reactivity standpoint, hydrazine can potentially react with aldehydes via a condensation reaction to yield a non-fluorescing hydrazone adduct (see scheme below, left panel), typically even more stable than the hydrazine starting materials. To control this, we now provide new data where this probe has been incubated in the presence of increasing amounts of various metals or formaldehyde. Our results show that only copper(II) leads to a fluorescent product of hydrolysis (see figure below, middle and right panels). Therefore this probe is a suitable tool to study lysosomal copper(II). Note that knocking down CD44 or using hyaluronidase, as suggested by this reviewer, reduces levels of lysosomal fluorescence (see Figure 1i and Extended Data Fig. 2d), which cannot be explained by a reaction with aldehydes. In contrast, knocking down the lysosomal copper exporter protein Ctr2 leads to increased lysosomal fluorescence of this probe supporting the fact that this probe becomes fluorescent upon increased copper exposure in lysosomes (see Extended Data Fig. 2e,f).

We had considered lysosome isolation and ICP-MS copper quantification before. Thank you for suggesting this experiment. Unfortunately, protocols to isolate endo-lysosomal vesicles have limitations and lead to co-isolation of other organelles. In addition, they require gradients that are prone to add substantial contaminations for ICP-MS experiments. Furthermore, while it may be informative for homogenous cells lines, there is a lot more variability with primary cells employed in this study. Since the lysosomal copper(II) probe is a suitable tool, and we showed extensively thanks to the previous helpful reviewer's comments the relevance of CD44-mediated copper uptake using knockdown approaches, hyaluronate supplementation, methylated hyaluronates and CD44-antibodies, we did not perform these additional experiments.

2. Although the authors are correct that the mitochondrial localized transporter SLC25A3 also transport phosphate, it is the primary route of mitochondrial matrix Cu uptake and separation of function mutants have been identified. The authors should amend the text to indicate that SLC25A3 is a mitochondrial Cu transporter. The authors should include the data on SLC25A3 knockdown impacting inflammatory signaling proteins in macrophages. It is an important piece of this study and corroborates their findings and the importance of the mitochondrial Cu pool. We agree. Thank you for this great suggestion. As requested by this reviewer, we have now included these new data as Extended Data Fig. 9h. The main text also indicates that SLC25A3 is a mitochondrial copper transporter (see line 275-276).

3. The uptake of LCC-12-4 by the mitochondria via a proton gradient is clear. However, how do the authors interpret their results that loss of SLC25A3 doesn't impact the LCC-12-4 reactivity with Cu in the mitochondria. Is this because it's only Cu(II) responsive

and SLC25A3 would be important for Cu(I)? If Cu(I) is the primary oxidation state in the mitochondria, then is it that aMDMs have more Cu(II)? If that's true, then is the SLC25A3 knockdown phenotype for macrophage inflammatory signaling proteins unrelated to the gene expression changes the authors observe? In a related line of questions, based on the potential increase in Cu(II) increase in the lysosome that is trafficked to the mitochondria presumably, can the authors test whether Cu(II) is increased in the cytosol via newly developed Cu(II) probes? We are pleased to see that this reviewer agrees with us that it is the proton gradient that drives accumulation of LCC-12 in mitochondria and not the mitochondrial copper content. To further validate this, we knocked down the copper transporter to deplete mitochondrial copper as requested by this reviewer and monitored localization of LCC-12,4 by means of in-cell click labeling/imaging. Because this is a copper catalyzed reaction and because we knocked down SLC25A3, we employed the standard protocol of click labeling that consists of adding exogenous copper(II)/Asc to fixed/permeabilized cells. This explains why knocking down SLC25A3 has no impact on in-cell labeling and LCC-12,4 imaging. We apologize that this was not more clearly written in our previous response. The purpose of these series of experiments, as we understood, was to monitor whether the mitochondrial copper content was driving LCC-12 accumulation in mitochondria. In Figures 2h and i, we used the endogenous copper pool to catalyze this reaction of labeling. This reaction worked only in activated macrophages and only when ascorbate was used. This demonstrates that copper(II) is increased in mitochondria of inflammatory macrophages. Targeting this pool with LCC-12, which forms complexes with copper(II) but not copper(I), altered metabolism and epigenetic programming of inflammation. This functionally validates the role of copper(II) in aMDM. We could certainly attempt to develop a cytosolic copper(II) probe as a follow-up study. However, experts in the field (e.g. Chang lab at Berkeley) are pioneering these type of probes and we shall reach out to them in the future.

4. The reviewer still doesn't agree with the authors that they have identified a new copper signaling axis. Instead it is this reviewer's opinion that they authors have identified a novel mechanism of Cu internalization and impacts mitochondrial biology and epigenetics. I would still suggest changing the title to reflect it. We agree with this reviewer's description of our findings. We now understand that this reviewer was referring to a more canonical copper-signaling where copper acts by means of a dynamic metalloallostery mechanism. We agree that in our case, copper(II) acts as a catalyst like an enzyme and not by allostery to activate an enzyme. We have clarified this point in the main manuscript and quote again the compelling review from Chang and co-workers (Ref 30, lines 318-320). We hope this reviewer will find this textual clarification in the manuscript acceptable. We agree that there are many cell signaling pathways involving allostery. Nevertheless, this is not always the case. For example interferon-gamma exerts its activity by inducing a dimerization of its receptor at the plasma membrane. According to Schreiber and others, many signaling events are the results of induction of proximity between chemical species. Here, copper(II) acting as a metal catalyst brings proximity between NADH and hydrogen peroxide, thereby reducing the energy barrier for oxidation. The activity of copper here, which is taken up from outside of cells, is reminiscent of that of an enzyme involved in signal transduction (e.g. JAK kinase), impacting on the production of metabolites required for epigenetic programming, transcription and acquisition of the inflammatory phenotype. We thank this reviewer for bringing this up, which has led to an improved manuscript.

Referee #3 (Remarks to the Author): The revisions have improved the paper. I have a few minor points that I think should be addressed: Thank you.

1) I don't think that the authors have looked at appropriate genes to be able to conclude that LCC-12 interferes with macrophage alternative activation (Ext Data 4). We agree with this reviewer that a complete gene expression analysis would certainly provide deeper insights on the effect of LCC-12 on M2-polarized macrophages. We have not performed these experiments as it was not initially requested by reviewer 1. Instead, this reviewer asked whether the CD44-copper axis was activated in M2-polarized macrophages. We have provided these data in the revised manuscript showing that CD44 increases upon M2 activation. We have investigated the effect of targeting copper with LCC-12 in this context and found that it leads to a reduction of M2-polarization marker CD163 as well as activation marker CD86 (See new Extended data Fig. 4a,b). This support the contention that mitochondrial copper(II) is involved in M2. We have altered the manuscript accordingly (lines 142-144).

2) Wouldn't the authors expect NADH levels to increase when NAD levels decrease after addition of LCC-12 (Fig. 3g)? This reviewer is correct stating that in untreated cells, NADH increases when NAD⁺ is consumed upon macrophage activation (untreated conditions). However, inactivating copper(II) with LCC-12 leads to a side reaction of NADH epoxidation in the presence of hydrogen peroxide (see Fig. 3e,g and Extended data Fig. 6a-c). This explains why LCC-12 treatment leads to a reduction of both NADH and NAD as detailed in the main text.

Reviewer Reports on the Second Revision:

Referees' comments:

Referee #2 (Remarks to the Author):

The authors have done exceptional job addressing my remaining concerns with new results or updates to the manuscript text. It's an impactful study that I look forward to seeing.

Referee #3 (Remarks to the Author):

The authors say on line 143 that "This treatment also altered the levels of cell surface markers characteristic of activated macrophages with an alternative anti-inflammatory polarity (Extended Data Fig. 4a,b)". This is an awkward sentence, and not really consistent with what the data are showing. Perhaps keep it vague? For example: "Next, we evaluated the effect of LCC-12 on other cell types that can upregulate CD44 upon exposure to specific biochemical stimuli. LCC-12 interfered with the activation of dendritic cells and T lymphocytes and the expression of several surface molecules on alternatively activated macrophages (Extended Data Fig. 4a,b)".

Author Rebuttals to Second Revision:

Prof. Raphaël Rodriguez, PhD, FRSC

Head of Chemical Biology of Cancer

Institut Curie

26 rue d'Ulm – 75005 Paris – France

CNRS UMR3666 – INSERM U1143

Phone: +33 648 482 191

Email: raphael.rodriguez@curie.fr

Web: <https://institut-curie.org/team/rodriguez>

Paris, March 9,
2023

Nature manuscript 2022-03-04136C

Responses to reviewers comments:

Referee #2 (Remarks to the Author):

The authors have done exceptional job addressing my remaining concerns with new results or updates to the manuscript text. It's an impactful study that I look forward to seeing. We thank this reviewer for her/his insightful comments.

Referee #3 (Remarks to the Author):

The authors say on line 143 that "This treatment also altered the levels of cell surface markers characteristic of activated macrophages with an alternative anti-inflammatory polarity (Extended Data Fig. 4a,b)". This is an awkward sentence, and not really consistent with what the data are showing. Perhaps keep it vague? For example: "Next, we evaluated the effect of LCC-12 on other cell types that can upregulate CD44 upon exposure to specific biochemical stimuli. LCC-12 interfered with the activation of dendritic cells and T lymphocytes and the expression of several surface molecules on alternatively activated macrophages (Extended Data Fig. 4a,b)". Thank you for pointing this out. We have now amended the manuscript accordingly.